

# Benthic Archaea as potential sources of tetraether membrane lipids in sediments across an oxygen minimum zone

Marc A. Besseling[1*], Ellen C. Hopmans[1], Jaap S. Sinninghe Damsté[1,2], and Laura Villanueva[1]

[1]NIOZ Royal Netherlands Institute for Sea Research, Department of Marine Microbiology and Biogeochemistry, and Utrecht University. P.O. Box 59, NL-1790 AB Den Burg, The Netherlands.

[2]Utrecht University, Faculty of Geosciences, Department of Earth sciences, P.O. Box 80.021, 3508 TA Utrecht, The Netherlands

*Correspondence to*: Marc A. Besseling (marc.besseling@nioz.nl)

**Abstract.** Benthic Archaea comprise a significant part of the total prokaryotic biomass in marine sediments. Recent genomic surveys suggest they are largely involved in anaerobic processing of organic matter but the distribution and abundance of these archaeal groups is still largely unknown. Archaeal membrane lipids composed of isoprenoid diethers or tetraethers (glycerol dibiphytanyl glycerol tetraether, GDGT) are often used as archaeal biomarkers. Here, we compare the archaeal diversity and intact polar lipid (IPL) composition in both surface (0–0.5 cm) and subsurface (10–12 cm) sediments recovered within, just below, and well below the oxygen minimum zone (OMZ) of the Arabian Sea. Archaeal 16S rRNA gene amplicon sequencing revealed a predominance of Thaumarchaeota (Marine Group I, MG-I) in oxygenated sediments. Quantification of archaeal 16S rRNA and ammonia monoxygenase (*amo*A) of Thaumarchaeota genes and their transcripts indicated the presence of an active *in situ* benthic population, which coincided with a high relative abundance of hexose phosphohexose crenarchaeol, a specific biomarker for living Thaumarchaeota. On the other hand, anoxic surface sediments within the OMZ and all subsurface sediments were dominated by archaea belonging to the Miscellaneous Crenarchaeota Group (MCG), the Thermoplasmatales and archaea of the DPANN superphylum. Members of the MCG were diverse with a dominance of subgroup MCG-12 in anoxic surface sediments. This coincided with a high relative abundance of IPL GDGT-0 with an unknown polar head group. Subsurface anoxic sediments were characterized by higher relative abundance of GDGT-0, 2 and 3 with dihexose IPL-types, as well as GDGT-0 with a cyclopentanetetraol molecule and a hexose, as well as the presence of specific MCG subgroups, suggesting that these groups could be the biological sources of these archaeal lipids.



## INTRODUCTION

Archaea are ubiquitous microorganisms in the marine system (DeLong et al., 1994). They occur in diverse environments, e.g. hydrothermal vents (Stetter et al., 1990), the marine water column (Karner et al., 2001; Massana et al., 2004), in the underlying sediments (Lloyd et al., 2013), and well below the seafloor (Biddle et al., 2006; Lipp et al., 2008), where they are considered key players in diverse biogeochemical processes (Offre et al., 2013, and references cited therein). Specifically marine sediments have been shown to contain a highly diverse archaeal community (Lloyd et al., 2013). The ammonia-oxidizing Thaumarchaeota of the marine group I.1a (further referred to as MG-I) is probably the most widely studied archaeal group in marine sediments. However, in comparison with studies of marine pelagic Thaumarchaeota, the diversity and distribution of benthic Thaumarchaeota is still quite unknown (e.g. Durbin & Teske, 2010; Jorgenson et al., 2012; Learman et al., 2016). Genomic studies have revealed the existence of uncultured archaeal groups other than Thaumarchaeota in marine, predominantly anoxic, sediments such as the Miscellaneous Crenarchaeota Group (MCG; Meng et al., 2014), archaea of the DPANN superphylum (composed of Micrarchaeota, Diapherotrites, Aenigmarchaeota, Nanohaloarchaeota, Parvarchaeota, Nanoarchaeota, Pacearchaeota and Woesearchaeota; Castelle et al., 2015; Rinke et al., 2013) and the Marine Benthic Group (MBG) B (Teske & Sørensen, 2008), and D (Lloyd et al., 2013). In the case of the archaea belonging to the groups of the MCG and MBG-D, metagenomic studies suggest that they are able to degrade extracellular proteins and aromatic compounds (Lloyd et al., 2013; Meng et al., 2014).

Archaeal diversity is currently determined through nucleic acid-based methods but the characterization of other cellular biomarkers such as membrane lipids has proven to be also effective in tracking the presence of archaeal groups in different ecosystems (e.g. Coolen et al., 2004a; Ingalls et al., 2012; Meador et al., 2015; Pitcher et al., 2011b; Sturt et al., 2004). One of the advantages of using lipid-based methods to determine the presence of archaeal groups is that lipids can be preserved in the sedimentary record. Therefore, they can also be used as biomarkers of the presence and metabolic potential of these microorganisms in past environments. On the contrary, other biomolecules like DNA have a more rapid turnover and they cannot be used for this purpose. In recent years, intact polar lipids (IPLs) have increasingly been applied for tracing 'living' bacteria and archaea in the environment (Lipp et al., 2008; Lipp and Hinrichs, 2009; Rossel et al., 2008). IPLs with polar head groups are present in living cells but upon cell lysis the polar head groups are lost, releasing the core lipids (CLs) that may be preserved in the fossil record. Since IPLs degrade relatively quickly after cell death (Harvey et al., 1986), it is possible to associate the presence of IPLs in the environment with the occurrence of their living producers (Lipp and Hinrichs, 2009; Schubotz et al., 2009).

Archaeal membrane lipids are typically a variation of two main structures, $sn$-2,3-diphytanylglycerol diether (archaeol) with phytanyl ($C_{20}$) chains in a bilayer structure, and $sn$-2,3-dibiphytanyl diglycerol tetraether (glycerol dibiphytanyl glycerol tetraether, GDGT), in which the two glycerol moieties are connected by two $C_{40}$ isoprenoid chains, allowing the formation of a monolayer membrane (Koga and Morii, 2007). GDGTs containing 0–4 cyclopentane moieties (Fig.





S1) are usually not exclusive to a specific archaeal group (Schouten et al., 2013) with the exception of the GDGT
crenarchaeol, containing 4 cyclopentane and one cyclohexane moiety, which is deemed to be exclusive to the
Thaumarchaeota phylum (De La Torre et al., 2008; Sinninghe Damsté et al., 2002, 2012). The newly described archaeal
groups detected by genetic methods are yet uncultured, therefore, their membrane lipid composition remains unknown.
In this study, we determined the archaeal diversity in a marine benthic system along a strong gradient in bottom water
oxygen concentrations and compared it with the diversity of archaeal lipids. We aimed to characterize changes in the
archaeal benthic community under different physicochemical conditions, as well as to provide clues on the potential
archaeal lipid biomarkers produced by uncultured benthic archaea. We analyzed sediments (surface 0–0.5 cm, and
subsurface 10–12 cm) of the Murray ridge in the Arabian Sea, which is impinged by one of the strongest present-day
oxygen minimum zones (OMZ). Previous studies observed changes in the diversity of archaeal lipids in the same
environmental setting in sediments under different oxygen and nutrient concentrations (Lengger et al., 2012; 2014). In
our study, we expand the repertoire of archaeal lipid diversity previously detected by Lengger et al. (2012; 2014) by re-
analyzing the samples with High Resolution Accurate Mass/Mass spectrometry (UHPLC-HRAM MS). In addition, we
determined the archaeal diversity by means of 16S rRNA gene amplicon sequencing, as well as the abundance and
potential activity of specific archaeal groups by quantitative PCR (QPCR) of 16S rRNA and the metabolic gene coding
for the ammonia monoxygenase (*amo*A gene) of Thaumarchaeota.
**MATERIAL and METHODS**
**Sampling**
Sediments were collected in the Northern Arabian Sea during the PASOM cruise in January 2009 with *R/V Pelagia*.
Sediment cores obtained with a multicorer were taken on the Murray ridge at four depths, 885 m below sea level (mbsl)
(within the OMZ), at 1306 mbsl (just below the OMZ), at 2470 mbsl and 3003 mbsl (both well below the OMZ) as
previously described by Lengger et al. (2012). Upon retrieval the cores were sliced in 0.5 cm resolution for the first 2
cm and at 2 cm resolution beyond 10 cm below the surface, and stored at -80°C until further analysis. For an overview
of the surface sediments physicochemical conditions see Table 1.
**Lipid extraction and analysis**
Total lipids were extracted from surface (upper 0–0.5 cm) and subsurface (10–12 cm) sediments after freeze-drying
using a modified Bligh and Dyer method (Bligh and Dyer, 1959) as previously described by Lengger et al. (2014). $C_{16}$-
PAF (1-O-hexadecyl-2-acetoyl-sn-glycero-3-phosphocholine) was added to the extracts as an internal standard and the
extracts were dried under a stream of nitrogen. The extracts with the added standard were then dissolved by adding
solvent (hexane:isopropanol:$H_2O$ 718:271:10 [v/v/v/v]) and filtered through a 0.45 µm, 4 mm-diameter True
Regenerated Cellulose syringe filter (Grace Davison, Columbia, MD, USA).





IPLs were analyzed according to Sturt et al. (2004) with some modifications. An Ultimate 3000 RS UHPLC, equipped

with thermostated auto-injector and column oven, coupled to a Q Exactive Orbitrap MS with Ion Max source with

heated electrospray ionization (HESI) probe (Thermo Fisher Scientific, Waltham, MA), was used. Separation was

achieved on a YMC-Triart Diol-HILIC column (250 x 2.0 mm, 1.9 μm particles, pore size 12 nm; YMC Co., Ltd,

Kyoto, Japan) maintained at 30 °C. The following elution program was used with a flow rate of 0.2 mL min$^{-1}$: 100% A

for 5 min, followed by a linear gradient to 66% A: 34% B in 20 min, maintained for 15 min, followed by a linear

gradient to 40% A: 60% B in 15 min, followed by a linear gradient to 30% A: 70% B in 10 min, where A = hexane/2-

propanol/formic acid/14.8 M NH$_{3aq}$ (79:20:0.12:0.04 [v/v/v/v]) and B = 2-propanol/water/formic acid/ 14.8 M NH$_{3aq}$

(88:10:0.12:0.04 [v/v/v/v]). Total run time was 70 min with a re-equilibration period of 20 min in between runs. HESI

settings were as follows: sheath gas (N$_2$) pressure 35 (arbitrary units), auxiliary gas (N$_2$) pressure 10 (arbitrary units),

auxiliary gas (N$_2$) T 50 ˚C, sweep gas (N$_2$) pressure 10 (arbitrary units), spray voltage 4.0 kV (positive ion ESI),

capillary temperature 275 °C, S-Lens 70 V. IPLs were analyzed with a mass range of $m/z$ 375 to 2000 (resolution

70,000 parts per million, ppm), followed by data dependent MS$^2$ (resolution 17,500 ppm), in which the ten most

abundant masses in the mass spectrum (with the exclusion of isotope peaks) were fragmented successively (stepped

normalized collision energy 15, 22.5, 30; isolation window 1.0 $m/z$). An inclusion list was used with a mass tolerance of

3 ppm to target specific compounds (Table S1). The Q Exactive Orbitrap MS was calibrated within a mass accuracy

range of 1 ppm using the Thermo Scientific Pierce LTQ Velos ESI Positive Ion Calibration Solution (containing a

mixture of caffeine, MRFA, Ultramark 1621, and $N$-butylamine in an acetonitrile-methanol-acetic acid solution).

The total lipid extract from the surface sediment at 885 mbsl was further analyzed by acid hydrolysis to determine the

composition and relative abundance of IPL-derived CL (resulting from the acid hydrolysis of IPLs) using the method

described by Lengger et al. (2012) and analyzed by HPLC-APCI/MS (according to Hopmans et al., 2016).

**Nucleic acids extraction, cDNA synthesis and quantitative PCR (QPCR) analyses**

Sediment was centrifuged and the excess of water was removed by pipetting before proceeding with the extraction of

nucleic acids from the sediment. DNA/RNA of surface (0−0.5 cm) and subsurface (10−12 cm) sediments was extracted

with the RNA PowerSoil® Total Isolation Kit plus the DNA elution accessory (Mo Bio Laboratories, Carlsbad, CA).

Concentration of DNA and RNA were quantified by Nanodrop (Thermo Scientific, Waltham, MA) and Fluorometric

with Quant-iT$^{TM}$ PicoGreen® dsDNA Assay Kit (Life technologies, Netherlands). RNA extracts were treated with

DNAse and reverse-transcribed to cDNA as described by Pitcher et al. (2011). Quantification of archaeal 16S rRNA

gene copies and $amo$A gene copies were estimated by QPCR by using the following primers; Parch519F and ARC915R

(archaeal 16S rRNA gene), CrenAmoAQ-F and CrenAmoAModR ($amo$A gene), as previously described (Pitcher et al.,

2011). For details on the QPCR conditions, efficiency and R$^2$ of the QPCR assays see Table S2.

**16S rRNA gene amplicon sequencing, analysis, and phylogeny**





PCR reactions were performed with the universal, Bacteria and Archaea, primers S-D-Arch-0159-a-S-15 and S-D-Bact-
785-a-A-21 (Klindworth et al., 2013) as previously described in Moore et al. (2015). The archaeal 16S rRNA gene
amplicon sequences were analyzed by QIIME v1.9 (Caporaso et al., 2010). Raw sequences were demultiplexed and
then quality-filtered with a minimum quality score of 25, length between 250–350, and allowing maximum two errors
in the barcode sequence. Taxonomy was assigned based on blast and the SILVA database version 123 (Altschul et al.,
1990; Quast et al., 2013). Representative operational taxonomic units (OTUs, clusters of reads with 97% similarity) of
archaeal groups were extracted through filter_taxa_from_otu_table.py and filter_fasta.py with QIIME (Caporaso et al.,
2010). The phylogenetic affiliation of the partial archaeal 16S rRNA gene sequences was compared to release 123 of
the Silva NR SSU Ref database (http://www.arb-silva.de/; Quast et al., 2013) using the ARB software package (Ludwig
et al., 2004). Sequences were added to the reference tree supplied by the Silva database using the ARB Parsimony tool.
MCG intragroup phylogeny for representative sequences of OTUs affiliated to the MCG lineage was carried out in
ARB (Ludwig et al., 2004). Sequences were added by parsimony to a previously-built phylogenetic tree composed of
reference sequences of the 17 MCG subgroups known so far (Kubo et al., 2012). Affiliation of any 16S rRNA gene
sequences to a given subgroup was done assuming a similarity cutoff of ≥85%.
**Cloning, sequencing and phylogeny of the archaeal *amo*A gene**
Amplification of the archaeal *amo*A gene was performed as described by Yakimov et al., (2011). PCR reaction mixture
was the following (final concentration): Q-solution 1× (PCR additive, Qiagen); PCR buffer 1×; BSA (200 μg ml$^{-1}$);
dNTPs (20 μM); primers (0.2 pmol μl$^{-1}$); MgCl$_2$ (1.5 mM); 1.25 U Taq polymerase (Qiagen, Valencia, CA, USA). PCR
conditions for these amplifications were the following: 95°C, 5 min; 35 × [95°C, 1 min; 55°C, 1 min; 72°C, 1 min];
final extension 72°C, 5 min. PCR products were gel purified (QIAquick gel purification kit, Qiagen) and cloned in the
TOPO-TA cloning® kit from Invitrogen (Carlsbad, CA, USA) and transformed in *E. coli* TOP10 cells following the
manufacturer's recommendations. Recombinant clones plasmid DNAs were purified by Qiagen Miniprep kit and
screening by sequencing (n ≥ 30) using M13R primer by Macrogen Europe Inc. (Amsterdam, The Netherlands).
Obtained archaeal *amo*A protein sequences were aligned with already annotated *amo*A sequences by using the Muscle
application (Edgar, 2004). Phylogenetic trees were constructed with the Neighbor-Joining method (Saitou and Nei,
1987) and evolutionary distances computed using the Poisson correction method with a bootstrap test of 1,000
replicates.
**RESULTS**
In this study, we analyzed both IPLs and DNA/RNA extracts from sediments previously collected along the Arabian
Sea Murray Ridge within the OMZ (885 mbsl), just below the lower interface (1306 mbsl), and well below the OMZ
(2470 and 3003 mbsl). The surface sediment (0-0.5 cm) at 885 mbsl was fully anoxic, however, the surface sediments





below the OMZ were partly oxygenated (1306 mbsl), and fully oxygenated at 2470 and 3003 mbsl (Table 1). The
subsurface sediments (10-12 cm) were fully anoxic at all stations (Table 1). For more details on the physicochemical
conditions in these sediments see Table 1.
**Archaeal IPL-GDGTs in the surface and subsurface sediments**
A range of IPL-GDGTs (GDGT-0 to 4 and crenarchaeol) with the IPL-types monohexose (MH), dihexose (DH) and
hexose-phosphohexose (HPH) was detected in surface and subsurface sediments across the Arabian Sea OMZ (Table
2). For the DH GDGT-0 two structural isomers (type-I with two hexose moieties at both ends of the CL, and type-II
with one dihexose moiety; Table 2) were detected and identified based on their mass spectral characteristics (Fig. S2).
In addition, GDGT-0 with both an ether-bound cyclopentanetetraol moiety and a hexose moiety as head groups was
identified (Fig. S2) in some sediments (Table 2). This IPL was previously reported as a glycerol dibiphytanyl nonitol
tetraether (GDNT; de Rosa et al. 1983) but was later shown to contain a 2-hydroxymethyl-1-(2,3-
dihydroxypropoxy)2,3,4,5-cyclopentanetetraol moiety by Sugai et al., (1995) on the basis of NMR spectroscopy
characterization.
In the surface sediment at 885 mbsl, crenarchaeol IPLs were dominant (44.6% of all detected IPL-GDGTs), occurring
predominantly with  DH as IPL-type (with a hexose head group on both ends; 43.1%; Table 2). IPL-GDGT-2 was the
second most abundant (29.6%), also mainly consisting of the IPL-type DH (29.5%; Table 2). IPL-GDGT-0,  -1, -3 and -
4 were occurring with relative abundances of 0.3%, 1.7%, 17.8% and 6.1%, respectively (Table 2). Overall, the
majority (98.1%; Table 3) of IPL-GDGTs in surface sediment at 885 mbsl with IPL-type DH (all with a hexose
molecule on both ends of the CL).
The surface sediment at 1306 mbsl contained mostly IPL-GDGT-0 (37.6%), almost entirely with the IPL-type HPH
(36.6%; Table 2). Slightly less abundant was the IPL-crenarchaeol (35.6%), with the IPL-types HPH (18.7%) and DH
type-I (15.5%) in equal amounts and with a minor relative abundance with MH (1.4%). Overall, the IPL-GDGTs in
surface sediment at 1306 mbsl mainly contained the IPL-types HPH (55.4%; Table 3) and DH (42.0%) (Table 3).
Well below the OMZ, surface sediments from 2470 and 3003 mbsl were both dominated by IPL-GDGT-0 (71.9 and
80.8%, respectively), predominantly with IPL-type HPH (Table 2; Fig. 1a). The IPL-crenarchaeol had a lower relative
abundance (26.6 and 17.6%, respectively) and again was dominated by the member with IPL-type HPH (Table 2). The
other IPL-GDGTs occurred in minor quantities (<1%). Overall, IPL-type HPH was, thus, by far the most abundant head
group detected in surface sediments at 2470 and 3003 mbsl (97.7% and 97.4%, respectively), in contrast to the other
two surface sediments studied (Table 3).
In all subsurface (10-12 cm) sediments (i.e. at 885, 1306, 2470 and 3003 mbsl) the most abundant IPL-GDGTs were
DH-crenarchaeol (28.9±3.8%; Table 2) and DH-GDGT-2 (25.5±3.5%; Table 2). DH was also the most commonly
observed IPL-type attached to GDGT-3 and GDGT-4 (Table 2). Overall the distributions of the IPL-GDGTs in all
subsurface sediments were relatively similar (Fig. 1a) in comparison to the substantial changes observed at the surface



(cf. Fig. 1a). Overall, the IPL-type DH was the predominant one detected in subsurface sediment with a relative
abundance ranging from 68.8% at 3003 mbsl to 90.1% at 1306 mbsl (Table 3). In contrast to all other sediments, in the
subsurface sediments at 885 mbsl and 1306 mbsl, two different isomers (Fig. S2) of the DH-GDGT-0 were detected
(Table 2). DH type-I (0.9% at 1306 mbsl) is also found in the other surface and subsurface sediments and in
combination with other core GDGT structures, whereas the other isomer (DH type-II) only occurs (7.8% at 885 mbsl;
1.8% at 1306 mbsl; Table 2; Fig. S2b). In addition, these subsurface sediments also contain small amounts of GDGT-0
with cyclopentanetetraol and MH head groups (IPL-type HCP; 1.6% at 885 mbsl; 0.4% at 1306 mbsl; Table 1; Fig.
S2c).
IPL derived CL-GDGTs from the 885 mbsl surface sediment were re-analyzed (after Lengger et al., 2012), in order to
exclude degradation within the stored samples, but the IPL derived CL-GDGTs composition showed the same
distribution as reported previously (Lengger et al., 2012) (p=1.00).
**Archaeal diversity in the surface and subsurface sediment**
Different archaeal groups were detected in surface and subsurface sediment across the Arabian sea OMZ. The surface
sediment at 885 mbsl, contained archaeal 16S rRNA gene sequences that were assigned to several archaeal groups (Fig.
1b). The most dominant group was MCG (Total 30.5%, 12.2% attributed to C3; also known as MCG-15, Kubo et al.,
2012). Another major group found was the DPANN Woesearchaeota Deep sea Hydrothermal Vent Group 6 (DHVEG-
6, 20.3%; Fig. 1b; Castelle et al., 2015). Marine Benthic Group (MBG) -B, -D and -E were also present with 12.2%,
7.7% and 6.9% of the archaeal 16S rRNA gene reads, respectively (Fig. 1b). Sequences affiliated to the Marine
Hydrothermal Vent Group (MHVG, 8.1%) of the phylum Euryarchaeota were also detected (Fig. 1b). Other groups,
with lower relative abundances, were Thermoplasmatales groups ANT06-05 (5.7%) and F2apm1A36 (3.3%) and the
DPANN Aenigmarchaeota (previously named Deep Sea Euryarchaeotic Group, DSEG; 1.6%; Fig. 1b).
Below the OMZ, in partly and fully oxygenated surface sediments at 1306, 2470 and 3003 mbsl (Table 1), the most
dominant archaeal group was Thaumarchaeota MG-I with relative abundances of 81.5%, 89.7% and 100%, respectively
(Fig. 1b). At 1306 mbsl other archaeal groups, such as MHVG (5.6%), Thermoplasmatales ASC21 (3.2%), DHVEG-6
(2.9%), MBG-B (2.4%) and MCG (1.3%) made up the rest of the archaeal community (Fig. 1b). At 2470 mbsl
DHVEG-6 (1.1%) was still detectable besides the MG-I (Fig. 1b).
In the subsurface sediments (10–12 cm), only the DNA extracted from the sediments at 885 and 1306 mbsl gave a
positive amplification signal. The archaeal composition of the subsurface (10–12 cm) sediments at 885 mbsl and 1306
mbsl was similar (Fig. 1b; Pearson correlation coefficient of 0.95), with most of the 16S rRNA gene reads classified
within the MCG (47.5% and 48.4%, respectively). Other archaeal groups, such as MBG-D (14.4% and 5.7%,
respectively), MBG-B (10.1% and 4.4%), the Woesearchaeota (7.8% and 10.4%), were also detected with comparable
relative abundances (Fig. 1b). Other archaeal groups such as Thaumarchaeota Terrestrial hot spring, the Euryarchaeota

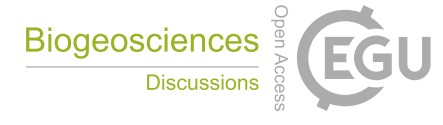

MHVG, MBG-E and the Aenigmarchaeota were detected but at low (< 10%) relative abundance (Fig. 1b). Only minor
amount of reads were classified as Thaumarchaeota MG-I (0.5% at 1306 mbsl) (Fig. 1b).
Considering the high relative abundance of the MCG detected in the surface sediment at 885 mbsl, as well as in the
subsurface (10–12 cm) sediments at 885 mbsl and 1306 mbsl (between 30.5-48.4% of total archaeal 16S rRNA gene
reads detected in those samples), we performed phylogenetic analyses to determine the diversity of subgroups of the
MCG within these sediments. A total of 57 representative 16S rRNA gene reads assigned to MCG were extracted from
the dataset and incorporated in a MCG phylogenetic tree of Fillol et al. (2015) (Fig. 2). The majority of MCG 16S
rRNA gene reads from the 885 mbsl surface sediment (77.3%; Table 4) clustered in subgroup 15. In the 885 mbsl
subsurface sediment, the majority of MCG reads clustered within subgroups 8 and 15 (33.6% and 19.6%, respectively;
Table 4). In the 1306 mbsl surface sediment there was only a low relative abundance of MCG (Fig. 1b); all MCG
archaea detected clustered in subgroup 15 (Table 4). On the other hand, in the 1306 mbsl subsurface sediment the reads
clustered in subgroups 15, 2 and 14 (34.3%, 10.9% and 10.9%, respectively; Fig. 2).
As the Thaumarchaeota MGI was dominant in oxygenated sediments at 1306, 2470 and 3003 mbsl (Fig. 1b), we further
analyzed the diversity of this group by performing a more detailed phylogeny of the recovered 16S rRNA gene reads
attributed to this group. Five OTUs dominated the Thaumarchaeota MG1 (Table 5); we will refer to them as OTU-1 to -
5. OTU-1, 2, 3 and 5 were phylogenetically closely related to other known benthic Thaumarchaeota MGI species, such
as '*Ca.* Nitrosoarchaeum koreensis MY1' or environmental 16S rRNA gene sequences from marine sediments (Fig.3).
On the other hand, OTU-4 clustered with 16S rRNA gene sequences from pelagic Thaumarchaeota MGI species, like
*Ca.* Nitrosopelagicus brevis, and also clustered with 16S rRNA sequences recovered from seawater SPM (Fig. 3).
OTU-3 was the most abundant OTU in the surface sediment at 1306, 2470, and 3003 mbsl with a relative abundance of
44-68% (Table 5). At 1306 mbsl OTU-4 was the second most abundant (35.1%). This OTU had a much lower relative
abundance (1.6% and 0.0%) at 2470 and 3003 mbsl, respectively (Table 5). The relative abundance of OTU-2 increased
with increasing sampling station depth (Table 5), OTU-1 and 5 had an abundance <5% in the surface sediments (Table

241    5).

The diversity of Thaumarchaeota MG1 was further assessed by amplification, cloning and sequencing of the archaeal
*amo*A gene. Most of the *amo*A gene sequences from surface (27 out of 29 clones) and subsurface sediment at 885 mbsl
(9 out of 10 clones) and just one from the surface sediment from 1306 mbsl (1 out of 58 clones) were closely related
with *amo*A gene sequences previously recovered from SPM at 1050 mbsl from this area of the Arabian Sea (Villanueva
et al., 2014). Phylogenetically they fall within the 'Water column B, subsurface water' *amo*A clade as defined by
Francis et al. (2005) (Fig. 4). At 1306 and 3003 mbsl (surface and subsurface) the majority of recovered *amo*A gene
sequences clustered within the 'shallow water/sediment' clade (100 and 98.3%, respectively) and are closely related
with *amo*A gene sequences from water column SPM at 170 mbsl (Villanueva et al., 2014) as well as *amo*A gene coding





sequences previously detected in sediments (Villanueva et al., 2014; Fig. 4). Of all recovered *amo*A gene sequences
from 885 mbsl only a small fraction (8.3%) clustered within the 'shallow water/sediment' clade (Fig. 4).
**Abundance and potential activity of archaea in surface and subsurface sediments**
The abundance of archaeal 16S rRNA gene copies in the surface sediments of different stations varied slightly: it was
lowest at 1306 mbsl ($9.8 \times 10^9$ copies g$^{-1}$ sediment) and highest at 2470 mbsl ($1.5 \times 10^{11}$; Fig. 5a). The potential
activity, based on the 16S rRNA gene transcripts of the archaeal 16S rRNA gene, was the lowest at 2470 mbsl ($5 \times 10^4$
transcripts g$^{-1}$ sediment), while a higher potential activity was detected at 885, 1306 and 3003 mbsl ($0.9$-$42 \times 10^7$; Fig.
5a). The abundance of archaeal 16S rRNA gene copies in the subsurface sediment varied also within one and a half
order of magnitude ($1.1$-$54 \times 10^9$; Fig. 5c), with a decrease with increasing water depth. The potential activity showed
less variation within the subsurface sediments ($1.2$-$22 \times 10^7$ 16S rRNA gene transcripts g$^{-1}$ of sediment; Fig. 5c) than in
the surface sediments.
The abundance of Thaumarchaeota was estimated by quantifying the archaeal *amo*A gene copies. The highest
abundance of *amo*A gene copies in surface sediment was detected at 2470 mbsl ($1.0 \times 10^9$ copies g$^{-1}$ sediment), and the
lowest at 885 mbsl ($5 \times 10^4$; Fig. 5b). *Amo*A gene transcripts in surface sediment were under the detection limit at 885
mbsl but were detected below the OMZ with $4 \times 10^2$, $2.3 \times 10^6$ and $8 \times 10^3$ gene transcripts g$^{-1}$ of sediment at 1306,
2470 and 3003 mbsl, respectively (Fig. 5b). In subsurface sediments, the abundance of *amo*A gene copies was low at
885 and 1306 mbsl ($5.4$-$19 \times 10^2$ gene transcripts g$^{-1}$ sediment) and higher at 2470 and 3003 mbsl ($4.1 \times 10^5$, $5.4 \times 10^6$,
respectively; Fig. 5d). *Amo*A gene transcripts were not detected in the subsurface sediments (Fig. 5d).
**DISCUSSION**
In this study, we assessed the changes in benthic archaeal diversity and abundance in sediments of the Arabian Sea
oxygen minimum zone along a gradient in bottom water oxygen concentrations. The steep Murray ridge protrudes the
OMZ, allowing the study of sediments deposited under varying bottom water oxygen concentrations but receiving
organic matter (OM), the most important fuel for benthic prokaryotic activity in sediments, produced in a relatively
small area of the ocean (i.e. the station within the OMZ, at 885 mbsl, and well below the OMZ, at 3003 mbsl, are only
110 km apart) and, therefore, likely composed of the same primary photosynthate. However, there were differences in
OM quality, as OM within the OMZ had a higher biochemical "quality" based on amino acid composition and intact
phytopigments compared to OM below the OMZ (Koho et al., 2013). Therefore, changes in the quality and flux of OM
received by the different sediment niches could also affect the archaeal community composition as several of the
archaeal groups (i.e. MCG and MBG-D) reported here have been suggested to use OM as carbon source in anoxic
conditions (Lloyd et al., 2013).
**Effect of oxygen availability on archaeal diversity and abundance in the surface sediments**



281 We detected large difference in archaeal diversity between the surface sediment deposited within the OMZ and those

282 deposited below the OMZ. In contrast to the diverse anaerobic archaeal community in the surface of 885 mbsl, in

283 surface sediments at 1306, 2470 and 3003 mbsl, Thaumarchaeota MGI were dominant, representing 80-100% of the

284 archaeal population (Fig. 1). This clear difference in the benthic archaeal population in the surface sediments can be

285 attributed to the oxygen availability as Thaumarchaeota are known to require oxygen for their metabolism (i.e.

286 nitrification; Könneke et al., 2005). In fact, the oxygen penetration depth (OPD) was observed to be 3, 10, and 19 mm

287 in sediments at 1306, 2470, and 3003 mbsl, respectively, while in sediments at 885 mbsl, the OPD was barely 0.1 mm

288 (Table S1; Kraal et al., 2012). The surface (0-5 mm) sediment at 1306 mbsl was not fully oxygenated (OPD of 3 mm),

289 which probably explains the detection in a relatively low abundance (ca. 20%) of the anaerobic archaea that thrive in

290 the anoxic sediment from 885 mbsl. The low OPD at 1306 mbsl also explains the low *amo*A gene expression in

291 comparison with the deeper surface sediments (Fig. 5b, d). Overall this indicates the presence of Thaumarchaeota with

292 lower activity at 1306 mbsl (Fig. 5). Within the Thaumarchaeota MG1 group, we also detected changes in the relative

293 abundance of specific OTUs in the surface sediments at 1306, 2470 and 3003 mbsl (Table 5). For example, OTU-2

294 becomes progressively more abundant with increasing water depth, suggesting that this OTU is favored at the higher

295 oxygen concentrations found in the surface sediment at 3003 mbsl. OTU-4 was closely affiliated with '*Ca.*

296 Nitrosopelagicus brevis', a pelagic MG-I member, which indicates that this DNA is most likely derived from the

297 overlaying water column (Table 5), and thus should be considered to represent fossil DNA.

298 High *amo*A gene abundances were detected in the surface sediment at 2470 and 3003 mbsl, while values in the surface

299 of 885 mbsl were approximately three orders of magnitude less. The lack of oxygen in the surface sediments at 885

300 mbsl and in the subsurface sediments, as well as undetectable *amo*A gene transcripts at those depths, suggest that in

301 these cases the *amo*A gene DNA signal is fossil. It is well known that under anoxic conditions DNA of marine pelagic

302 microbes may become preserved in sediments even for periods of thousands of years (Boere et al., 2011; Coolen et al.,

303 2004b). The fossil origin of the Thaumarchaeotal *amo*A gene is supported by the phylogenetic affiliation of the *amo*A

304 gene fragments amplified from the 885 mbsl surface sediment, as those sequences were closely related to *amo*A gene

305 sequences detected in the suspended particulate matter in the subsurface water column (Villanueva et al., 2014), thus

306 suggesting that the detected DNA originated from pelagic Thaumarchaeota present in the subsurface water column, as

307 proposed for the presence of OTU-4 16S rRNA gene sequences in the surface sediment (see earlier).

308 There is a discrepancy between the 16S rRNA gene copy numbers and the *amo*A gene copy numbers within the

309 sediments (Fig. 5). *Amo*A gene copies were consistently lower than the 16S rRNA gene copies, even within sediments

310 that were completely dominated by Thaumarchaeota MG-I. This may be caused by the *amo*A gene primer mismatches

311 and/or the disparity of gene copy numbers within the archaeal genomes (Park et al., 2008).

312 In the anoxic surface sediment at 885 mbsl (within the OMZ), we detected a highly diverse archaeal population

313 composed of MCG, Thermoplasmatales, MBG-B, -D and -E, Woesearchaeota, and MHVG. Archaeal groups such as





MCG and MBG-B and E have been previously described in anoxic marine sediments, where they have been suggested
to be involved in anaerobic OM degradation (e.g. Biddle et al., 2006; Inagaki et al., 2003; Castelle et al., 2015).
Members of the DPANN Woesearchaeota were only present in the surface sediment at 885 mbsl but not in the
subsurface anoxic sediments at 885 and 1306 mbsl, suggesting that their presence here is not solely dependent on the
absence of oxygen but possibly also of the OM composition and availability in surface and subsurface sediments.
Alternatively, the Woesearchaeota 16S rRNA gene signal could also originate from the water column and deposited in
the surface sediment at 885 mbsl as fossil DNA as observed for the case of Thaumarchaeota as mentioned above.
**Archaeal community composition in the anoxic subsurface sediments**
The archaeal diversity in the subsurface sediment (10–12 cm) in both 885 and 1306 mbsl (i.e. dominated by MCG,
MBG-B, -D and –E) is similar to that observed in the surface sediment at 885 mbsl. This supports that oxygen
availability is an important factor for determining the diversification of archaeal groups (Fig. 1b). MCG, one of the
dominant archaeal groups in these sediments showed substantial differences in the distribution of its subgroups. A high
relative abundance of the subgroup MCG-15 was noted in surface sediments, while the subsurface sediments had a
higher intra-group diversity. A recent survey of the ecological niches and substrate preferences of the MCG in estuarine
sediments based on genomic data pointed to MCG-6 archaea as degraders of complex extracellular carbohydrate
polymers (plant-derived) as detrital proteins, while subgroups 1, 7, 15 and 17 have mainly the potential to degrade
detrital proteins (Lazar et al., 2016). Lazar et al. (2016) also described the presence of aminopeptidases coded in the
genome bin of MCG-15, suggesting that this subgroup could be specialized in degradation of extracellular peptides in
comparison with the other MCG subgroups, which would be restricted to the use of amino acid and oligopeptides.
Considering the dominance of the MCG-15 subgroup in the surface sediments analyzed in this study, we can
hypothesize that the protein content in the OM deposited in the surface sediment, mainly originated from photosynthate,
could be still quite undegraded and therefore the MCG-15 would be favored in this niche and fueled its metabolism by
the degradation of peptides extracellularly, while in subsurface sediments, other MCG groups such as 2, 8 and 14 would
be more favored.
The archaeal 16S rRNA gene abundance progressively declined in subsurface sediments with increasing water depth,
while the potential activity was similar. This can be due to the expected decrease in the flux of OM being delivered to
these anoxic sediments layers attributed to higher degradation of OM in oxygenated bottom waters and the
progressively larger oxic zone in the sediments (Lengger et al., 2012; Nierop et al., 2017). This results in lower organic
carbon concentrations and a decreased biochemical quality of the OM (Koho et al., 2013; Nierop et al., 2017) to sustain
the heterotrophic archaeal population inhabiting the anoxic subsurface sediments. Also the presence or lack of
macrofauna in the analyzed sediments would have an effect on the OM composition, sediments within the OMZ are less
prone to bioturbation which most likely resulted in higher OM preservation (Koho et al., 2013). Differences in the OM



biochemical composition can influence the microbial community composition as was shown recently for North Sea
sediments (Oni et al., 2015).
**Benthic archaea as potential sources for archaeal IPLs**
Previous studies have evaluated the archaeal core lipid composition derived from IPLs and CLs as well as estimating
the different IPLs of crenarchaeol attributed to Thaumarchaeota in the same sediments studied here (Lengger et al.,
2012, 2014), as well as in the overlaying water column (Pitcher et al., 2011; Schouten et al., 2012). The studies by
Lengger et al. (2012, 2014) were limited to the determination of MH-, DH- and HPH-crenarchaeol with HPLC/ ESI-
MS[2] using a specific selected reaction monitoring method (SRM; Pitcher et al., 2011). In our study we expanded the
screening for IPLs carrying different polar head groups in combination with multiple CLs using high resolution
accurate mass/mass spectrometry (see Table S1). By applying this method, we unraveled the unknown diversity of IPL-
GDGTs in the sediments under study, which allows a more direct comparison with the archaeal diversity detected by
gene-based methods.
Fully oxygenated surface sediments showed a dominance of GDGT-0 and crenarchaeol mostly with HPH as IPL-type
(Table 1). This is the expected IPL-GDGT signature of Thaumarchaeota as previously observed in pure cultures
(Pitcher et al., 2010; Qin et al., 2015; Schouten et al., 2008; Sinninghe Damsté et al., 2012). The predominance of the
HPH IPL-type in these sediments was previously interpreted by Lengger et al. (2012) as an indication of the presence of
an active Thaumarchaeota population synthesizing membrane lipids *in situ* (Lengger et al., 2012 & 2014), taking into
account the labile nature of HPH IPLs (Harvey et al., 1986; Schouten, et al., 2010). This hypothesis is strongly
supported by (i) the fact that the archaeal community is dominated by Thaumarchaeota (Fig. 1) and (ii) the high
abundance of thaumarchaeotal *amo*A gene copies and gene transcripts detected in oxygenated sediments in our study.
On the other hand, in the anoxic surface sediment at 885 mbsl, crenarchaeol was predominantly present  with DH as the
IPL-type (Table 1). This was considered to be a fossil signal of Thaumarchaeota deposited from the water column due
to a higher preservation potential of glycolipid head groups (as present in DH) as previously suggested (Lengger et al.,
2012 & 2014). The presence of *amo*A gene sequences from the deep water column in the 885 mbsl surface sediment as
well as the much lower *amo*A gene abundance and lack of *amo*A gene expression (Fig. 5b, d) supports the contention
that the crenarchaeol IPLs are fossil.
We detected an increase in the ratio IPL-GDGT-0/IPL-crenarchaeol within the oxygenated surface sediments with
increasing water depth. This trend may be explained to be due to decreasing bottom water temperatures with increasing
water depth, which results in a higher ratio of GDGT-0/crenarchaeol (Schouten et al., 2002), and would thus point to a
population of Thaumarchaeota actively adjusting their membrane lipid composition according to environmental
conditions.
A striking result is the low relative abundance of GDGT-0 IPLs in the surface sediment at 885 mbsl (Table 1). Only
MH-GDGT-0 was detected in low relative abundance (0.3 %), whereas GDGT-0 with any other of the IPL-types that



were screened for in our study (Table S2; Fig. 1b) was absent. In contrast, Lengger et al. (2012) reported a significant
amount of IPL-derived CL-GDGT-0 (when the head groups are removed by acid hydrolyses and the remaining core
lipid is analyzed) in this surface sediment. This discrepancy must be due to the presence of an IPL-type with unknown
head groups not included in our analytical window. To test this hypothesis, we re-analyzed the IPL-derived CL-GDGT
composition in the surface sediment at 885 mbsl and we recovered an identical CL-GDGT distribution as reported by
Lennger et al. (2014), confirming the presence of a GDGT-0 of unknown IPL-type. This IPL could originate from any
of the archaeal groups present in the surface sediment at 885 mbsl, such as MCG, Thermoplasmatales, MBG-B, MBG-
E and Euryarchaeota MHVG. Woesearchaeota also occur relatively abundant in the surface sediments at 885 mbsl (Fig.
1) but recent studies suggest that their small genomes lack the genes coding for the enzymes of the GDGT biosynthetic
pathway (Jahn et al., 2004; Podar et al., 2013; Villanueva et al., 2017; Waters et al., 2003). Therefore, they are not
expected to contribute to the IPL-GDGT pool. Ruling out the Woesearchaeota as a possible source of IPL-GDGTs, the
IPL GDGT-0 with unknown polar head group(s) in the surface sediment at 885 mbsl may be attributed to the MCG,
which make up to 30.5% of the archaeal 16S rRNA gene reads in this sample. Most of these MCG archaea fall into
subgroup MCG-15 (Table 4). Previous studies proposed butanetriol dibiphytanyl glycerol tetraethers (BDGTs) as
putative biomarker of the MCG based on the correlation between this compound and the presence of MCG in estuarine
sediment (Meador et al., 2014). However, we did not detect any IPL BDGTs in the sediments analyzed in our study.
Buckles et al. (2013) also suggested that members of the MCG and Crenarchaeota group 1.2 could be the biological
source of IPL GDGT-0 found in the anoxic hypolimnion of a tropical lake. Considering these evidences, it is possible
that the unknown IPL GDGT-0 present in the surface sediment at 885 mbsl could be a biomarker for MCG. We also
detected a high relative abundance of MCG (up to 48.4% relative abundance) in surface and subsurface sediment at 885
mbsl and subsurface sediment at 1306 mbsl (Fig. 1). However, the diversity of the MCG population was higher in the
subsurface sediments in comparison with diversity in surface sediments at 885 mbsl as sequences closely related to the
MCG subgroups, 2, 8, 10, 14, 5b, 15, and 17 were detected both in the 885 mbsl and 1306 mbsl subsurface sediment
samples (Fig. 2). This niche differentiation of MCG members may also be the origin of the different DH-GDGT-0
moieties (and the one GDGT-0 with IPL-type HCP) that we detected within the subsurface sediments (Table 1). There
is also a high abundance of IPL-GDGT-0 present within SPM in the OMZ (Schouten et al., 2012). We, therefore,
cannot exclude the possibility that the detected IPL derived CL-GDGT-0 is derived from archaea in the water column.
In subsurface sediments, the IPL GDGT distribution was remarkably different from that detected in the surface
oxygenated sediment as higher relative abundances of GDGT-1, 2, 3 and 4 were detected in detriment of GDGT-0,
similar to the distribution detected in the surface sediments at 885 mbsl. It remains unclear whether this represents new
archaeal production in the anoxic sediments or selective preservation of archaeal lipids produced in the water column.
At 885 mbsl and at 1306 mbsl, in the subsurface sediment we also found two variants of the IPL-type DH GDGT-0 and
the IPL-type HCP (Fig. S2). GDGT-0 with the HCP IPL-type and the GDGT-0 with a DH headgroup attached were not



detected in any other sediments or in combination with other GDGTs. Since the GDGT-0 were not detected in the surface sediments it is likely that they are produced *in situ*. Unfortunately, we only obtained archaeal community compositions at those two depths so we cannot compare these with the samples missing these moieties. GDGT-0 with a cyclopentanetetraol head group has been previously detected in cultures of the hyperthermophilic crenarchaeal *Sulfolobales* (Langworthy et al., 1974; Sturt et al., 2004). However, the source(s) of the DH-GDGT-0 and the GDGT-0 with a cyclopentanetetraol head group remains unknown.

**CONCLUSIONS**

By using a combined 16S rRNA gene amplicon sequencing and IPL analysis with high-resolution accurate mass/mass spectrometry we have unraveled the high diversity of benthic archaea harbored specially in anoxic sediments, as well as increasing the repertoire of archaeal intact polar lipid detected. DNA-based analyses revealed a dominance of active benthic *in situ* Thaumarchaeota in those sediment where oxygen was present, which coincided with high relative abundance of the HPH-crenarchaeol previously suggested to be a marker of living Thaumarchaeota (Pitcher et al., 2011a). In the anoxic marine sediments analyzed here, members of the MCG, DPANN and Euryarchaeota Thermoplasmatales dominated. We also observed a high diversity within the MCG with a more diverse population in subsurface sediments. Surface anoxic sediments with an important MGC population coincided with a high abundance of an IPL GDGT-0 with a yet-unknown polar head group. This result, together with previous studies, suggests that this IPL GDGT-0 could be a potential biomarker for MGC. Besides, subsurface anoxic sediments had a high relative abundance of IPL GDGT-1, 2, and 3 with DH headgroups, which could be attributed to fossil signal due to the more recalcitrant nature of the glycosidic bonds (Schouten et al., 2010) or being IPLs synthesized by the archaeal groups detected in those sediments. In addition, unusual headgroups were also detected, as an IPL GDGT-0 with a hexose head group on both ends of the core lipid, two hexoses on one end, and a cyclopentanetetraol molecule bound to the core lipid and a hexose attached to it. Possibly ruling out the involvement of archaeal members of the DPANN in making those lipids due to the lack of lipid biosynthetic pathway (Podar et al., 2013; Waters et al., 2003), dominant archaeal members in those sediments such as the MCG and Thermoplasmatales, could be potential biological sources of these IPLs. To conclude, this combined approach has shed light on the possible biological sources of specific archaeal IPLs and also detected a higher diversity of benthic archaeal than previously thought.

**Acknowledgments**

Elda Panoto is thanked for assistance with molecular analyses. We would like to thank the captain and crew of the RV *Pelagia* as well as the cruise leader, technicians and scientists participating in cruise 64PE301. This PASOM cruise was funded by the Earth and Life Science and Research Council (ALW) with financial aid from the Netherlands Organization for Scientific Research (NWO) (grant 817.01.015) to Prof. G.J. Reichart (PI). NIOZ is acknowledged for



the studentship of MAB. This research was further supported by the NESSC and SIAM Gravitation Grants
(024.002.001 and 024.002.002) from the Dutch Ministry of Education, Culture and Science (OCW) to JSSD.

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



**Figure legends**

**Fig. 1.** (**A**) Relative abundances of the IPL-GDGTs (sum of the IPL-types MH, DH and HPH) for the different core GDGTs in the surface and subsurface sediments and (**B**) the archaeal community composition as revealed by 16S rRNA gene reads (with average abundance above of > 1%) in the surface sediments at 885, 1306, 2470, and 3003 mbsl and in the subsurface sediments at 885 and 1306 mbsl.

**Fig. 2.** Maximum likelihood phylogenetic tree of the archaeal groups MCG+C3 (modified from Fillol et al., 2015). Extracted OTUs from the Arabian Sea sediments assigned as MCG were inserted in the tree. The number of detected reads per OTU per samples are indicated. Per MCG subgroup the relative abundance is given as detected at the different stations and sediments depths, this is also noted in Table 4. Scale bar represents a 2% sequence dissimilarity.

**Fig. 3.** Maximum likelihood phylogenetic tree of MG-I OTUs recovered within the sediment based on the 16S rRNA gene (colored in blue). Sequences from cultured representatives of Thaumarchaeota MG-I are indicated in red. Environmental sequences of MG-I members are indicated in black with their origin specified. The relative abundances of the various OTUs are listed in Table 4. Scale bar represents a 2% sequence dissimilarity.

**Fig. 4.** Maximum likelihood phylogenetic tree of *amo*A gene coding sequences recovered from surface (S) and subsurface (SS) sediments (colored in blue) at 885 mbsl, 1306 mbsl and 3003 mbsl (155 clones). *Amo*A gene coding sequences recovered from SPM (colored in orange) at 170 mbsl (28 clones), SPM at 1050 (25 clones) reported by Villanueva et al. (2014). ** indicates *amo*A gene sequences recovered from surface sediments at 3003 mbsl previously reported in Villanueva et al., (2015). Scale bar represents a 2% sequence dissimilarity.

**Fig. 5.** Abundance of Thaumarchaeotal 16S rRNA (**A,C**) and *amo*A (**B,D**) gene fragment copies per gram of dry weight in the surface sediment (0-0.5 cm) (**A,B**) and the subsurface sediment (10-12 cm) (**C,D**). Black bars indicate the amount of DNA 16S rRNA or *amo*A gene fragment copies and the gray bars indicate the RNA (gene transcripts) of 16S rRNA or *amo*A gene fragment copies. Error bars indicate standard deviation based on $n = 3$ experimental replicates.



**Table 1. Bottom water temperature and oxygen concentration, oxygen penetration depth in the sediment, and**
**TOC content and pore water composition of the surface (0-0.5 cm) sediment[a]**

| Station (mbsl) | T (°C) | BWO ($\mu mol \cdot L^{-1}$) | OPD (mm) | TOC (wt %) | $NH_4^+$ ($\mu M$) | $NO_2^-$ ($\mu M$) | $NO_3^-$ ($\mu M$) | $HPO_4^{2-}$ ($\mu M$) |
|---|---|---|---|---|---|---|---|---|
| 885 | 10 | 2.0 | 0.1 | 5.6 (± 0.2) | 2 | 1.2 | 1.3 | 9.2 |
| 1306 | 6.7 | 14.3 | 2.9 | 2.9 (± 0.1) | 2.6[*] | 0.1[*] | 36.2[*] | 5.6 |
| 2470 | 2.1 | 63.8 | 9.8 | 0.8 (± 0.1) | -[b] | - | - | - |
| 3003 | 1.4 | 82.9 | 19 | 0.7 (± 0.1) | 55.6 | 8.3 | 46.2 | 3.8 |

[a] Data from Kraal et al. (2012) and Lengger et al. (2014)

[b] no data available







**Table 2. Heatmap[a] of the relative abundance (%) of the detected IPLs and sum (not color coded) per IPL-**
**GDGT.**

| Sample | Depth (mbsl) | GDGT-0 MH | GDGT-0 DH I[b] | GDGT-0 DH II[b] | GDGT-0 HCP[c] | GDGT-0 HPH | GDGT-0 Sum | GDGT-1 MH | GDGT-1 DH I[b] | GDGT-1 HPH | GDGT-1 Sum | GDGT-2 MH | GDGT-2 DH I[b] | GDGT-2 HPH | GDGT-2 Sum |
|---|---|---|---|---|---|---|---|---|---|---|---|---|---|---|---|
| Surface | 885 | 0.3 | ND[d] | ND | ND | ND | 0.3 | 0.1 | 1.6 | ND | 1.7 | 0.1 | 29.5 | ND | 29.6 |
| | 1306 | 1.1 | ND | ND | ND | 36.6 | 37.6 | 0.1 | 1.5 | 0.2 | 1.7 | ND | 15.4 | ND | 15.4 |
| | 2470 | 0.2 | 0.1 | ND | ND | 71.5 | 71.9 | 0.0 | 0.1 | 0.4 | 0.5 | ND | 0.8 | ND | 0.8 |
| | 3003 | 0.5 | 0.1 | ND | ND | 80.3 | 80.8 | ND | 0.2 | ND | 0.2 | ND | 0.9 | ND | 0.9 |
| Subsurface | 885 | 0.3 | ND | 7.8 | 1.6 | 2.1 | 11.9 | 0.1 | 1.7 | 0.1 | 1.9 | 0.2 | 27.0 | ND | 27.1 |
| | 1306 | 2.2 | 0.9 | 1.8 | 0.4 | 2.1 | 7.4 | 0.2 | 6.7 | ND | 6.9 | 0.1 | 29.7 | ND | 29.7 |
| | 2470 | 4.3 | 2.7 | ND | ND | 18.6 | 25.6 | 0.1 | 5.8 | ND | 5.9 | ND | 23.2 | ND | 23.2 |
| | 3003 | 9.1 | 3.4 | ND | ND | 13.0 | 25.5 | 0.2 | 4.3 | ND | 4.6 | ND | 21.9 | ND | 21.9 |


| Sample | Depth (mbsl) | GDGT-3 MH | GDGT-3 DH I[b] | GDGT-3 HPH | GDGT-3 Sum | GDGT-4 MH | GDGT-4 DH I[b] | GDGT-4 HPH | GDGT-4 Sum | Crenarchaeol MH | Crenarchaeol DH I[b] | Crenarchaeol HPH | Crenarchaeol Sum |
|---|---|---|---|---|---|---|---|---|---|---|---|---|---|
| Surface | 885 | ND | 17.8 | ND | 17.8 | ND | 6.1 | ND | 6.1 | 1.3 | 43.1 | 0.3 | 44.6 |
| | 1306 | 0.0 | 6.9 | ND | 6.9 | ND | 2.7 | ND | 2.7 | 1.4 | 15.5 | 18.7 | 35.6 |
| | 2470 | ND | 0.2 | ND | 0.2 | ND | 0.0 | ND | 0.0 | 0.2 | 0.6 | 25.8 | 26.6 |
| | 3003 | ND | 0.4 | ND | 0.4 | ND | 0.0 | ND | 0.0 | 0.4 | 0.2 | 17.1 | 17.6 |
| Subsurface | 885 | 0.1 | 15.9 | ND | 15.9 | ND | 9.4 | ND | 9.4 | 1.1 | 31.1 | 1.5 | 33.8 |
| | 1306 | 0.0 | 14.5 | ND | 14.5 | ND | 6.1 | ND | 6.1 | 2.7 | 32.4 | 0.4 | 35.5 |
| | 2470 | ND | 9.6 | ND | 9.6 | ND | 2.9 | ND | 2.9 | 3.5 | 28.3 | 1.0 | 32.8 |
| | 3003 | ND | 9.7 | ND | 9.7 | ND | 5.6 | ND | 5.6 | 8.2 | 23.9 | 0.6 | 32.7 |

[a] Green colors indicate a low relative abundance, red colors indicate a high relative abundance
[b] DH isomers were detected as a GDGT with a glycosidically-bound hexose moiety on both ends of the core (I) and
with one glycosidically-bound dihexose moiety on one end (II).
[c] HCP is an IPL-type with an ether-bound cyclopentanetetraol moiety on one end and an hexose moiety on the other
(previously reported as GDNT; e.g. De Rosa and Gambacorta, 1988; Sturt et al., 2004).
[d] ND = not detected





**Table 3. Relative abundance of IPL-GDGTs grouped by polar head group[a].**

| Sediment | Depth (mbsl) | MH | DH | HCP | HPH |
|---|---|---|---|---|---|
| Surface | 885 | 1.7% | 98.1% | ND[b] | 0.3% |
| | 1306 | 2.6% | 42.0% | ND | 55.4% |
| | 2470 | 0.5% | 1.8% | ND | 97.7% |
| | 3003 | 0.8% | 1.8% | ND | 97.4% |
| Deep | 885 | 1.8% | 92.9% | 1.6% | 3.7% |
| | 1306 | 5.2% | 91.9% | 0.4% | 2.5% |
| | 2470 | 7.9% | 72.6% | ND | 19.6% |
| | 3003 | 17.6% | 68.8% | ND | 13.6% |

Polar head group types detected: MH = monohexose, DH = dihexose, both isomers combined, HCP = monohexose and
cyclopentanetetraol, HPH = monohexose and phosphohexose.
[b] ND = not detected





**Table 4. Relative abundance (in %) of MCG and C3 assigned 16S rRNA gene reads relative to total archaeal**
**reads and distribution (in %) of various subgroups for a station within and a station just below the OMZ**

| Subgroup | 885 mbsl | | 1306 mbsl | |
|---|---|---|---|---|
| | surface | subsurface | surface | subsurface |
| | 30.5 | 47.5 | 1.3 | 48.8 |
| 1 | | 4.6 | | 8.6 |
| 2 | | 9.7 | | 10.9 |
| 3 | | <1 | | 2.3 |
| 4 | | <1 | | |
| 5b | | <1 | | |
| 8 | 2.3 | 33.6 | | 10.3 |
| 10 | | 13.4 | | 4.0 |
| 12 | 13.6 | 7.7 | | 8.0 |
| 13 | | 1.2 | | 2.3 |
| 14 | 2.3 | 3.1 | | 10.9 |
| 15 | 77.3 | 19.6 | 100 | 34.3 |
| 17 | 4.5 | 5.7 | | 8.6 |






**Table 5.** Total Thaumarchaeota MG-I 16S rRNA gene reads and distribution per OTU (%) in surface sediments.

| | Depth (mbsl) | | | |
|---|---|---|---|---|
| | 885 | 1306 | 2470 | 3003 |
| Total reads | 0 | 915 | 1341 | 1305 |
| OTU ID #1 | n.a.[a] | 4.3 | 2.5 | 3.0 |
| OTU ID #2 | n.a. | 3.9 | 8.1 | 13.6 |
| OTU ID #3 | n.a. | 43.6 | 67.6 | 61.8 |
| OTU ID #4 | n.a. | 35.1 | 1.6 | 0 |
| OTU ID #5 | n.a. | 3.3 | 4.7 | 2.1 |

[a] n.a. = not applicable
**Fig. 1.**

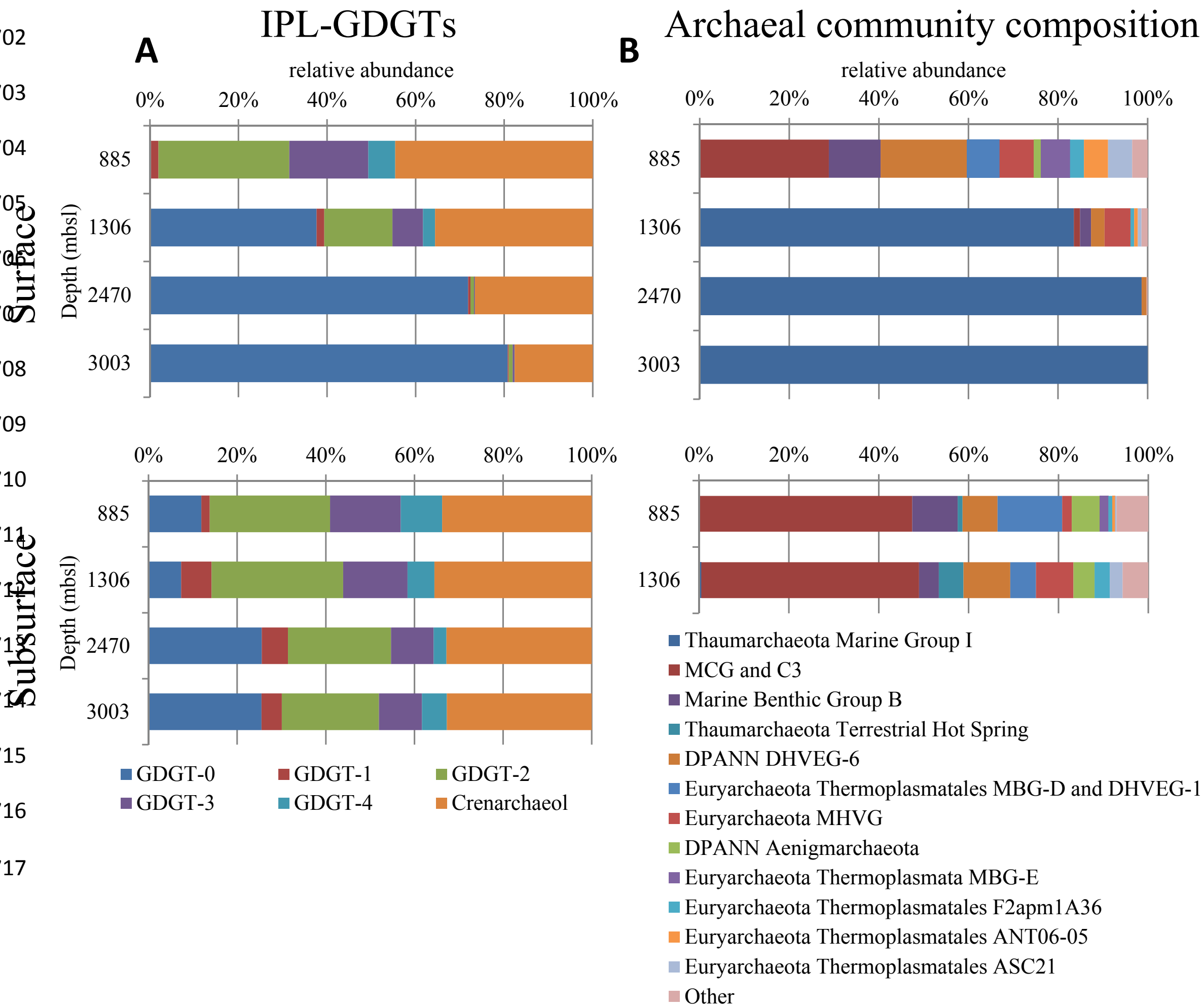
A
IPL-GDGTs
relative abundance
0%
20%
40%
60%
80%
100%
Surface
Depth (mbsl)
885
1306
2470
3003
Subsurface
GDGT-0
GDGT-1
GDGT-2
GDGT-3
GDGT-4
Crenarchaeol
B
Archaeal community composition
Thaumarchaeota Marine Group I
MCG and C3
Marine Benthic Group B
Thaumarchaeota Terrestrial Hot Spring
DPANN DHVEG-6
Euryarchaeota Thermoplasmatales MBG-D and DHVEG-1
Euryarchaeota MHVG
DPANN Aenigmarchaeota
Euryarchaeota Thermoplasmata MBG-E
Euryarchaeota Thermoplasmatales F2apm1A36
Euryarchaeota Thermoplasmatales ANT06-05
Euryarchaeota Thermoplasmatales ASC21
Other



**Fig. 2.**









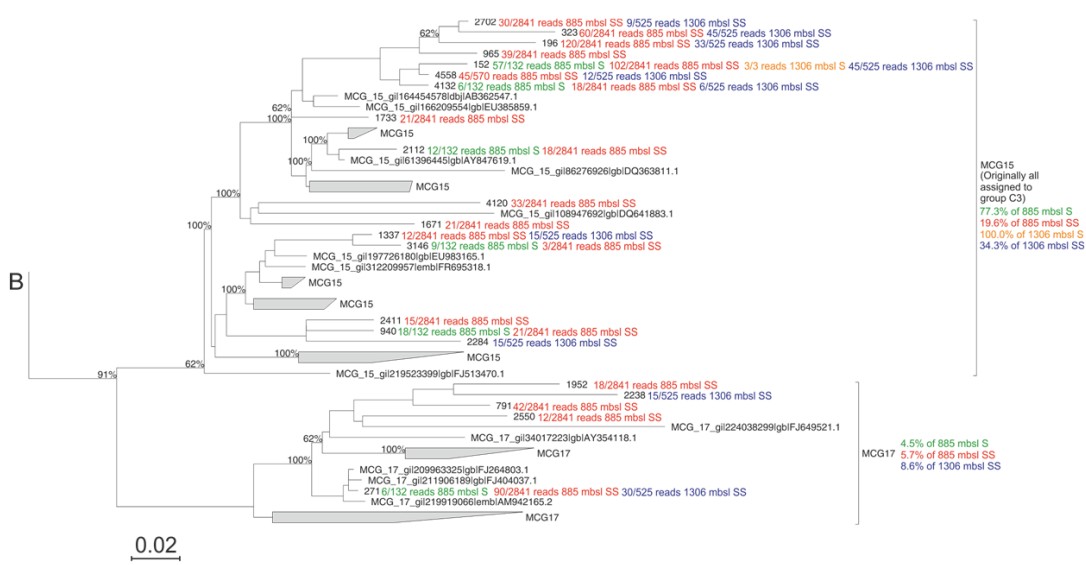




**Fig. 3.**

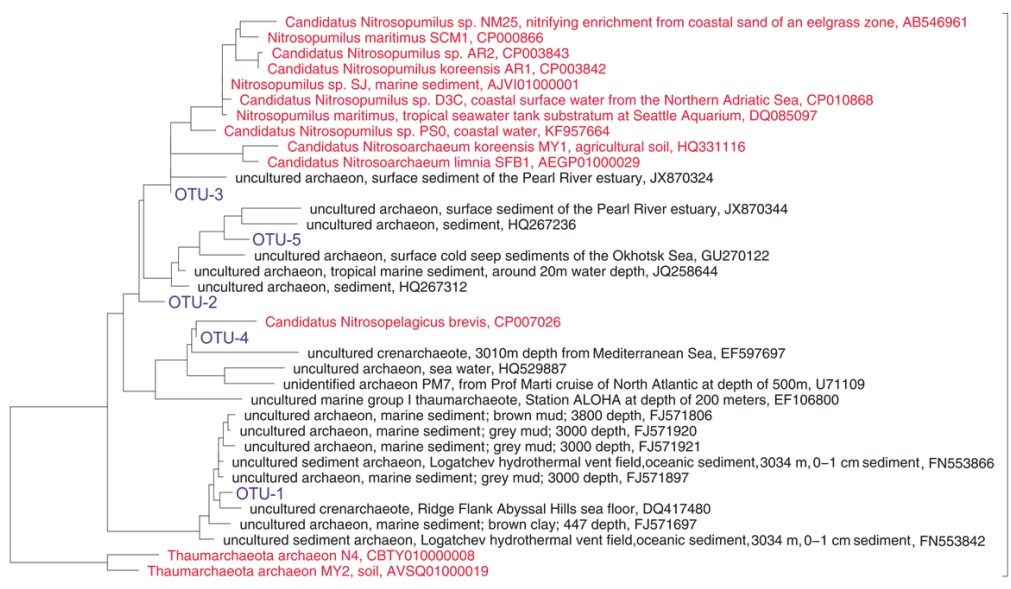







**Fig. 4.**

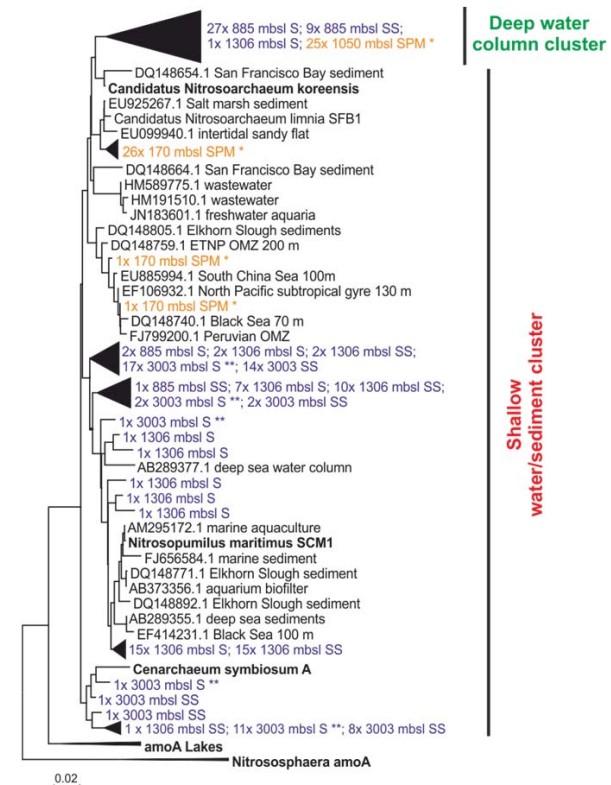



**Fig. 5.**

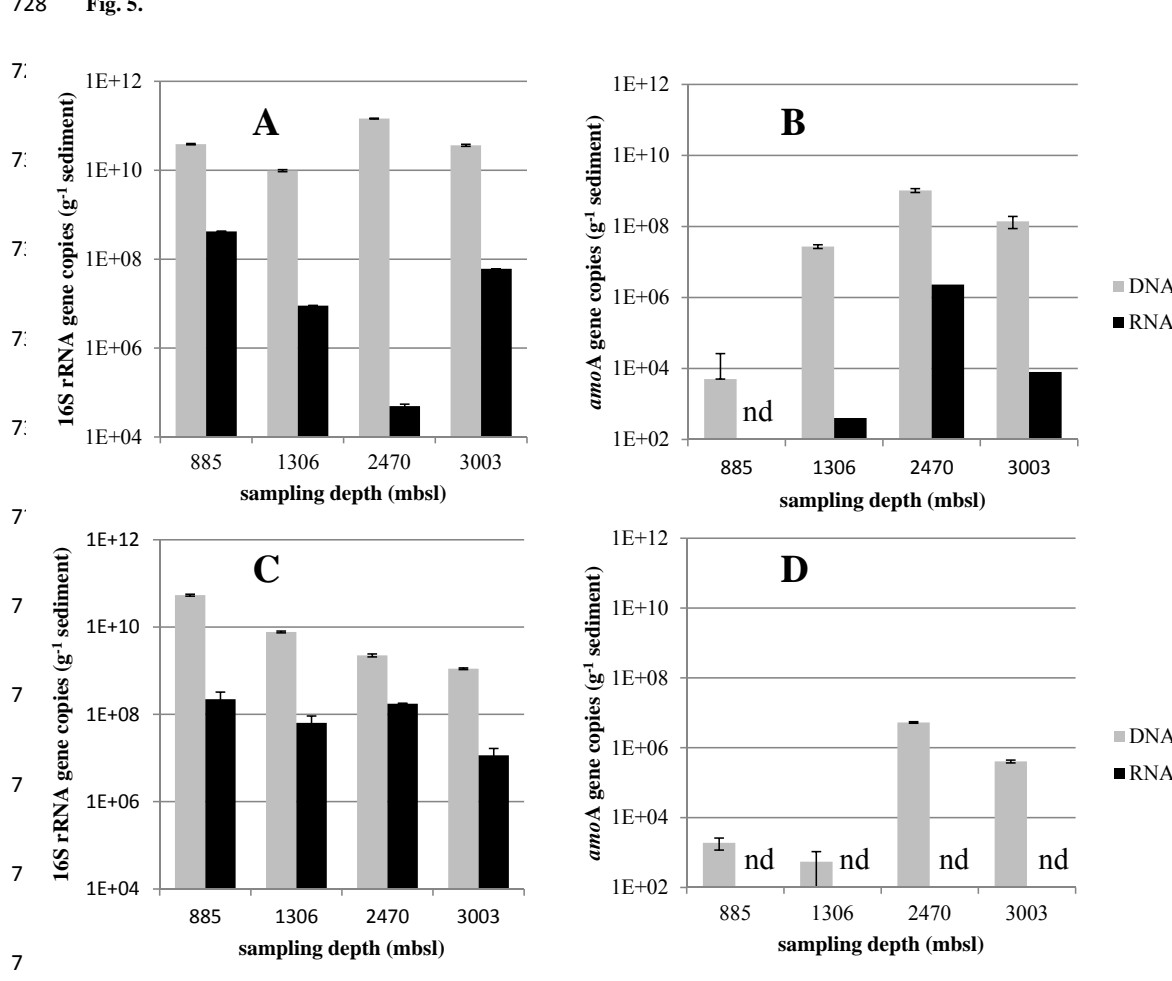