# Peer review of "Benthic Archaea as potential sources of tetraether membrane lipids in sediments across an oxygen minimum zone"

_Biogeosciences, 2017_

## Referee Comment (RC1) · Anonymous Referee #1 · 30 Sep 2017

The manuscript by Besseling et al. describes the distribution of archaeal intact polar tetraether lipids and their covariation with archaeal 16S rRNA genes in the Arabian Sea. Given the ever-expanding diversity of uncultivated archaea discovered by next-gen sequencing, "next gen" lipidomics techniques such as UHPLC-high-resolution mass spectrometry techniques need to be further developed and applied to allow lipid source assignments that in turn will improve the use of intact polar lipid biomarkers and improve our understanding of the mechanisms of, and problems with, proxies such as TEX86. The high-throughput sequencing methods used by the authors are state of the art. However, the methodology for intact polar lipid analysis employed by the authors is sub-par. While I agree that the use of high-resolution mass spectrometry represents

a significant improvement over previous analyses on the same samples, the chromatographic method used by the authors does not satisfy this potential. In fact, I have reason to believe that the results of this study are severely flawed by the choice of chromatographic method. Thus, I cannot recommend publication of this work in Biogeosciences in the current form.

I urge the authors to re-analyze their data using more appropriate reversed phase HPLC-MS methods. Previous studies have shown that diol column HPLC-MS and other normal phase methods lead to severe underestimation of the abundances of glycosidic GDGTs (Wörmer et al., 2013; Zhu et al., 2013). This would explain the unusually high abundances of HPH-GDGT reported in the present study and the lack of other ubiquitous compounds such as hydroxylated and unsaturated GDGT (Liu et al., 2012; Zhu et al., 2014b) and BDGT/PDGT (Becker et al., 2016; Meador et al., 2015; Zhu et al., 2014a). Alternatively, re-analysis of the samples along with authentic standards would help to correct for different ionization efficiencies based on headgroup types. If the authors did perform such a correction (and if they did not do this) they need to acknowledge this in the methods and discussion sections.

Judging from the supplementary spreadsheet, the authors may have considered a wider range of archaeal lipids, not exclusively traditional intact polar tetraethers, than reported in the manuscript. If so, where are these data? However, it is questionable if this comprehensive analysis is possible with the author's choice of chromatography. A more comprehensive analysis of the archaeal lipidome (e.g., by analyzing intact polar archaeols including core lipid structural modifications) in these samples would enhance source assignments and comparison to previous studies (Meador et al., 2015; Yoshinaga et al., 2015), and provide new biomarkers or source assignment for groups such as MCG and Woesearchaeota. This analysis would then satisfy the author's conclusion that the known diversity of archaeal IPLs was significantly expanded.

Further, because the authors used a column different from the one described in their method reference, the suitability of the chromatography for IPL analysis cannot be assessed. The authors should provide annotated TIC or EIC traces for representative samples in supplementary figures. Importantly, the type of chromatography used here does not provide chromatographic separation of IPLs by core lipid type (e.g., GDGT-0, GDGT-1), but the way the authors present their data implies the contrary. If the different core lipid types of each IPL co-eluted in their analyses, the relative abundances reported would be questionable. At the very least the authors would need to do an isotope peak correction and then report in detail how this was done (correction factors for each IPL-core lipid combination). It would be much better to use a method than can actually chromatographically separate IPL-GDGTs by core lipid structure (Zhu et al., 2013). Additionally, the method used for this study likely cannot distinguish between the core lipid crenarchaeol and it's regioisomer. However, crenarchaeol more abundant than the regioisomer in MH GDGT and HPH GDGT but lower than its regioisomer in DH GDGT in thaumarchaeal cultures (Elling et al., 2017). This limitation needs to be addressed either by using a different chromatographic method or at least by discussing this issue in the manuscript.

Given these fundamental limitations, major parts of the methods, results, and discussion sections need to be revised. However, HPLC-MS (re-)analysis with a different method could be achieved within a day or two, given that the current study only includes 8 samples. I hope that the authors will agree that this approach has the potential to greatly expand the significance of their work with only moderate additional effort.

Other comments: Line 27: DeLong et al. report only on Antarctic samples. You may want to choose a reference that discusses a more diverse set of environments (e.g. DeLong and Pace, 2013; Schleper et al., 2005) or a collection of references e.g., (DeLong, 1992; Fuhrman et al., 1992; Teske and Sørensen, 2008).

Line 31: Lloyd et al. only provide data on two archaeal groups. You may rather cite a paper that actually discusses archaeal diversity, such as Teske and Sørensen (2008) or Teske (2013) some of the more recent literature, e.g., Hug et al. (2016) or Spang et al. (2017).

Line 40-42: What are the metabolisms of the other archaeal groups?

Line 59-61: What about the Lincoln et al. (Lincoln et al., 2014) paper?

Line 86: Why was this standard added? Was it used for any correction?

Line 90: Specify the modifications.

Line 159-161: Two DH isomers were also reported earlier by Elling et al. (2014; 2017).

Line 166-171: Did you detect crenarchaeol regioisomer? If not, why not? Co-elution with crenarchaeol? This should be pointed out here or elsewhere.

Line 172-173: The way the % values are used in this section is very confusing (...37.6% of 36.6%) etc. It could help to provide a further table in addition to Table 2 that shows the samples arranged by depth or headgroup type instead of sorting by core GDGT type.

Line 196: What does "p=1.00" represent?

Line 254-260: How were the transcripts analyzed? I did not find this information in the methods section. How long were the samples stored before analysis?

Line 270-274: This sentence is a bit long and complex. Revise?

Line 281: "Differences"?

Line 297: Rather "overlying"?

Line 319-321: Could you point out whether there is any evidence for the occurrence of Woesearchaeota in the water column and whether you would expect this group and the other archaeal groups to be present in oxic vs. anoxic environments (or both).

Line 353-357: I disagree with the statement that the diversity of detected (detectable) IPLs was greatly enhanced by this study. There are two IPL types described in addition to those reported by (Lengger et al., 2012). However, the diversity of IPL-GDGTs reported by previous studies is much higher, e.g. Yoshinaga et al. (2015), with respect to both the headgroup types as well as the structural modifications in the core lipid such as hydroxylation, unsaturation, methylation, monoalkylation, trialkylation, or substitution of glycerol with butanetriol or pentanetriol. Many of these would likely be detectable in the presented samples using different methods.

Line 359-360: GDGT-0 and crenarchaeol are also dominant core lipids of MH-GDGT in Thaumarchaeota (Elling et al., 2014; Elling et al., 2015; Elling et al., 2017; Schouten et al., 2008). MH-GDGT also appears to be the dominant membrane lipid in many Thaumarchaeota. Also, how does your interpretation fit to the results of Lincoln et al. (2014) who suggested production of crenarchaeol by other archaeal groups?

Line 360: Qin et al. did not study IPLs. A more appropriate reference would be Elling et al. (2017).

Line 360-363: I am quite confident that the predominance of HPH-GDGTs in the sediment is an artifact of the chromatographic method that results in severe underestimation of MH-GDGT (and potentially DH-GDGT) relative abundances. However, the notion that higher HPH-content relates to higher activity was also supported by culture studies (Elling et al., 2014).

Line 363: Neither Harvey et al. or Schouten et al. discuss HPH IPLs and do not provide any experimental evidence for degradation rates of HPH versus MH or DH or any other GDGT IPL types. Rephrase.

Line 369-371: Do you think these fossil IPLs extracellular or intracellular (Braun et al., 2016)? Is some of the DNA also fossil? Would the DNA be preserved differently (degradation rates) than the IPLs and how would this affect the interpretation of your results?

Line 372-373: How did the 0/crenarchaeol ratio change for the anoxic sites? If they change in a similar way, does that point to different sources of 0 versus crenarchaeol or to accumulation of fossil IPLs?

Line 380: Typo. "Acid hydrolysis"

Line 384-386: You should discuss here or elsewhere that some cultures exist from the Thermoplasmatales cluster, although most clades remain uncultivated. E.g. there are many (acidophilic, thermoacidophilic) Thermoplasmatales cultures for which lipids have been analysed and all of them produce GDGTs. Further, the recently cultivated Methanomassiliicoccales (closely related to the uncultivated TMEG group) have been shown to produce IPL-GDGTs such as MH-GDGT-0 (Becker et al., 2016). It is thus very likely that the rest of the uncultivated Thermoplasmatales-like archaea can produce GDGTs.

Line 388: The way these papers are referenced is highly misleading. Four of the five references do not relate to Woesearchaeota and the inference of Villanueva et al. (2017) that Woesearchaeota do not produce GDGTs is circumstantial at best without knowledge of the actual GDGT biosynthetic pathway. Rephrase.

Line 392-394: Becker et al. (2016) showed that BDGTs are detectable in a globally distributed set of marine sediments. It is therefore likely that these compounds would be present in your samples. I suspect that you would detect these compounds using different chromatographic conditions. Recently, Thermoplasma-related methanogens (Methanomassiliicoccales) have been identified as a source of BDGTs in the environment (Becker et al., 2016). Meador et al. (2015) also identified further phosphatidic and MH, DH-GDGTs in MCG-rich samples. Could the MCG be sources of these compounds in your samples?

Line 416: As stated earlier, DH-GDGT-0 isomers have been detected in Thaumarchaeota (Elling et al., 2014; Elling et al., 2015).

Line 422: Typo. "sediments"

Line 423: Also Elling et al. (2014).

Line 430: Please provide experimental evidence for the "recalcitrant nature of glycosidic bonds".

Line 433-434: Again, very misleading references, as in Line 388. These works do not discuss the GDGT lipid biosynthetic pathway. Delete the sentence or rephrase.

Line 435: I disagree: Could you please specify how the source assignment of IPL-GDGTs has been improved?

Figure 1: It would be very helpful to have another set of two plots (surface, subsurface) next to A and B that shows summed IPLs by headgroup type or a synthesis of the data in Table 2.

References

Becker K. W., Elling F. J., Yoshinaga M. Y., Söllinger A., Urich T. and Hinrichs K.-U. (2016) Unusual Butane- and Pentanetriol-Based Tetraether Lipids in Methanomassili-icoccus luminyensis, a Representative of the Seventh Order of Methanogens ed. V. Müller. Appl. Environ. Microbiol. 82, 4505–4516. Braun S., Morono Y., Becker K. W., Hinrichs K.-U., Kjeldsen K. U., Jørgensen B. B. and Lomstein B. A. (2016) Cellular content of biomolecules in sub-seafloor microbial communities. Geochim. Cosmochim. Acta 188, 330–351. DeLong E. F. (1992) Archaea in coastal marine environments. Proc. Natl. Acad. Sci. U. S. A. 89, 5685–5689. DeLong E. F. and Pace N. R. (2013) Environmental diversity of Bacteria and Archaea. Syst. Biol. 50, 470–478. Elling F. J., Könneke M., Lipp J. S., Becker K. W., Gagen E. J. and Hinrichs K.-U. (2014) Effects of growth phase on the membrane lipid composition of the thaumarchaeon Nitrosopumilus maritimus and their implications for archaeal lipid distributions in the marine environment. Geochim. Cosmochim. Acta 141, 579–597. Elling F. J., Könneke M., Mußmann M., Greve A. and Hinrichs K.-U. (2015) Influence of temperature, pH, and salinity on membrane lipid composition and TEX86 of marine planktonic thaumarchaeal isolates. Geochim. Cosmochim. Acta 171, 238–255. Elling F. J., Könneke M., Nicol G. W., Stieglmeier M., Bayer B., Spieck E., de la Torre J. R., Becker K. W., Thomm M., Prosser J. I., Herndl G. J., Schleper C. and Hinrichs K.-U. (2017) Chemotaxonomic

characterisation of the thaumarchaeal lipidome. Environ. Microbiol. 19, 2681–2700.
Fuhrman J. A., McCallum K. and Davis A. A. (1992) Novel major archaebacterial group
from marine plankton. Nature 356, 148–149. Hug L. A., Baker B. J., Anantharaman
K., Brown C. T., Probst A. J., Castelle C. J., Butterfield C. N., Hernsdorf A. W., Amano
Y., Kotaro I., Suzuki Y., Dudek N., Relman D. A., Finstad K. M., Amundson R., Thomas
B. C. and Banfield J. F. (2016) A new view of the tree and life's diversity. , Manuscript
submitted for publication. Lengger S. K., Hopmans E. C., Reichart G.-J., Nierop K.
G. J., Sinninghe Damsté J. S. and Schouten S. (2012) Intact polar and core glycerol
dibiphytanyl glycerol tetraether lipids in the Arabian Sea oxygen minimum zone. Part II:
Selective preservation and degradation in sediments and consequences for the TEX86.
Geochim. Cosmochim. Acta 98, 244–258. Lincoln S. A., Wai B., Eppley J. M., Church
M. J., Summons R. E. and DeLong E. F. (2014) Planktonic Euryarchaeota are a signif-
icant source of archaeal tetraether lipids in the ocean. Proc. Natl. Acad. Sci. U. S. A.
111, 9858–9863. Liu X.-L., Lipp J. S., Simpson J. H., Lin Y.-S., Summons R. E. and
Hinrichs K.-U. (2012) Mono- and dihydroxyl glycerol dibiphytanyl glycerol tetraethers
in marine sediments: Identification of both core and intact polar lipid forms. Geochim.
Cosmochim. Acta 89, 102–115. Meador T. B., Bowles M., Lazar C. S., Zhu C., Teske
A. and Hinrichs K.-U. (2015) The archaeal lipidome in estuarine sediment dominated
by members of the Miscellaneous Crenarchaeotal Group. Environ. Microbiol. 17,
2441–2458. Schleper C., Jurgens G. and Jonuscheit M. (2005) Genomic studies of
uncultivated archaea. Nat. Rev. Microbiol. 3, 479–488. Schouten S., Hopmans E. C.,
Baas M., Boumann H., Standfest S., Könneke M., Stahl D. A. and Sinninghe Damsté
J. S. (2008) Intact Membrane Lipids of "Candidatus Nitrosopumilus maritimus," a Cul-
tivated Representative of the Cosmopolitan Mesophilic Group I Crenarchaeota. Appl.
Environ. Microbiol. 74, 2433–2440. Spang A., Caceres E. F. and Ettema T. J. G. (2017)
Genomic exploration of the diversity, ecology, and evolution of the archaeal domain of
life. Science 357. Teske A. (2013) Marine Deep Sediment Microbial Communities. In
The Prokaryotes – Prokaryotic Communities and Ecophysiology (eds. E. Rosenberg,
E. F. DeLong, S. Lory, E. Stackebrandt, and F. Thompson). Springer, Berlin, Heidelberg. pp. 123–138. Teske A. and Sørensen K. B. (2008) Uncultured archaea in deep marine subsurface sediments: have we caught them all? ISME J. 2, 3–18. Wörmer L., Lipp J. S., Schröder J. M. and Hinrichs K.-U. (2013) Application of two new LC-ESI-MS methods for improved detection of intact polar lipids (IPLs) in environmental samples. Org. Geochem. 59, 10–21. Yoshinaga M. Y., Lazar C. S., Elvert M., Lin Y.-S., Zhu C., Heuer V. B., Teske A. and Hinrichs K.-U. (2015) Possible roles of uncultured archaea in carbon cycling in methane-seep sediments. Geochim. Cosmochim. Acta 164, 35–52. Zhu C., Lipp J. S., Wörmer L., Becker K. W., Schröder J. and Hinrichs K.-U. (2013) Comprehensive glycerol ether lipid fingerprints through a novel reversed phase liquid chromatography-mass spectrometry protocol. Org. Geochem. 65, 53–62. Zhu C., Meador T. B., Dummann W. and Hinrichs K.-U. (2014a) Identification of unusual butanetriol dialkyl glycerol tetraether and pentanetriol dialkyl glycerol tetraether lipids in marine sediments. Rapid Commun. Mass Spectrom. 28, 332–338. Zhu C., Yoshinaga M. Y., Peters C. A., Liu X.-L., Elvert M. and Hinrichs K.-U. (2014b) Identification and significance of unsaturated archaeal tetraether lipids in marine sediments. Rapid Commun. Mass Spectrom. 28, 1144–1152.

---

## Author Comment (AC1) · 13 Oct 2017

We thank the reviewer for the constructive comments. We are pleased that that he/she liked the sequencing part of our study. The reviewer raises several concerns regarding our IPL analysis to which we would like to respond below.

1. I urge the authors to re-analyze their data using more appropriate reversed phase HPLC-MS methods. Previous studies have shown that diol column HPLC-MS and other normal phase methods lead to severe underestimation of the abundances of glycosidic GDGTs (Wörmer et al., 2013; Zhu et al., 2013). This would explain the unusually high abundances of HPH-GDGT reported in the present study and the lack

[Figure]

of other ubiquitous compounds such as hydroxylated and unsaturated GDGT (Liu et al., 2012; Zhu et al., 2014b) and BDGT/PDGT (Becker et al., 2016; Meador et al., 2015; Zhu et al., 2014a). Alternatively, re-analysis of the samples along with authentic standards would help to correct for different ionization efficiencies based on headgroup types. If the authors did perform such a correction (and if they did not do this) they need to acknowledge this in the methods and discussion sections

We are familiar with the reversed phase HPLC-MS method published by Wörmer et al., (2013). This is certainly an elegant addition to the lipidomics toolbox, however, we do not agree with the complete dismissal of the LC-MS method originally introduced by Sturt et al. (2014) using a diol column. We have introduced several improvements on the original method (see response to comment below for details), which have considerably improved the performance of the method. The reviewer hints at a systematic bias in the used diol column HPLC-MS method that would lead to an extreme overrepresentation of HPH-GDGTs vs. glycosidic GDGTs. Indeed it is well known that identical core lipids with different head groups can have very different response factors. Additional drivers of response factors are structural features such as length of the core lipid, degree of unsaturation, number of rings or functionalities like hydroxylations. Van Mooy and Fredricks, (2010) already published an estimation of these response differences using the diol column-MS method, based on diacylglycerols with various head groups. Interestingly, they report very similar response factors for diacyls with either a phosphoglycerol or a hexose headgroup, while a diacyl with a dihexose head group has a response factor of one third of the PG and half of the monohexose. While we acknowledge that HPH-GDGTs are likely to have a higher response factor than MH-GDGTs or DH-GDGTs, these differences are unlikely to explain the 40 fold difference in apparent relative abundance we observed in e.g. the surface sample at 3000 m depth. The other IPLs mentioned, i.e. unsaturated GDGT and BDGT/PDGT core lipids with varying head groups, have been reported to be absent in several settings (Becker et al., 2016; Zhu et al., 2014b) or were linked to archaeal producers not detected in our reported archaeal community, and it is, therefore, not a given fact that we should

detect them in the Arabian Sea sediments analysed in our study. Unsaturated GDGT IPLs are not commonly detected; Zhu et al. (2014) reported that unsaturated GDGT IPLs were not detected in sediments in the upwelling region of NW Africa and only in trace amounts in the eastern Mediterranean Sea. The absence of unsaturated GDGT IPLs in our samples is therefore not unsuspected and certainly does not prove that our analysis method is inappropriate as stated by the referee. We screened our extracts for a variety of IPL BDGTs and PDGTs with an extensive list of possible polar head group combinations as indicated in our supplementary Table S1. However none of these IPL compounds were detected. Becker et al., (2016) reported that BDGTs and PDGTs were only detected in Methanomassiliicoccales and not in the other studied archaeal lipidomes. Meador et al., (2015) detected MH-GDGTs in an estuarine sediment sample and linked this to a high relative abundance of MCG. However the notable 13C depletion of the BDGTs reported by Meador et al., (2015) links this compound to relatives of members of Methanomassiliicoccales, members of Methanomassiliicoccales were not detected in our sequencing data. Becker et al., (2015) did not detect BDGTs in samples with relative low concentrations of total organic carbon (TOC), our samples have relative low TOC which can explain the absence of BDGTs. Regarding hydroxylated GDGT-IPLs, we routinely detect these in biomass as well as in environmental samples (data not yet published). We are therefore confident that, would they have been present, we would have detected them. The reviewer suggests to reanalyse the samples against standards to aid in quantification. We have purposefully refrained from quantifying the detected GDGT-IPLs because authentic quantitative standards truly representative of the GDGT-IPLS are not available and quantifying these compounds based on standards that have very different core lipids and much less complex head groups, in our opinion, does not lead to a valid quantification. We do use standards to continuously monitor the performance of the analytical system. We will add text to the materials and methods section, clarifying that the reported abundances are based on peak responses that are not corrected for differences in response factors between various IPL-types.

Finally, re-analysis of the samples at this time would probably lead to biased results due to the storage time of 2-3 years. In our hands, samples that are re-analysed after such a lengthy storage period do show very different ratios for several IPL-GDGTs.

2. Judging from the supplementary spreadsheet, the authors may have considered a wider range of archaeal lipids, not exclusively traditional intact polar tetraethers, than reported in the manuscript. If so, where are these data? However, it is questionable if this comprehensive analysis is possible with the author's choice of chromatography. A more comprehensive analysis of the archaeal lipidome (e.g., by analyzing intact polar archaeols including core lipid structural modifications) in these samples would enhance source assignments and comparison to previous studies (Meador et al., 2015; Yoshinaga et al., 2015), and provide new biomarkers or source assignment for groups such as MCG and Woesearchaeota. This analysis would then satisfy the author's conclusion that the known diversity of archaeal IPLs was significantly expanded.

For our analysis, we now make use of high resolution MS with a mass range of 375-2000 m/z. Our analytical routine involves a data dependent analysis of the 10 most abundant masses in each MS1 spectrum with a dynamic exclusion window of 6 sec (we will add this information to the method section), maximizing our capacity to identify lipids. In addition, our method works with an inclusion list, where we specify masses of compounds that, if detected, will be targeted for a product spectrum. This inclusion list contains the exact mass of most GDGT and archaeol-IPLs known at the time of analysis of this sample set. We have used all this available data to search for the compounds in the inclusion list and for additional unknown IPLs with a GDGT or archaeol core. We have successfully detected additional GDGTs as well as archaeol-based IPLs in other environmental sample sets and archaeal biomass (data not yet published). We are of the opinion that there is also sometimes information in the absence of a certain set of compounds and therefore we reported our inclusion list to indicate the breadth of our search. In contrast to what the referee suggests, we do not conclude that we are expanding the known diversity of archaeal IPLs. We wrote that we expanded the

screening for IPLs in our lab based on the use of UHPLC chromatography and high resolution mass spectrometry (quadrupole-orbitrap hybrid). We will adjust the manuscript to make this point more clear.

3. Further, because the authors used a column different from the one described in their method reference, the suitability of the chromatography for IPL analysis cannot be assessed. The authors should provide annotated TIC or EIC traces for representative samples in supplementary figures. Importantly, the type of chromatography used here does not provide chromatographic separation of IPLs by core lipid type (e.g., GDGT-0, GDGT-1), but the way the authors present their data implies the contrary. If the different core lipid types of each IPL co-eluted in their analyses, the relative abundances reported would be questionable. At the very least the authors would need to do an isotope peak correction and then report in detail how this was done (correction factors for each IPL-core lipid combination). It would be much better to use a method than can actually chromatographically separate IPL-GDGTs by core lipid structure (Zhu et al., 2013). Additionally, the method used for this study likely cannot distinguish between the core lipid crenarchaeol and it's regioisomer. However, crenarchaeol more abundant than the regioisomer in MH GDGT and HPH GDGT but lower than its regioisomer in DH GDGT in thaumarchaeal cultures (Elling et al., 2017). This limitation needs to be addressed either by using a different chromatographic method or at least by discussing this issue in the manuscript

Indeed, the column used for the work described here is a different column from the original method. It is a UHPLC silica based diol column used with the same solvent system as originally described by Sturt et al., (2014) and the overall chromatographic patterns have thus not changed but the resolution of the chromatography has significantly improved. We will add information on the improved chromatography to the supplemental information. Due to the improved separation power of the column, we can separate GDGT-IPLs with different cores but the same head group. The suggested Isotopic corrections are therefore not necessary. The reviewer is correct that we cannot

separate crenarchaeol from crenarchaeol regio-isomer with identical head groups. We are aware that Zhu et al. (2013) reported a separation between IPL crenarchaeol and crenarchaeol regio-isomer with the same head group. With our method this is not feasible, we will add to the material and method section that IPL crenarchaeol is the sum of crenarchaeol and crenarchaeol regio-isomer. Lengger et al., (2012) reported core lipid content in these samples and did not find dominance of crenarchaeol regio-isomer over crenarchaeol. We therefore do not think that the culture study the reviewer refers to is relevant. We will add TIC and EIC traces to the supplementary data to clarify our peak separation and identification.

4. Given these fundamental limitations, major parts of the methods, results, and discussion sections need to be revised. However, HPLC-MS (re-)analysis with a different method could be achieved within a day or two, given that the current study only includes 8 samples. I hope that the authors will agree that this approach has the potential to greatly expand the significance of their work with only moderate additional effort.

As discussed above, we do not agree with the dismissal of the diol column based HPLC-MS method. As explained above, we feel that reanalysis of the samples at this time with the reversed phase method would not lead to valid results due to the age of the extracts. We respectfully disagree with the reviewer that our results, and therefore our interpretation and discussion, are invalid.

5. Other comments: Line 27: DeLong et al. report only on Antarctic samples. You may want to choose a reference that discusses a more diverse set of environments (e.g. DeLong and Pace, 2013; Schleper et al., 2005) or a collection of references e.g., (DeLong, 1992; Fuhrman et al., 1992; Teske and Sørensen, 2008).

The addition of literature would benefit the introduction, we will modify this.

6. Line 31: Lloyd et al. only provide data on two archaeal groups. You may rather cite a paper that actually discusses archaeal diversity, such as Teske and Sørensen (2008) or Teske (2013) some of the more recent literature, e.g., Hug et al. (2016) or Spang et

al. (2017).

We will modify the manuscript and add some of the proposed literature references. However, we feel that it is also important to cite original references where pioneering efforts have been described.

7. Line 40-42: What are the metabolisms of the other archaeal groups?

We mention the metabolism of detected groups in our discussion, however for most of the detected groups metabolism is unknown as neither cultures nor metagenomes are available to predict their metabolic capabilities. For the rest i.e. MGD-B and E, potential metabolism is derived from (meta)genome information as shown already in lines 314-315.

8. Line 59-61: What about the Lincoln et al. (Lincoln et al., 2014) paper?

Lincoln et al. (2014) indeed reported the marine group II as potential producers of crenarchaeol. However there is still an ongoing discussion on this hypothesis (Lincoln et al., 2014b; Schouten et al., 2014). We will add this literature and discussion into our introduction (lines 59-61).

9. Line 86: Why was this standard added? Was it used for any correction?

This standard (PAF; 1-O-hexadecyl-2-acetoyl-sn-glycero-3-phosphocholine) was added to continuously monitor the performance of the MS response and to correct for matrix effects. We also corrected peak areas accordingly. We will clarify this in the materials and methods section.

10. Line 90: Specify the modifications.

The modifications are already described in detail in the text after the referral to the original Sturt et al. (2004) method, lines 90-110. We will add more details as mentioned above.

11. Line 159-161: Two DH isomers were also reported earlier by Elling et al. (2014;

2017).

Indeed, Elling et al. (2014 reported an "early eluting " DH-GDGT and Elling et al. (2017) reports early eluting isomers of DH- and DH-OH-GDGTs. We will change our text to acknowledge these reports. However, although Elling et al. (2017) speculates on the nature of the isomerisation, no spectral details are given. We show, for the first time, spectral data on these isomers and based on these spectra, interpret the nature of the isomerization. We therefore feel it is worthwhile to include this information in the supplementary information.

12. Line 166-171: Did you detect crenarchaeol regioisomer? If not, why not? Co-elution with crenarchaeol? This should be pointed out here or elsewhere.

As stated above, the crenarchaeol regio-isomer IPLs co-elute with crenarchaeol with the same head group. We will clarify this in the manuscript.

13. Line 172-173: The way the % values are used in this section is very confusing (. . .37.6% of 36.6%) etc. It could help to provide a further table in addition to Table 2 that shows the samples arranged by depth or headgroup type instead of sorting by core GDGT type.

We will clarify this in the manuscript and will consider how to alter our tables to make our data more clear.

14. Line 196: What does "p=1.00" represent?

The correlation between our IPL derived CL-GDGTs and the ones reported by Lengger et al., (2012). We will clarify this in the manuscript and note this differently.

15. Line 254-260: How were the transcripts analyzed? I did not find this information in the methods section. How long were the samples stored before analysis?

The transcripts were analysed as described in Pitcher et al., (2011). We refer to this in our material and method section. As mentioned, the sediments were sliced on board

and directly stored at -80C. They were kept at -80C until lipid and DNA/RNA extractions. After the extractions the DNA/RNA and the cDNA were stored at -80C until analyses.

16. Line 270-274: This sentence is a bit long and complex. Revise?

The referee is correct, we will modify this sentence in our manuscript.

17. Line 281: "Differences"?

The referee is correct, it should be differences instead of difference.

18. Line 297: Rather "overlying"?

The referee is correct, it should be overlying.

19. Line 319-321: Could you point out whether there is any evidence for the occurrence of Woesearchaeota in the water column and whether you would expect this group and the other archaeal groups to be present in oxic vs. anoxic environments (or both).

There is evidence for Woesearchaeota in marine pelagic oxygenated environments, most recently by (Liu et al., 2017). The Arabian Sea contains an oxic environment and an oxygen minimum zone, so far it is uncertain if the Woesearchaeota could reside in the anoxic pelagic zone.

20. Line 353-357: I disagree with the statement that the diversity of detected (detectable) IPLs was greatly enhanced by this study. There are two IPL types described in addition to those reported by (Lengger et al., 2012). However, the diversity of IPL-GDGTs reported by previous studies is much higher, e.g. Yoshinaga et al. (2015), with respect to both the headgroup types as well as the structural modifications in the core lipid such as hydroxylation, unsaturation, methylation, monoalkylation, trialkylation, or substitution of glycerol with butanetriol or pentanetriol. Many of these would likely be detectable in the presented samples using different methods.

In contrast to what the referee suggests, we do not conclude that we are expanding the known diversity of archaeal IPLs (see remark on point 2). However we can make this

more clear in the manuscript. We will rewrite the sentences.

21. Line 359-360: GDGT-0 and crenarchaeol are also dominant core lipids of MH-GDGT in Thaumarchaeota (Elling et al., 2014; Elling et al., 2015; Elling et al., 2017; Schouten et al., 2008). MH-GDGT also appears to be the dominant membrane lipid in many Thaumarchaeota. Also, how does your interpretation fit to the results of Lincoln et al. (2014) who suggested production of crenarchaeol by other archaeal groups?

We also see a high relative abundance of IPL GDGT-0 and IPL crenarchaeol in the surface sediment samples coinciding with a high relative abundance of thaumarchaeota. Lincoln et al. (2014) suggested that marine group II (MGII) may be significant contributors of crenarchaeol in oceanic surface waters based on the combination of core lipid (CL)-GDGT and metagenomic analyses. However, the lack of culture representatives of this group and the difference in the resilience times between CL-GDGTs (attributed to dead biomass) and DNA weakens the arguments of MGII as important GDGTs producers (Lincoln et al., 2014b; Schouten et al., 2014). Members of the MGII are, so far, only found in the pelagic and not in the benthic environment therefore we did not include this in our manuscript. The archaeal groups that we detected are, currently, lacking cultured representatives and therefore their membrane lipid composition is still uncertain.

References: Becker, K. W., Elling, F. J., Yoshinaga, M. Y., Söllinger, A., Urich, T. and Hinrichs, K. U.: Unusual butane- and pentanetriol-based tetraether lipids in Methanomassiliicoccus Luminyensis, a representative of the seventh order of methanogens, Appl. Environ. Microbiol., 82(15), 4505–4516, doi:10.1128/AEM.00772-16, 2016. Elling, F. J., Konneke, M., Lipp, J. S., Becker, K. W., Gagen, E. J. and Hinrichs, K. U.: Effects of growth phase on the membrane lipid composition of the thaumarchaeon Nitrosopumilus maritimus and their implications for archaeal lipid distributions in the marine environment, Geochim. Cosmochim. Acta, 141, 579–597, doi:10.1016/j.gca.2014.07.005, 2014. Elling, F. J., Könneke, M., Nicol, G. W., Stieglmeier, M., Bayer, B., Spieck, E., de la Torre, J. R., Becker, K. W., Thomm,

[Figure]

M., Prosser, J. I., Herndl, G. J., Schleper, C. and Hinrichs, K. U.: Chemotaxonomic characterisation of the thaumarchaeal lipidome, Environ. Microbiol., 19(7), 2681–2700, doi:10.1111/1462-2920.13759, 2017. Lengger, S. K., Hopmans, E. C., Reichart, G.-J., Nierop, K. G. J., Sinninghe Damsté, J. S. and Schouten, S.: Intact polar and core glycerol dibiphytanyl glycerol tetraether lipids in the Arabian Sea oxygen minimum zone. Part II: Selective preservation and degradation in sediments and consequences for the TEX86, Geochim. Cosmochim. Acta, 98, 244–258, doi:10.1016/j.gca.2012.05.003, 2012. Lincoln, S. a, Wai, B., Eppley, J. M., Church, M. J., Summons, R. E. and DeLong, E. F.: Planktonic Euryarchaeota are a significant source of archaeal tetraether lipids in the ocean., Proc. Natl. Acad. Sci. U. S. A., 111(27), 9858–63, doi:10.1073/pnas.1409439111, 2014a. Lincoln, S. a, Wai, B., Eppley, J. M., Curch, M. J., Summons, R. E. and Delong, E. F.: Reply to Schouten et al .: Marine Group II planktonic Euryarchaeota are significant contributors to tetraether lipids in the ocean, Proc. Natl. Acad. Sci., 111(41), 4286, doi:10.1073/pnas.1416736111, 2014b. Liu, H., Zhang, C. L., Yang, C., Chen, S., Cao, Z., Zhang, Z. and Tian, J.: Marine Group II dominates planktonic archaea in water column of the Northeastern South China Sea, Front. Microbiol., 8(JUN), 1–11, doi:10.3389/fmicb.2017.01098, 2017. Meador, T. B., Bowles, M., Lazar, C. S., Zhu, C., Teske, A. and Hinrichs, K.-U.: The archaeal lipidome in estuarine sediment dominated by members of the Miscellaneous Crenarchaeotal Group, Environ. Microbiol., 17(7), 2441–2458, doi:10.1111/1462-2920.12716, 2015. Van Mooy, B. A. S. and Fredricks, H. F.: Bacterial and eukaryotic intact polar lipids in the eastern subtropical South Pacific: Water-column distribution, planktonic sources, and fatty acid composition, Geochim. Cosmochim. Acta, 74(22), 6499–6516, doi:10.1016/j.gca.2010.08.026, 2010. Pitcher, A., Villanueva, L., Hopmans, E. C., Schouten, S., Reichart, G.-J. and Sinninghe Damsté, J. S.: Niche segregation of ammonia-oxidizing archaea and anammox bacteria in the Arabian Sea oxygen minimum zone., ISME J., 5(12), 1896–904, doi:10.1038/ismej.2011.60, 2011. Schouten, S., Villanueva, L., Hopmans, E. C., van der Meer, M. T. J. and Sinninghe Damsté, J. S.: Are Marine Group II Euryarchaeota significant contributors to tetraether lipids in the ocean?, Proc. Natl. Acad. Sci. U. S. A., 111(41), E4285, doi:10.1073/pnas.1416176111, 2014. Sturt, H. F., Summons, R. E., Smith, K., Elvert, M. and Hinrichs, K.-U.: Intact polar membrane lipids in prokaryotes and sediments deciphered by high-performance liquid chromatography/electrospray ionization multistage mass spectrometry—new biomarkers for biogeochemistry and microbial ecology, Rapid Commun. Mass Spectrom., 18(6), 617–628, doi:10.1002/rcm.1378, 2004. Wörmer, L., Lipp, J. S., Schröder, J. M. and Hinrichs, K. U.: Application of two new LC-ESI-MS methods for improved detection of intact polar lipids (IPLs) in environmental samples, Org. Geochem., 59, 10–21, doi:10.1016/j.orggeochem.2013.03.004, 2013. Zhu, C., Lipp, J. S., Wörmer, L., Becker, K. W., Schröder, J. and Hinrichs, K. U.: Comprehensive glycerol ether lipid fingerprints through a novel reversed phase liquid chromatography-mass spectrometry protocol, Org. Geochem., 65, 53–62, doi:10.1016/j.orggeochem.2013.09.012, 2013. Zhu, C., Yoshinaga, M. Y., Peters, C. A., Liu, X. L., Elvert, M. and Hinrichs, K. U.: Identification and significance of unsaturated archaeal tetraether lipids in marine sediments, Rapid Commun. Mass Spectrom., 28(10), 1144–1152, doi:10.1002/rcm.6887, 2014a. Zhu, C., Meador, T. B., Dummann, W. and Hinrichs, K. U.: Identification of unusual butanetriol dialkyl glycerol tetraether and pentanetriol dialkyl glycerol tetraether lipids in marine sediments, Rapid Commun. Mass Spectrom., 28(November 2013), 332–338, doi:10.1002/rcm.6792, 2014b.

---

## Referee Comment (RC2) · Anonymous Referee #1 · 30 Nov 2017

I agree with many of the modifications proposed by the authors. Most importantly, since they use an unpublished method, they will need to demonstrate in detail the separation of the different IPL-GDGT cyclized GDGT-headgroup combinations, and exclude, e.g. the co-elution of unsaturated GDGTs with regular GDGTs.

I appreciate that determining response factors for GDGTs is very challenging due to the lack of commercial standards but I think that this would have been perfectly feasible for the small number of compounds the authors have analyzed. I urge the authors to clearly state that they have not used response factors and how this affects interpretation of the results. It is not acceptable to cite the response factors from Van Mooy

& Fredricks (2010) since they were measured on a different chromatographic method and different instruments and on compounds that are not comparable to GDGT-IPLs. GDGT-IPLs are twice the size of the standards measured in that paper and thus the relative ionization efficiencies will be dramatically different (this will also change the relative effect of one versus two glycosidic moieties). Further, ionization of GDGT-IPLs will heavily depend on elution time, since the method chosen by the authors uses a strong polarity gradient. This will strongly affect the efficiency of ionization of GDGTs relative to their retention time and will likely lead to a systematic underestimation of more apolar GDGT-IPLs (e.g. MH) that elute earlier when the mobile phase consists mostly of hexane (which is not beneficial for ionization).

Lastly, the authors should to assess why the fractional abundances of IPLs are so different between the current study and their earlier study on the same samples (Lengger et al., 2012)

References Lengger, S.K., Hopmans, E.C., Reichart, G.-J., Nierop, K.G.J., Sinninghe Damsté, J.S., and Schouten, S. (2012) Intact polar and core glycerol dibiphytanyl glycerol tetraether lipids in the Arabian Sea oxygen minimum zone. Part II: Selective preservation and degradation in sediments and consequences for the TEX86. Geochim. Cosmochim. Acta 98: 244–258. Van Mooy, B.A.S. and Fredricks, H.F. (2010) Bacterial and eukaryotic intact polar lipids in the eastern subtropical South Pacific: Water-column distribution, planktonic sources, and fatty acid composition. Geochim. Cosmochim. Acta 74: 6499–6516.

---

## Referee Comment (RC3) · J.nbsp;S. Lipp (Referee) · 11 Dec 2017

The manuscript by Besseling et al. describes a combination of molecular biological gene-based techniques with UHPLC/MS analysis of intact polar lipids to find potential archaeal sources of tetraether membrane lipids in sediments recovered from an oxygen minimum zone in the Arabian Sea. The authors compare archaeal lipid diversity and abundances with corresponding data gathered from sequencing of 16S-DNA/RNA and amoA functional genes and qPCR analysis. The results confirm previous observations of an active thaumarchaeotal community producing high quantities of HPH-GDGTs. Further, the authors speculate about the significance of members of the MCG

as sources for an intact polar lipid with a yet unknown polar head group. I received my review assignment after reviewer 1 had already posted his/her comments. As you can see below, I raise similar issues and support most of the reviewer's concerns.

The study is part of a series of manuscripts from the same group of authors where identical Arabian Sea samples have been analyzed. Naturally, the previous results are a point of reference. While the overall study design is state-of-the-art and the topic is of interest for the readers of Biogeosciences, I have some concerns regarding the applied analytical protocol for IPLs and the presentation of the quantitative data. Further, there are significant discrepancies with previous data gathered from the same samples. Finally, the reference list is not up to date - there are several studies concerning the thaumarchaeal lipidome which are relevant.

1) The method has not been described before (with this choice of column) and it is not completely clear how the new results compare to previous installments. Most importantly, it is not clear which compounds are captured in the analytical window as according to the authors some important GDGT-0-based lipid which can be found during APCI analysis of hydrolysate cannot be found during IPL analysis. Also, the comparison of relative proportions of lipids between samples in the current way is misleading. The ionization behavior of lipids varies strongly according to their structure and accordingly using the sum of peak areas for "total lipid abundance" and relative abundance calculated from this value is problematic. Since the authors have not used standards (and do not plan to) for correction of response factors, its perhaps best to compare absolute peak areas and report them in a table. The methodological shortcomings need to be addressed in a revised version (see also comments below).

2) The data only compare poorly to previous results reported in Lengger et al. 2014 where absolute quantities and relative proportions of MH, DH, and HPH-GDGTs were reported. In this study, the hydrolysis method with subsequent APCI quantification was used and problematic response factors are avoided. However, the numbers just do not add up (see detailed comments). I am very surprised that this study is not included in

the discussion at all. This also needs to be addressed thoroughly in a revised version.

Best regards, Julius Lipp

Comment to author's reply to response factors raised by reviewer 1: The study by Van Mooy and Fredricks (2010, GCA) explicitly states that "...these RFs are not applicable for use in any future quantitative analyses of IP-DAGs using a ThermoFinnigan LCQ Deca XP ion-trap mass spectrometer or any other mass spectrometer...". I urge that no conclusions should be drawn from these data. It is also speculative that a 40-fold different is unlikely, as the two ion sources are different (heated vs unheated ESI) and the technology of the two mass spectrometers is also different (ion trap vs orbitrap). We probably all can agree that quantification with standards analyzed on the same machine as the environmental samples would be more appropriate here. I am more worried that the quantitative data here does not match previous data from the same samples (Lengger et al., 2014, see comments below). This hints to some methodological bias and supports suspicions from reviewer 1. These issues should be discussed.

Detailed comments:

Line 70: typo "repertoire ".

Line 86: was the PAF standard used somehow for quantification?

Lines 93-98: this is a novel LCMS method that has not been published before. As reviewer 1 noted more information would be desirable in this case (the best way would be to report the new method in another peer-reviewed manuscript). It seems that the individual GDGTs with rings can be separated (judging from the supplemental figure), here some more mass chromatograms as supplemental figures would indeed be good for illustration. What other compounds can be separated, are the BDGTs/other methylated GDGTs, unsaturated GDGTs, hydroxylated GDGTs in the analytical window? Especially OMZ sediments should have abundant unsaturated GDGTs (Zhu et al., 2014, RCM). How do quantitative results compare to other published methods? Is

the method suitable to comprehensively capture the archaeal lipidome?

Line 101: the unit of resolution is not ppm. I also suggest to use "resolving power" instead of "resolution" as it is a better term.

Line 166: according to Tab 2 it is 44.7% - I am sure this is due to rounding. However it would be great to have consistent numbers in the table and the text. Other values are also different in text and table, please check.

Lines 156-196: the "relative abundance" is given as proportion of peak area. This should at least be reported and the shortcomings of not using authentic standards need to be discussed thoroughly (see also major comments and line 381-397). Perhaps add absolute peak area values to the table?

Lines 194-196: Consistency with Lengger et al. 2012 study: please show the data of the comparison (table?). The Greek letter rho should be used for Pearson correlation coefficients - I assume that is what has been calculated here. Please add information on what is compared and how it is calculated. The letter "p" is usually used for statistical significance, p=1 would be really bad.

Line 360: add Elling et al. 2014, 2015, 2017 and Schouten et al. 2008 references for more complete IPL inventory of Thaumarchaeota.

Line 362-363 and 367-369: the cited studies have not looked at stability of HPH-GDGTs. DH-GDGT stability also has not been experimentally assessed by the cited Lengger et al. 2012 and 2014 studies. Please discuss the stability of phosho vs gly-colipids (and possibly ester vs ether lipids as no study has compared purely phospho vs glyco ETHER lipids, cf. Logemann et al. 2010) in a more balanced way and refrain from speculation without evidence; (additional) useful references in this context are Lipp and Hinrichs 2009, Logemann et al. 2010, Schouten et al. 2010, Xie et al 2013.

Line 371: What about the possibility of other archaeal sources for crenarchaeol, e.g. Lincoln et al. 2014?

Line 381-397: regarding the unknown IPL type for GDGT-0, what other compounds might have been missed? It seems like potentially a major proportion! There is a lot of speculation regarding the source of an undetected GDGT-0, but how sure can you be that there is no methodological problem with the method, especially as it seems to be used for the first time? Lengger et al. 2014 have done semi-preparative IPL separation into head group classes and found abundant GDGT-0 connected to MH, DH and HPH headgroups (Fig. 5 and Table A3 in the supplemental material, here station P900 0-2 cm). Comparison of these values and Tab. 2 for the 885 mbsl surface sample shows major differences and multiple values do not match: e.g. DH-GDGT-0 and DH-GDGT-cren are 63.7 and (144+86.5=230.5; incl. regioisomer) ng/g sed, respectively, a ratio of ~1:3. According to Tab 2 in the current manuscript the corresponding values are "ND" and 43.1 %. Why is the DH-GDGT-0 not detected here, assuming a ratio of 1:3 it should have roughly 10% contribution? Another example is found in Tab 3, it seems that the 885 mbsl surface sample is dominated by 98.1% of DH lipids. However, Lengger et al. 2014 report much higher MH abundances than DH (roughly 800 ng/g vs 400 ng/g). What is the authors' explanation for this major difference? Please discuss all data in comparison to Lengger et al. 2014. Is the diol column method not sensitive enough to capture what had been seen with the prep-hydrolysis-APCI method before? Are some compounds not detected with similar efficiency as suspected by reviewer 1? Can the discrepancies be due to that what the authors report as "relative abundance" is in fact the relative proportion of total peak area for compounds which are known to behave dramatically different during ionization and cannot be simply summed together? A meaningful comparison is only possible with standards and correction of response factors. Again, this comparison is important and needs to be discussed, especially before invoking unidentified and undetected IPL headgroup types for GDGT-0 which are somehow not in the analytical window of a new analytical method which has not been previously published.

Line 384: is the reference Lengger et al. 2014 correct? Or do the authors want to cite the 2012 study? As explained above, the 2014 manuscript is probably as important as

the 2012 study.

Line 403: A known source are the Sulfolobales. Add to discussion, see also line 414-417 comment.

Line 409: is there evidence for selective preservation of water column GDGTs at this station? This would require a dramatic degradation of GDGT-0 which is abundant in SPM within the OMZ (Line 404). It seems more likely that the lipids are produced within the sediments. Rearrange.

Line 410-412: the DH isomers have been reported previously in Elling et al. 2014 and 2017. Elling et al. 2017 studied the lipidome of several thaumarchaeal cultures and found isomers not only for DH-GDGTs but also for hydroxylated DH-GDGTs. Also, they found these structures not only for the acyclic structure (GDGT-0) but also for core lipids with more rings. Please add references to prior studies and discuss why only GDGT-0 has been found in this study.

Line 413-414: which moieties? And what cannot be compared? Please clarify.

Line 414-417: the GDGT-0 with a cyclopentanetetraol (formerly called GDNT) has not only been found in pure cultures of Sulfolobales but was also detected in sediments (Sturt et al., 2014; Lipp and Hinrichs, 2009). The sources in the present study are likely the same as in these two studies. I assume the authors mean "microbial sources" and not "sources" (Line 416). Please rephrase. Have Sulfolobales been found in the molecular biological data?

Lines 418-437: I do not think there should be references in the conclusions section. If you want to have references here, add all relevant ones (see below and reviewer 1 comments).

Line 420: what does "specially" mean here? I would rephrase to ". . .we have unraveled the high diversity of benthic archaea harbored in anoxic sediments of the Arabian Sea, as well as. . .". Remove "specially" and add "Arabian Sea".

Line 421: "increasing the repertoire of archaeal intact polar lipids detected" sounds as if many new archaeal IPLs were found. However, all the described lipids (and many many more that have been found in similar environmental samples) have already been described in the literature (see also reviewer 1 comments). Please rephrase or remove statement.

Line 424: add Elling et al. 2014 reference.

Line 426: change "important" to "abundant".

Line 429: rephrase to "...which could either be attributed to a fossil signal... or being IPLs synthesized..." (add "either").

Line 431: why are these GDGT-0 derivatives "unusual"? They have been described in the literature before (e.g. Elling et al. 2014, 2017 for DH-GDGT isomers, HCP (then labelled as GDNT) in Sturt et al. 2004 and Lipp and Hinrichs, 2009).

Line 437: this implies the authors have only assumed a very low diversity before they analyzed the samples. Why is that? It is known that there is a large variety of IPLs in the environment (see all the details that reviewer 1 has posted in his/her comments), what was the rationale for a low diversity in these samples? I suggest removing this statement as it is not relevant what the authors have assumed.

Table 2/3: use consistent names "subsurface/deep".

Fig 1: add "10-12 cm" to "subsurface", also in tables and other figures.

―――――――――――――――――――――

---

## Author Comment (AC2) · 10 Jan 2018

We thank the reviewer again for the additional comments. Here are our replies.

1) I agree with many of the modifications proposed by the authors. Most importantly, since they use an unpublished method, they will need to demonstrate in detail the separation of the different IPL-GDGT cyclized GDGT-headgroup combinations, and exclude, e.g. the co-elution of unsaturated GDGTs with regular GDGTs.

As stated in our previous reply to comments regarding the use of a modifications to the original normal phase IPL method, we do not agree with the reviewer that this represents an "unpublished method". The basis of the method is unchanged and thus we do not feel the functioning of the method deserves a lengthy discussion in a paper that focusses on different issues. As indicated before, we will illustrate the improvements made in the method by adding illustrative figures in the supplemental information. We cannot fully exclude the co-elution of unsaturated GDGTs-IPLs with regular GDGT-IPLs. Per suggestion of the reviewer we re-analyzed the anoxic sediment samples (surface at 885 mbsl, and subsurface of all the stations) using a reversed phase LC/MS method to estimate the contribution of unsaturated core GDGTs to the GDGT-CL pool. As expected after a couple of years of storage the IPLs were partially decayed and could therefore not be reinterpreted. We did observe a trace of potentially unsaturated CL-GDGTs (e.g. GDGT-0:1) but these represented <0.25% of the total CL pool.

2) I appreciate that determining response factors for GDGTs is very challenging due to the lack of commercial standards but I think that this would have been perfectly feasible for the small number of compounds the authors have analyzed. I urge the authors to clearly state that they have not used response factors and how this affects interpretation of the results. It is not acceptable to cite the response factors from Van Mooy & Fredricks (2010) since they were measured on a different chromatographic method and different instruments and on compounds that are not comparable to GDGT-IPLs. GDGT-IPLs are twice the size of the standards measured in that paper and thus the relative ionization efficiencies will be dramatically different (this will also change the relative effect of one versus two glycosidic moieties). Further, ionization of GDGT-IPLs will heavily depend on elution time, since the method chosen by the authors uses a strong polarity gradient. This will strongly affect the efficiency of ionization of GDGTs relative to their retention time and will likely lead to a systematic underestimation of more apolar GDGT-IPLs (e.g. MH) that elute earlier when the mobile phase consists mostly of hexane (which is not beneficial for ionization).

We do not agree with the referee that isolation of standards to the purity required for use as standards (i.e. >95%) to determine response factors is "perfectly feasible". In

fact, it is a large project in itself that requires a large amount of source material (hard to culture archaea), many man months of experimental work and large amounts of solvents to do the purification as well as proofing of the material by NMR and mass spectrometry. Given the fact that isolated IPLs are also unstable, and the fact that absolute quantitation does not contribute to the discussion (as we focus on relative distributions in comparisons with the genetic fingerprint of the sediments), we do not feel the effort was or is justified. We have already indicated in our response to the reviewer's earlier comments that we will clarify in our materials and methods that we do not use response factors to correct peak areas. We therefore also do not plan to cite the Van Mooy & Fredricks (2010), or apply their published response factors, but merely used this work to illustrate what the impact of the difference in response factors might be.

We have assessed the difference between the HPLC-MS method using a diol-column used for the study here and the one that uses a reversed phase LC column (Wörmer et al., 2013) using a fresh sample of North Atlantic SPM. Looking at the ratio of MH:DH:HPH for cren and GDGT-0, the NP IPL method may underestimate the MH-IPLs by a factor of 10. However, even when taking this into account, this does not change the conclusion of this study. We have focused our discussions on the HPH variety of the IPLs as this is the best life marker among the IPLs. MH-GDGTs are certainly produced directly by archaea but can also have a sizable fossil contribution and are decay products of DH- and HPH-GDGTs, especially in stored extracts.

3) Lastly, the authors should to assess why the fractional abundances of IPLs are so different between the current study and their earlier study on the same samples (Lengger et al., 2012)

Lengger et al. (2012) studied only IPLs with a crenarchaeol core using an SRM-MS method, therefore in fact quantifying the fragments and not the parent molecules. Lengger et al. (2014) isolated various IPLs and quantified core GDGTs after hydrolysis. These methods are very different from the more direct detection method used in the

present study where signals are assessed in the MS1 signal. It is, therefore, not un-expected that results may vary. However, the distribution of IPLs described in our manuscript, when excluding the MH IPLS (for the reasons discussed above), actually resembles the data shown in the 2014 paper very closely. We will add some sentences concerning this topic to the manuscript.

References Lengger, S. K., Hopmans, E. C., Reichart, G.-J., Nierop, K. G. J., Sinninghe Damsté, J. S. and Schouten, S.: Intact polar and core glycerol dibiphytanyl glycerol tetraether lipids in the Arabian Sea oxygen minimum zone. Part II: Selective preservation and degradation in sediments and consequences for the TEX86, Geochim. Cosmochim. Acta, 98, 244–258, doi:10.1016/j.gca.2012.05.003, 2012. Lengger, S. K., Hopmans, E. C., Sinninghe Damsté, J. S. and Schouten, S.: Impact of sedimentary degradation and deep water column production on GDGT abundance and distribution in surface sediments in the Arabian Sea: Implications for the TEX86 paleothermometer, Geochim. Cosmochim. Acta, 142, 386–399, doi:10.1016/j.gca.2014.07.013, 2014. Wörmer, L., Lipp, J. S., Schröder, J. M. and Hinrichs, K. U.: Application of two new LC-ESI-MS methods for improved detection of intact polar lipids (IPLs) in environmental samples, Org. Geochem., 59, 10–21, doi:10.1016/j.orggeochem.2013.03.004, 2013. Zhu, C., Yoshinaga, M. Y., Peters, C. A., Liu, X. L., Elvert, M. and Hinrichs, K. U.: Identification and significance of unsaturated archaeal tetraether lipids in marine sediments, Rapid Commun. Mass Spectrom., 28(10), 1144–1152, doi:10.1002/rcm.6887, 2014.

---

## Author Comment (AC3) · 15 Jan 2018

We thank Dr. Lipp for his comments. We have responded to these below. Many comments are a reiteration of those made by reviewer 1. In those cases, we will refer back to our earlier replies made to reviewer 1.

1) The method has not been described before (with this choice of column) and it is not completely clear how the new results compare to previous installments. Most importantly, it is not clear which compounds are captured in the analytical window as according to the authors some important GDGT-0-based lipid which can be found during APCI analysis of hydrolysate cannot be found during IPL analysis. Also, the comparison of

relative proportions of lipids between samples in the current way is misleading. The ionization behavior of lipids varies strongly according to their structure and accordingly using the sum of peak areas for "total lipid abundance" and relative abundance calculated from this value is problematic. Since the authors have not used standards (and do not plan to) for correction of response factors, its perhaps best to compare absolute peak areas and report them in a table. The methodological shortcomings need to be addressed in a revised version (see also comments below).

As stated in the replies to reviewer 1, we do not agree with the reviewer that we used an undescribed method. All compounds captured with the previous type of diol column (standard analytical HPLC size) are also captured using the UHPLC version of the diol column. We also would like to add that no analytical method is fully comprehensive. There will always be compounds that will fall out of the analytical method due to polarity, molecular weight or other properties. The notion that there may be a source of GDGT-0 that is perhaps outside of the current analytical window is, therefore, not a sign that the method does not work, but simply an acknowledgement of the fact that, despite the efforts of the IPL community, we might not know everything yet. We, therefore, feel it is important to retain a discussion on this observation in the manuscript but will remove the statements regarding this "missing" GDGT-0 from the conclusion and abstract. With regards to the comparison of IPLs in relative abundances, we feel it is a fair representation of the observed peak areas and their differing contributions to the overall lipid profile to create a "summed area" and then calculate the relative contribution (expressed as a percentage) of a peak to this summed area. We don't see the difference with what the referee proposes (absolute peak areas) except that the numbers obtained are more easy to handle. As this normalization is done per depth (necessary to compare the lipid profile to the DNA profile), we agree that this way the differences in abundance between depths is unclear. We will therefore add a table to supplementary showing the total IPL peak area per sediment depth. However, we stress that although response factors may vary for different IPLs, this does not undermine the comparison of IPL occurrence and archaeal diversity (which is the main

thrust of the paper).

2) The data only compare poorly to previous results reported in Lengger et al. 2014 where absolute quantities and relative proportions of MH, DH, and HPH-GDGTs were reported. In this study, the hydrolysis method with subsequent APCI quantification was used and problematic response factors are avoided. However, the numbers just do not add up (see detailed comments). I am very surprised that this study is not included in the discussion at all. This also needs to be addressed thoroughly in a revised version.

As we state in our previous reply to reviewer 1, we acknowledge that MH has a reduced response in the NP IPL method compared to the RP IPL method. However when compared with the study of Lengger et al. (2014), we do see a good correspondence between the datasets when it comes to the DH and HPH IPLs. We will address this in more detail in the discussion of our revised manuscript.

3) Comment to author's reply to response factors raised by reviewer 1: The study by Van Mooy and Fredricks (2010, GCA) explicitly states that ". . .these RFs are not applicable for use in any future quantitative analyses of IP-DAGs using a ThermoFinnigan LCQ Deca XP ion-trap mass spectrometer or any other mass spectrometer. . .". I urge that no conclusions should be drawn from these data. It is also speculative that a 40-fold different is unlikely, as the two ion sources are different (heated vs unheated ESI) and the technology of the two mass spectrometers is also different (ion trap vs orbitrap). We probably all can agree that quantification with standards analyzed on the same machine as the environmental samples would be more appropriate here. I am more worried that the quantitative data here does not match previous data from the same samples (Lengger et al., 2014, see comments below). This hints to some methodological bias and supports suspicions from reviewer 1. These issues should be discussed.

As we state in our reply to the second comment of reviewer 1 we merely used this reference to discuss the possible differences in response factors between IPLs. We do

not intend to use it in the revised manuscript. The ion source used in this experiment is basically the same ion source that we have used for all our previous IPL work with the difference that the ESI probe now has a heating option. The heater in this experiment was set to a minimal T of 50 °C. Without active heating the source already reaches a temperature of ca. 40 C due to passive heating from the MS. However, this heating is dependent on ambient temperature and therefore fluctuates. The applied temperature of 50 °C is merely applied to stabilize this temperature and any effect it could have on ionization to make the method more reproducible but does not affect response or ionization behavior.

Detailed comments:

Line 70: typo "repertoire ".

We will modify this in our manuscript.

Line 86: was the PAF standard used somehow for quantification?

The response of the PAF standard was used to normalize for matrix effects and shifts in performance of the MS. The reported peak areas are after this correction. We will clarify this in the material and methods section of the revised manuscript.

Lines 93-98: this is a novel LCMS method that has not been published before. As reviewer 1 noted more information would be desirable in this case (the best way would be to report the new method in another peer-reviewed manuscript). It seems that the individual GDGTs with rings can be separated (judging from the supplemental figure), here some more mass chromatograms as supplemental figures would indeed be good for illustration. What other compounds can be separated, are the BDGTs/other methylated GDGTs, unsaturated GDGTs, hydroxylated GDGTs in the analytical window? Especially OMZ sediments should have abundant unsaturated GDGTs (Zhu et al., 2014, RCM). How do quantitative results compare to other published methods? Is the method suitable to comprehensively capture the archaeal lipidome?

This comment is merely a repeat of earlier comments by reviewer 1 and we therefore refer to our rebuttal to reviewer 1.

Line 101: the unit of resolution is not ppm. I also suggest to use "resolving power" instead of "resolution" as it is a better term.

We will modify this in our revised manuscript.

Line 166: according to Tab 2 it is 44.7% - I am sure this is due to rounding. However it would be great to have consistent numbers in the table and the text. Other values are also different in text and table, please check.

We will carefully check this and make the numbers consistent throughout the revised manuscript.

Lines 156-196: the "relative abundance" is given as proportion of peak area. This should at least be reported and the shortcomings of not using authentic standards need to be discussed thoroughly (see also major comments and line 381-397). Perhaps add absolute peak area values to the table?

As discussed above in our reply to Dr. Lipp's general comment, we will modify this in our manuscript. We will clearly state in the revised manuscript that no absolute quantifications were made and that the data cannot be interpreted as such.

Lines 194-196: Consistency with Lengger et al. 2012 study: please show the data of the comparison (table?). The Greek letter rho should be used for Pearson correlation coefficients - I assume that is what has been calculated here. Please add information on what is compared and how it is calculated. The letter "p" is usually used for statistical significance, p=1 would be really bad.

We do not agree with the reviewer that this would really add additional information to the manuscript. We analyzed the anoxic surface sediment sample at 885 mbsl with the same method as Lengger et al., 2012, where they studied IPL-derived GDGTs in similar sediment samples. We described this in our material and method section, lines

108-110. We separated the IPL fraction from the core lipids with the use of a silica column and flushing with MeOH. This IPL fraction was hydrolyzed for 3 hours. This to determine if the IPL-derived CL-GDGT distribution was altered due to degradation during storage. We compared the distribution of IPL derived GDGTs published by Lengger et al., 2012 to our data and found a significant correlation ( = < 0.001) between the two analyses. It was a single measurement and minor test in our study. We will clarify this more clearly in our manuscript.

Line 360: add Elling et al. 2014, 2015, 2017 and Schouten et al. 2008 references for more complete IPL inventory of Thaumarchaeota.

We will add some of the suggested references to the manuscript. It is undoable to list for every statement a complete list of references. We will select the ones that are most original or appropriate.

Line 362-363 and 367-369: the cited studies have not looked at stability of HPH-GDGTs. DH-GDGT stability also has not been experimentally assessed by the cited Lengger et al. 2012 and 2014 studies. Please discuss the stability of phosho vs gly-colipids (and possibly ester vs ether lipids as no study has compared purely phospho vs glyco ETHER lipids, cf. Logemann et al. 2010) in a more balanced way and refrain from speculation without evidence; (additional) useful references in this context are Lipp and Hinrichs 2009, Logemann et al. 2010, Schouten et al. 2010, Xie et al 2013.

We acknowledge that Harvey et al. (1986) and Schouten et al. (2010) did not study the degradation of HPH-GDGT but sedimentary phospholipids in general. We will clarify this in the manuscript. Lengger et al. (2012) and (2014) did observe that the abundance of HPH-GDGTs decreased with increasing sediment depth in contrast to the MH- and the DH-GDGTs that remained equal or even increased in abundance with increasing sediment depth. This was interpreted to reveal the less stable nature of HPH-GDGTs. In the revised version of our manuscript, we will more carefully phrase this and expand the discussion with the references mentioned.

Line 371: What about the possibility of other archaeal sources for crenarchaeol, e.g. Lincoln et al. 2014?

Lincoln et al. (2014) describes the possibility of the (archaeal) Marine Group II as a potential source for crenarchaeol. We did not detect members of the Marine Group II in our sequencing data, therefore, we do not find it relevant to discuss this here.

Line 381-397: regarding the unknown IPL type for GDGT-0, what other compounds might have been missed? It seems like potentially a major proportion! There is a lot of speculation regarding the source of an undetected GDGT-0, but how sure can you be that there is no methodological problem with the method, especially as it seems to be used for the first time? Lengger et al. 2014 have done semi-preparative IPL separation into head group classes and found abundant GDGT-0 connected to MH, DH and HPH headgroups (Fig. 5 and Table A3 in the supplemental material, here station P900 0-2 cm). Comparison of these values and Tab. 2 for the 885 mbsl surface sample shows major differences and multiple values do not match: e.g. DH-GDGT-0 and DH-GDGTcren are 63.7 and (144+86.5=230.5; incl. regioisomer) ng/g sed, respectively, a ratio of âĹij1:3. According to Tab 2 in the current manuscript the corresponding values are "ND" and 43.1 %. Why is the DH-GDGT-0 not detected here, assuming a ratio of 1:3 it should have roughly 10% contribution? Another example is found in Tab 3, it seems that the 885 mbsl surface sample is dominated by 98.1% of DH lipids. However, Lengger et al. 2014 report much higher MH abundances than DH (roughly 800 ng/g vs 400 ng/g). What is the authors' explanation for this major difference? Please discuss all data in comparison to Lengger et al. 2014. Is the diol column method not sensitive enough to capture what had been seen with the prep-hydrolysis-APCI method before? Are some compounds not detected with similar efficiency as suspected by reviewer 1? Can the discrepancies be due to that what the authors report as "relative abundance" is in fact the relative proportion of total peak area for compounds which are known to behave dramatically different during ionization and cannot be simply summed together? A meaningful comparison is only possible with

standards and correction of response factors. Again, this comparison is important and needs to be discussed, especially before invoking unidentified and undetected IPL headgroup types for GDGT-0 which are somehow not in the analytical window of a new analytical method which has not been previously published.

We feel we have sufficiently replied to the analytical issues and will not repeat our answers here. We would like to point out that Lengger et al. (2012) and (2014) performed an indirect quantification of the GDGT-IPLs. Since this previous work used different approaches than we applied in the current study, it is not unexpected to find some difference. MH-IPL is also a degradation product of other IPLs and a portion of the differences may simply be due to differences in sample handling, which we now keep to an absolute minimum. Additionally, and as stated earlier, we acknowledge the underestimation of MH-GDGTs but as stated above, we would like to retain a paragraph of discussion on this. We had a detailed look into our analyses with regards to the discrepancy between the DH-GDGT-0 abundance between our study and Lengger et al. (2014). However, we truly did not detect any DH-GDGT-0 within our sample, the surface sediment at 885 mbsl, whereas our method is clearly able in doing so. This discrepancy may perhaps be explained by resolution differences in these two studies. Lengger et al., 2014 analyzed the top 0-2 cm, while we analyzed the surface with a resolution of 0-1 cm.

Line 384: is the reference Lengger et al. 2014 correct? Or do the authors want to cite the 2012 study? As explained above, the 2014 manuscript is probably as important as the 2012 study.

Lengger et al. (2014) is a valid study on the same suite of samples but uses an entirely different method. Also the sampling resolution in the 2014 paper is different compared to the current study. Nevertheless, we see the point of the referee and we will compare the Lengger et al (2014) data more closely with our results in the revised discussion.

Line 403: A known source are the Sulfolobales. Add to discussion, see also line 414-

417 comment.

This is mentioned further on in the manuscript as indicated by the reviewer (lines 414-416): "GDGT-0 with a cyclopentanetetraol head group has been previously detected in cultures of the hyperthermophilic crenarchaeal Sulfolobales (Langworthy et al., 1974; Sturt et al., 2004)." We did not find any evidence in our sequencing data for the presence of these hyperthermophilic archaea.

Line 409: is there evidence for selective preservation of water column GDGTs at this station? This would require a dramatic degradation of GDGT-0 which is abundant in SPM within the OMZ (Line 404). It seems more likely that the lipids are produced within the sediments. Rearrange.

As stated before, the lack of IPL-GDGT-0 could be partially explained by the underestimation of the MH IPLs. We will clarify this in the manuscript.

Line 410-412: the DH isomers have been reported previously in Elling et al. 2014 and 2017. Elling et al. 2017 studied the lipidome of several thaumarchaeal cultures and found isomers not only for DH-GDGTs but also for hydroxylated DH-GDGTs. Also, they found these structures not only for the acyclic structure (GDGT-0) but also for core lipids with more rings. Please add references to prior studies and discuss why only GDGT-0 has been found in this study.

We will add the references to our manuscript and clarify the text accordingly.

Line 413-414: which moieties? And what cannot be compared? Please clarify.

We are refereeing to the DH moieties, as mentioned in the previous sentences. We could not compare this because of the missing archaeal community compositions in these depths. We will clarify this in the manuscript.

Line 414-417: the GDGT-0 with a cyclopentanetetraol (formerly called GDNT) has not only been found in pure cultures of Sulfolobales but was also detected in sediments (Sturt et al., 2014; Lipp and Hinrichs, 2009). The sources in the present study are

likely the same as in these two studies. I assume the authors mean "microbial sources" and not "sources" (Line 416). Please rephrase. Have Sulfolobales been found in the molecular biological data?

See reply to earlier comment on line 403. Line 416, we will modify the text of the manuscript accordingly.

Lines 418-437: I do not think there should be references in the conclusions section. If you want to have references here, add all relevant ones (see below and reviewer 1 comments).

We agree with the reviewer, we will remove the references from the conclusion.

Line 420: what does "specially" mean here? I would rephrase to ". . .we have unraveled the high diversity of benthic archaea harbored in anoxic sediments of the Arabian Sea, as well as. . .". Remove "specially" and add "Arabian Sea".

With specially we referred to the high diversity of Archaeal groups in the anoxic sediments compared to the oxygenated sediments. We will change this in the manuscript.

Line 421: "increasing the repertoire of archaeal intact polar lipids detected" sounds as if many new archaeal IPLs were found. However, all the described lipids (and many many more that have been found in similar environmental samples) have already been described in the literature (see also reviewer 1 comments). Please rephrase or remove statement.

We agree with the reviewer on the ambiguity of the sentence. We will clarify in the manuscript that we did not detect any unknown IPLs. The focus of our manuscript is not the identification of new archaeal IPLs but to provide more information on their potential sources by a comparison with genetic data.

Line 424: add Elling et al. 2014 reference.

We will add the reference to our manuscript.

Line 426: change "important" to "abundant".

We will modify this into "relatively abundant".

Line 429: rephrase to ". . .which could either be attributed to a fossil signal. . . or being IPLs synthesized. . ." (add "either").

We will modify this in our manuscript.

Line 431: why are these GDGT-0 derivatives "unusual"? They have been described in the literature before (e.g. Elling et al. 2014, 2017 for DH-GDGT isomers, HCP (then labelled as GDNT) in Sturt et al. 2004 and Lipp and Hinrichs, 2009).

Fair enough, we will mention this in the revised version.

Line 437: this implies the authors have only assumed a very low diversity before they analyzed the samples. Why is that? It is known that there is a large variety of IPLs in the environment (see all the details that reviewer 1 has posted in his/her comments), what was the rationale for a low diversity in these samples? I suggest removing this statement as it is not relevant what the authors have assumed.

We will remove the statement

Table 2/3: use consistent names "subsurface/deep".

We will alter this in our manuscript.

Fig 1: add "10-12 cm" to "subsurface", also in tables and other figures.

We will add this in our manuscript.

References Lengger, S. K., Hopmans, E. C., Reichart, G.-J., Nierop, K. G. J., Sinninghe Damsté, J. S. and Schouten, S.: Intact polar and core glycerol dibiphytanyl glycerol tetraether lipids in the Arabian Sea oxygen minimum zone. Part II: Selective preservation and degradation in sediments and consequences for the TEX86, Geochim. Cosmochim. Acta, 98, 244–258, doi:10.1016/j.gca.2012.05.003, 2012. Lengger, S. K.,

[Figure]

Hopmans, E. C., Sinninghe Damsté, J. S. and Schouten, S.: Impact of sedimentary degradation and deep water column production on GDGT abundance and distribution in surface sediments in the Arabian Sea: Implications for the TEX86 paleothermometer, Geochim. Cosmochim. Acta, 142, 386–399, doi:10.1016/j.gca.2014.07.013, 2014.

---

## Author Response (AR1)

**Comments to reviewer 1 and modifications to the manuscript.**

We thank the reviewer for the constructive comments. We are pleased that that he/she liked the sequencing part of our study. The reviewer raises several concerns regarding our IPL analysis to which we would like to respond below.

1.  I urge the authors to re-analyze their data using more appropriate reversed phase HPLC-MS methods. Previous studies have shown that diol column HPLC-MS and other normal phase methods lead to severe underestimation of the abundances of glycosidic GDGTs (Wörmer et al., 2013; Zhu et al., 2013). This would explain the unusually high abundances of HPH-GDGT reported in the present study and the lack of other ubiquitous compounds such as hydroxylated and unsaturated GDGT (Liu et al., 2012; Zhu et al., 2014b) and BDGT/PDGT (Becker et al., 2016; Meador et al., 2015; Zhu et al., 2014a). Alternatively, re-analysis of the samples along with authentic standards would help to correct for different ionization efficiencies based on headgroup types. If the authors did perform such a correction (and if they did not do this) they need to acknowledge this in the methods and discussion sections

    *We are familiar with the reversed phase HPLC-MS method published by Wörmer et al., (2013). This is certainly an elegant addition to the lipidomics toolbox, however, we do not agree with the complete dismissal of the LC-MS method originally introduced by Sturt et al., (2004) using a diol column. We have introduced several improvements on the original method (see response to comment below for details), which have considerably improved the performance of the method. The reviewer hints at a systematic bias in the used diol column HPLC-MS method that would lead to an extreme overrepresentation of HPH-GDGTs vs. glycosidic GDGTs. Indeed, it is well known that identical core lipids with different head groups can have very different response factors. Additional drivers of response factors are structural features such as length of the core lipid, degree of unsaturation, number of rings or functionalities like hydroxylation. Van Mooy and Fredricks (2010) already published an estimation of these response differences using the diol column-MS method, based on diacylglycerols with various head groups. Interestingly, they report very similar response factors for diacyls with either a phosphoglycerol or a hexose headgroup, while a diacyl with a dihexose head group has a response factor of one third of the PG and half of the monohexose. While we acknowledge that HPH-GDGTs are likely to have a higher response factor than MHGDGTs or DH-GDGTs, these differences are unlikely to explain the 40 fold difference in apparent relative abundance we observed in e.g. the surface sample at 3000 m depth. The other IPLs mentioned, i.e. unsaturated GDGT and BDGT/PDGT core lipids with varying head groups, have been reported to be absent in several settings (Becker et al., 2016; Zhu et al., 2014b) or were linked to archaeal producers not detected in our reported archaeal community, and it is, therefore, not a given fact that we should detect them in the Arabian Sea sediments analyzed in our study. Unsaturated GDGT IPLs are not commonly detected; Zhu et al. (2014a) reported that unsaturated GDGT IPLs were not detected in sediments in the upwelling region of NW Africa and only in trace amounts in the eastern Mediterranean Sea. The absence of unsaturated GDGT IPLs in our samples is therefore not unsuspected and certainly does not prove that our analysis method is inappropriate as stated by the referee. We screened our extracts for a variety of IPL BDGTs and PDGTs with an extensive list of possible polar head group combinations as indicated in our supplementary Table S1.*

*However, none of these IPL compounds were detected. Becker et al, (2016) reported that BDGTs and PDGTs were only detected in Methanomassiliicoccales and not in the other studied archaeal lipidomes. Meador et al. (2015) detected MH-GDGTs in an estuarine sediment sample and linked this to a high relative abundance of MCG. However, the notable 13C depletion of the BDGTs reported by Meador et al. (2015) links this compound to relatives of members of Methanomassiliicoccales, members of Methanomassiliicoccales were not detected in our sequencing data. Becker et al. (2015) did not detect BDGTs in samples with relative low concentrations of total organic carbon (TOC), our samples have relative low TOC which can explain the absence of BDGTs. Regarding hydroxylated GDGT-IPLs, we routinely detect these in biomass as well as in environmental samples (data not yet published). We are, therefore, confident that, would they have been present, we would have detected them. The reviewer suggests to reanalyze the samples against standards to aid in quantification. We have purposefully refrained from quantifying the detected GDGT-IPLs because authentic quantitative standards truly representative of the GDGT-IPLS are not available and quantifying these compounds based on standards that have very different core lipids and much less complex head groups, in our opinion, does not lead to a valid quantification. We do use standards to continuously monitor the performance of the analytical system.*

*We have added text to the materials and methods section clarifying that we for correct peak area using the PAF internal standard but do not correct for any differences in response factors (line 114 – 116)*

*Finally, re-analysis of the samples at this time would probably lead to biased results due to the storage time of 2-3 years. In our hands, samples that are re-analyzed after such a lengthy storage period do show very different ratios for several IPL-GDGTs.*

2. Judging from the supplementary spreadsheet, the authors may have considered a wider range of archaeal lipids, not exclusively traditional intact polar tetraethers, than reported in the manuscript. If so, where are these data? However, it is questionable if this comprehensive analysis is possible with the author's choice of chromatography. A more comprehensive analysis of the archaeal lipidome (e.g., by analyzing intact polar archaeols including core lipid structural modifications) in these samples would enhance source assignments and comparison to previous studies (Meador et al., 2015; Yoshinaga et al., 2015), and provide new biomarkers or source assignment for groups such as MCG and Woesearchaeota. This analysis would then satisfy the author's conclusion that the known diversity of archaeal IPLs was significantly expanded.

*For our analysis, we made use of high resolution MS with a mass range of 375- 2000 m/z. Our analytical routine involves a data dependent analysis of the 10 most abundant masses in each MS1 spectrum with a dynamic exclusion window of 6 sec (we have added this information to the method section), maximizing our capacity to identify lipids. In addition, our method works with an inclusion list, where we specify masses of compounds that, if detected, will be targeted for a product spectrum. This inclusion list contains the exact mass of most GDGT and archaeol-IPLs known at the time of analysis of this sample set. We have used all this available data to search for the compounds in the inclusion list and for additional unknown IPLs with a GDGT or archaeol core. We have successfully detected additional GDGTs as well as archaeol-based IPLs in other environmental sample sets and archaeal biomass (data not yet published). We are of the opinion that there is also sometimes information in*

*the absence of a certain set of compounds and therefore we reported our inclusion list to indicate the breadth of our search. In contrast to what the referee suggests, we do not conclude that we are expanding the known diversity of archaeal IPLs. We wrote that we expanded the screening for IPLs in our lab based on the use of UHPLC chromatography and high resolution mass spectrometry (quadrupole-orbitrap hybrid). We adjusted the manuscript to make this point more clear (line 444-447).*

3. Further, because the authors used a column different from the one described in their method reference, the suitability of the chromatography for IPL analysis cannot be assessed. The authors should provide annotated TIC or EIC traces for representative samples in supplementary figures. Importantly, the type of chromatography used here does not provide chromatographic separation of IPLs by core lipid type (e.g., GDGT-0, GDGT-1), but the way the authors present their data implies the contrary. If the different core lipid types of each IPL co-eluted in their analyses, the relative abundances reported would be questionable. At the very least the authors would need to do an isotope peak correction and then report in detail how this was done (correction factors for each IPL-core lipid combination). It would be much better to use a method than can actually chromatographically separate IPL-GDGTs by core lipid structure (Zhu et al., 2013). Additionally, the method used for this study likely cannot distinguish between the core lipid crenarchaeol and it's regioisomer. However, crenarchaeol more abundant than the regioisomer in MH GDGT and HPH GDGT but lower than its regioisomer in DH GDGT in thaumarchaeal cultures (Elling et al., 2017). This limitation needs to be addressed either by using a different chromatographic method or at least by discussing this issue in the manuscript

   *Indeed, the column used for the work described here is a different column from the original method. It is a UHPLC silica based diol column used with the same solvent system as originally described by Sturt et al., (2004) and the overall chromatographic patterns have thus not changed but the resolution of the chromatography has significantly improved. We added information on the improved chromatography to the supplemental information. Due to the improved separation power of the column, we can separate GDGT-IPLs with different cores but the same head group. The suggested isotopic corrections are, therefore, not necessary. The reviewer is correct that we cannot separate crenarchaeol from crenarchaeol regio-isomer with identical head groups. We are aware that Zhu et al. (2013) reported a separation between IPL crenarchaeol and crenarchaeol regio-isomer with the same head group. With our method this is not feasible. We have added a statement to the material and method section that IPL crenarchaeol is the sum of crenarchaeol and crenarchaeol regio-isomer (line 116-120). We stress that for the purpose of our paper this separation is not required. Lengger et al. (2012) reported core lipid content in these samples and did not find dominance of crenarchaeol regio-isomer over crenarchaeol. We, therefore, do not think that the culture study the reviewer refers to is relevant in this context. We have added EIC traces for MH-GDGTs, DH GDGTs and HPH-GDGTs to the supplementary data to clarify our peak separation.*

4. Given these fundamental limitations, major parts of the methods, results, and discussion sections need to be revised. However, HPLC-MS (re-)analysis with a different method could be achieved within a day or two, given that the current study only includes 8 samples. I hope that the authors will agree that this approach has the potential to greatly expand the significance of their work with only moderate additional effort.

*As discussed above, we do not agree with the dismissal of the diol column based HPLC-MS method. As explained above, we feel that reanalysis of the samples at this time with the reversed phase method would not lead to valid results due to the age of the extracts. We respectfully disagree with the reviewer that our results, and therefore our interpretation and discussion, are invalid.*

5. Other comments: Line 27: DeLong et al. report only on Antarctic samples. You may want to choose a reference that discusses a more diverse set of environments (e.g. DeLong and Pace, 2013; Schleper et al., 2005) or a collection of references e.g., (DeLong, 1992; Fuhrman et al., 1992; Teske and Sørensen, 2008).

*The addition of literature would benefit the introduction, we have added references to the manuscript as suggested by the reviewer. We assumed that the reviewer referred to the paper of DeLong and Pace (2001). (line 27-28)*

6. Line 31: Lloyd et al. only provide data on two archaeal groups. You may rather cite a paper that actually discusses archaeal diversity, such as Teske and Sørensen (2008) or Teske (2013) some of the more recent literature, e.g., Hug et al. (2016) or Spang et al. (2017).

*We modified the manuscript and added some of the proposed literature references. However, we feel that it is also important to cite original references where pioneering efforts have been described. (line 32-33)*

7. Line 40-42: What are the metabolisms of the other archaeal groups?

*We mention the metabolism of detected groups in our discussion, however for most of the detected groups metabolism is unknown as neither cultures nor metagenomes are available to predict their metabolic capabilities. For the rest i.e. MBG-B and E, potential metabolism is derived from (meta)genome information as shown already in lines 334-336.*

8. Line 59-61: What about the Lincoln et al. (Lincoln et al., 2014) paper?

*Lincoln et al. (2014) indeed reported the marine group II as potential producers of crenarchaeol. However, there is still an ongoing discussion on this hypothesis (Lincoln et al., 2014b; Schouten et al., 2014). We added this reference and discussion into our introduction (lines 62-64).*

9. Line 86: Why was this standard added? Was it used for any correction?

*This standard (PAF; 1-O-hexadecyl-2-acetoyl-sn-glycero-3-phosphocholine) was added to continuously monitor the performance of the MS response and to correct for matrix effects. We also corrected peak areas accordingly. This has been clarified in the materials and methods section. (line 114-116)*

10. Line 90: Specify the modifications.

*The modifications are already described in detail in the text after the referral to the original Sturt et al. (2004) method, lines 93-120. We have added more details as mentioned above in the response to previous comments*

11. Line 159-161: Two DH isomers were also reported earlier by Elling et al. (2014; 2017).

*Indeed, Elling et al. (2014) reported an "early eluting " DH-GDGT and Elling et al. (2017) reports early eluting isomers of DH- and DH-OH-GDGTs. We changed our manuscript to acknowledge these reports. However, although Elling et al. (2017) speculates on the nature of the isomerisation, no spectral details are given. We show, for the first time, spectral data on these isomers and based on these spectra, interpret the nature of the isomerization. We, therefore, feel it is worthwhile to include this information in the supplementary information.*

12. Line 166-171: Did you detect crenarchaeol regioisomer? If not, why not? Coelution with crenarchaeol? This should be pointed out here or elsewhere.

    *As stated above, the crenarchaeol regio-isomer IPLs co-elute with crenarchaeol with the same head group. We have added a statement regarding this to our material and method section. (line 116-120)*

13. Line 172-173: The way the % values are used in this section is very confusing (. . .37.6% of 36.6%) etc. It could help to provide a further table in addition to Table 2 that shows the samples arranged by depth or headgroup type instead of sorting by core GDGT type.

    *We clarified this in the manuscript to make our data more clear. We also refer to the original Table 3 in our manuscript where the data are sorted according to the different headgroup types per sampling depth.*

14. Line 196: What does "p=1.00" represent? The correlation between our IPL derived CL-GDGTs and the ones reported by Lengger et al., (2012).

    *The noted correlation is adjusted in the manuscript. We also added some sentences to the material and method section in order to clarify our goal with the analysis.* ==(material and method section: line 121-125, corrected correlation: line 213-215)==

15. Line 254-260: How were the transcripts analyzed? I did not find this information in the methods section. How long were the samples stored before analysis? The transcripts were analysed as described in Pitcher et al., (2011).

    *We refer to this in our material and method section. As mentioned, the sediments were sliced on board and directly stored at -80C. They were kept at -80C until lipid and DNA/RNA extractions. After the extractions the DNA/RNA and the cDNA were stored at -80C until analyses.*

16. Line 270-274: This sentence is a bit long and complex. Revise?

    *The referee is correct,* w*e have adjusted this in the manuscript.* ==(line 290-294)==

17. Line 281: "Differences"?

    *The referee is correct, it should be differences instead of difference. We altered this. (line 302)*

18. Line 297: Rather "overlying"?

    *The referee is correct, it should be overlying. We altered this. (line 318)*

19. Line 319-321: Could you point out whether there is any evidence for the occurrence of Woesearchaeota in the water column and whether you would expect this group and the other archaeal groups to be present in oxic vs. anoxic environments (or both).

*There is evidence for Woesearchaeota in marine pelagic oxygenated environments, (Liu et al., 2017). The Arabian Sea contains an oxic environment and an oxygen minimum zone, so far it is uncertain if the Woesearchaeota could reside in the anoxic pelagic zone. Woesearchaeota were recently also detected, in relative high abundances, in oxygenated surface sediments (Lipsewers et al., 2017)*

20. Line 353-357: I disagree with the statement that the diversity of detected (detectable) IPLs was greatly enhanced by this study. There are two IPL types described in addition to those reported by (Lengger et al., 2012). However, the diversity of IPLGDGTs reported by previous studies is much higher, e.g. Yoshinaga et al. (2015), with respect to both the headgroup types as well as the structural modifications in the core lipid such as hydroxylation, unsaturation, methylation, monoalkylation, trialkylation, or substitution of glycerol with butanetriol or pentanetriol. Many of these would likely be detectable in the presented samples using different methods.

*In contrast to what the referee suggests, we do not conclude that we are expanding the known diversity of archaeal IPLs (see remark on point 2). However, we adjusted the sentences in order to be more clear about our findings. (line 445-447)*

21. Line 359-360: GDGT-0 and crenarchaeol are also dominant core lipids of MHGDGT in Thaumarchaeota (Elling et al., 2014; Elling et al., 2015; Elling et al., 2017; Schouten et al., 2008). MH-GDGT also appears to be the dominant membrane lipid in many Thaumarchaeota. Also, how does your interpretation fit to the results of Lincoln et al. (2014) who suggested production of crenarchaeol by other archaeal groups?

*We also see a high relative abundance of IPL GDGT-0 and IPL crenarchaeol in the surface sediment samples coinciding with a high relative abundance of thaumarchaeota. Lincoln et al. (2014) suggested that marine group II (MGII) may be significant contributors of crenarchaeol in oceanic surface waters based on the combination of core lipid (CL)-GDGT and metagenomic analyses. However, the lack of culture representatives of this group and the difference in the resilience times between CL-GDGTs (attributed to dead biomass) and DNA weakens the arguments of MGII as important GDGTs producers (Lincoln et al., 2014b; Schouten et al., 2014). Members of the MGII are, so far, only found in the pelagic and not in the benthic environment therefore we did not discuss them in our manuscript. The archaeal groups that we detected are, currently, lacking cultured representatives and therefore their membrane lipid composition is still uncertain.*

**Second comments to reviewer 1 and modifications to the manuscript.**

We thank the reviewer again for the additional comments. Here are our replies.

1. I agree with many of the modifications proposed by the authors. Most importantly, since they use an unpublished method, they will need to demonstrate in detail the separation of the different IPL-GDGT cyclized GDGT-headgroup combinations, and exclude, e.g. the co-elution of unsaturated GDGTs with regular GDGTs.

*As stated in our previous reply to comments regarding the use of a modifications to the original normal phase IPL method, we do not agree with the reviewer that this represents an "unpublished method". The basis of the method is unchanged and thus we do not feel the functioning of the method deserves a lengthy discussion in a paper that focusses on different issues. As indicated before, we illustrated the improvements made in the method by adding illustrative figures in the supplemental information. We cannot fully exclude the co-elution of unsaturated GDGTs-IPLs with regular GDGT-IPLs. Per suggestion of the reviewer we re-analyzed the anoxic sediment samples (surface at 885 mbsl, and subsurface of all the stations) using a reversed phase LC/MS method to estimate the contribution of unsaturated core GDGTs to the GDGT-CL pool. As expected after a couple of years of storage the IPLs were partially decayed and could therefore not be reinterpreted. We did observe a trace of potentially unsaturated CL-GDGTs (e.g. GDGT-0:1) but these represented <0.25% of the total CL pool.*

*We have added figures (Supplemental Figs. 3a, b and c) to our supplementary information to illustrate our used method.*

2. I appreciate that determining response factors for GDGTs is very challenging due to the lack of commercial standards but I think that this would have been perfectly feasible for the small number of compounds the authors have analyzed. I urge the authors to clearly state that they have not used response factors and how this affects interpretation of the results. It is not acceptable to cite the response factors from Van Mooy & Fredricks (2010) since they were measured on a different chromatographic method and different instruments and on compounds that are not comparable to GDGT IPLs. GDGT-IPLs are twice the size of the standards measured in that paper and thus the relative ionization efficiencies will be dramatically different (this will also change the relative effect of one versus two glycosidic moieties). Further, ionization of GDGT-IPLs will heavily depend on elution time, since the method chosen by the authors uses a strong polarity gradient. This will strongly affect the efficiency of ionization of GDGTs relative to their retention time and will likely lead to a systematic underestimation of more apolar GDGT-IPLs (e.g. MH) that elute earlier when the mobile phase consists mostly of hexane (which is not beneficial for ionization).

*We do not agree with the referee that isolation of standards to the purity required for use as standards (i.e. >95%) to determine response factors is "perfectly feasible". In fact, it is a large project in itself that requires a large amount of source material (hard to culture archaea), many months of experimental work and large amounts of solvents to do the purification as well as proofing of the material by NMR and mass spectrometry. Given the fact that isolated IPLs are also unstable, and the fact that absolute quantitation does not contribute to the discussion (as we focus on relative distributions in comparisons with the genetic fingerprint of the sediments), we do not feel the effort was or is justified. We have already indicated in our response to the reviewer's earlier comments that we clarified in our materials and methods that we do not use response factors to correct peak areas. We therefore also do not plan to cite the Van Mooy & Fredricks (2010), or apply their published response factors, but merely used this work to illustrate what the impact of the difference in response factors might be.*

*We have assessed the difference between the HPLC-MS method using a diol-column used for the study here and the one that uses a reversed phase LC column (Wörmer et al., 2013) using a fresh sample of North Atlantic SPM. Looking at the ratio of MH:DH:HPH for cren and GDGT-0, the NP IPL method may underestimate the MH IPLs by a factor of 10. However, even when taking this into account, this does not change the conclusion of this study. We have focused our discussions on the HPH variety of the IPLs as this is the best life marker among the IPLs. MH-GDGTs are certainly produced directly by archaea but can also have a sizable fossil contribution and are decay products of DH- and HPH-GDGTs, especially in stored extracts.*

*For the anoxic surface sediment at 885 mbsl we observed a discrepancy between the relative abundance of IPL derived CL-GDGT-0 and the IPL-GDGT-0 relative abundance reported in our manuscript. Even if our method underestimates the relative abundance of MHs, this discrepancy cannot be fully explained. We, therefore, remained this part in our discussion. However, we also highlight the fact that we underestimate the relative abundance of MHs in our study.*

3. Lastly, the authors should to assess why the fractional abundances of IPLs are so different between the current study and their earlier study on the same samples (Lengger et al., 2012)

   *Lengger et al. (2012) studied only IPLs with a crenarchaeol core using an SRM-MS method, therefore in fact quantifying the fragments and not the parent molecules. Lengger et al. (2014) isolated various IPLs and quantified core GDGTs after hydrolysis. These methods are very different from the more direct detection method used in the present study where signals are assessed in the MS1 signal. It should also be noted that Lengger et al. (2014) studied at a different sampling resolution (surface sediment used was 0-2 compared to 0-0.5 cm in our case). It is, therefore, not unexpected that results may vary. However, the distribution of IPLs described in our manuscript, when excluding the MH IPLS (for the reasons discussed above), actually resembles the data shown in the 2014 paper very closely. We added some sentences to our discussion comparing the data published by Lengger et al. (2012; 2014) with our data. We also highlighted that differences were likely resulted from ionization efficiencies. (line 378-380, 409-410)*

**Comments to Dr. Lipp (reviewer 2) and modifications to the manuscript.**

We thank Dr. Lipp for his comments. We have responded to these below. Many comments are a reiteration of those made by reviewer 1. In those cases, we referred back to our earlier replies made to reviewer 1.

1. The method has not been described before (with this choice of column) and it is not completely clear how the new results compare to previous installments. Most importantly, it is not clear which compounds are captured in the analytical window as according to the authors some important GDGT-0-based lipid which can be found during APCI analysis of hydrolysate cannot be found during IPL analysis. Also, the comparison of relative proportions of lipids between samples in the current way is misleading. The ionization behavior of lipids varies strongly according to their structure and accordingly using the sum of peak areas for "total lipid abundance" and relative abundance calculated from this value is problematic.

Since the authors have not used standards (and do not plan to) for correction of response factors, its perhaps best to compare absolute peak areas and report them in a table. The methodological shortcomings need to be addressed in a revised version (see also comments below).

*As stated in the replies to reviewer 1, we do not agree with the reviewer that we used an undescribed method. All compounds captured with the previous type of diol column (standard analytical HPLC size) are also captured using the UHPLC version of the diol column. We also would like to add that no analytical method is fully comprehensive. There will always be compounds that will fall out of the analytical method due to polarity, molecular weight or other properties. The notion that there may be a source of GDGT-0 that is perhaps outside of the current analytical window is, therefore, not a sign that the method does not work, but simply an acknowledgement of the fact that, despite the efforts of the IPL community, we might not know everything yet. We, therefore, feel it is important to retain a discussion on this observation in the manuscript but will remove the statements regarding this "missing" GDGT-0 from the conclusion. With regards to the comparison of IPLs in relative abundances, we feel it is a fair representation of the observed peak areas and their differing contributions to the overall lipid profile to create a "summed area" and then calculate the relative contribution (expressed as a percentage) of a peak to this summed area. We don't see the difference with what the referee proposes (absolute peak areas) except that the numbers obtained are more easy to handle. As this normalization is done per depth (necessary to compare the lipid profile to the DNA profile), we agree that this way the differences in abundance between depths is unclear.*
*We added an additional column to table 2, indicating the total IPL peak area per sample.*

2. The data only compare poorly to previous results reported in Lengger et al. 2014 where absolute quantities and relative proportions of MH, DH, and HPH-GDGTs were reported. In this study, the hydrolysis method with subsequent APCI quantification was used and problematic response factors are avoided. However, the numbers just do not add up (see detailed comments). I am very surprised that this study is not included in the discussion at all. This also needs to be addressed thoroughly in a revised version.

*As we state in our previous reply to reviewer 1, we acknowledge that MH has a reduced response in the NP IPL method compared to the RP IPL method. However when compared with the study of Lengger et al. (2014), we do see a good correspondence between the datasets when it comes to the DH and HPH IPLs. Again, it should be noted that Lengger et al. (2014) used a different sampling resolution (surface sediment used was 0-2 compared to our 0-0.5 cm). It is, therefore, not unexpected that results may vary. We have added a large part in our discussion to compare the differences IPL distributions between our study and the previously published studies of Lengger et al., (2012; 2014). In this section we also raised the issue of varying ionization efficiencies among IPLs. (line 378-380, 409-410)*

3. Comment to author's reply to response factors raised by reviewer 1: The study by Van Mooy and Fredricks (2010, GCA) explicitly states that ". . .these RFs are not applicable for use in any future quantitative analyses of IP-DAGs using a ThermoFinnigan LCQ Deca XP ion-trap mass spectrometer or any other mass spectrometer. . .". I urge that no conclusions should be drawn from these data. It is also speculative that a 40-fold different is unlikely, as the two ion sources are different (heated vs unheated ESI) and the technology of the two mass spectrometers is also different (ion trap vs orbitrap). We probably all can agree that quantification with standards analyzed on the same machine as the environmental samples would be more appropriate here. I am more worried that the quantitative data here does not match previous data from the same samples (Lengger et al., 2014, see comments below). This hints to some methodological bias and supports suspicions from reviewer 1. These issues should be discussed.

*As we state in our reply to the second comment of reviewer 1 we merely used this reference to discuss the possible differences in response factors between IPLs. We do not intend to use it in the revised manuscript. The ion source used in this experiment is basically the same ion source that we have used for all our previous IPL work with the difference that the ESI probe now has a heating option. The heater in this experiment was set to a minimal T of 50 °C. Without active heating the source already reaches a temperature of ca. 40 C due to passive heating from the MS. However, this heating is dependent on ambient temperature and therefore fluctuates. The applied temperature of 50 °C is merely applied to stabilize this temperature and any effect it could have on ionization to make the method more reproducible but does not affect response or ionization behavior.*

Detailed comments:

4. Line 70: typo "repertoire ".

   *Modified in the manuscript (line 73)*

5. Line 86: was the PAF standard used somehow for quantification?

   *The response of the PAF standard was used to normalize for matrix effects and shifts in performance of the MS. The reported peak areas are after this correction. We have added a couple of sentences on the use of PAF as internal standard and the correction. (line 114-117)*

6. Lines 93-98: this is a novel LCMS method that has not been published before. As reviewer 1 noted more information would be desirable in this case (the best way would be to report the new method in another peer-reviewed manuscript). It seems that the individual GDGTs with rings can be separated (judging from the supplemental figure), here some more mass chromatograms as supplemental figures would indeed be good for illustration. What other compounds can be separated, are the BDGTs/other methylated GDGTs, unsaturated GDGTs, hydroxylated GDGTs in the analytical window? Especially OMZ sediments should have abundant unsaturated GDGTs (Zhu et al., 2014, RCM). How do quantitative results compare to other published methods? Is the method suitable to comprehensively capture the archaeal lipidome?

   *This comment is a repeat of earlier comments by reviewer 1 and we therefore refer to our rebuttal to reviewer 1.*

7. Line 101: the unit of resolution is not ppm. I also suggest to use "resolving power" instead of "resolution" as it is a better term.

   *We changed "resolution" into "resolving power" in our manuscript. (line 104-106)*

8. Line 166: according to Tab 2 it is 44.7% - I am sure this is due to rounding. However it would be great to have consistent numbers in the table and the text. Other values are also different in text and table, please check.

*We have checked the manuscript and changed the numbers accordingly.*

9. Lines 156-196: the "relative abundance" is given as proportion of peak area. This should at least be reported and the shortcomings of not using authentic standards need to be discussed thoroughly (see also major comments and line 381-397). Perhaps add absolute peak area values to the table?

*As discussed above in our reply to Dr. Lipp's general comment, we have modified this in our manuscript. We added a statement in the revised manuscript that no absolute quantifications were made and that the data cannot be interpreted as such.*

10. Lines 194-196: Consistency with Lengger et al. 2012 study: please show the data of the comparison (table?). The Greek letter rho should be used for Pearson correlation coefficients - I assume that is what has been calculated here. Please add information on what is compared and how it is calculated. The letter "p" is usually used for statistical significance, p=1 would be really bad.

*We do not agree with the reviewer that adding the Lengger et al. data would really add additional information to the manuscript. We analyzed the anoxic surface sediment sample at 885 mbsl with the same method as Lengger et al. (2012), where they studied IPL-derived GDGTs in similar sediment samples. We described this in our material and method section, lines 121-125. We separated the IPL fraction from the core lipids with the use of a silica column and flushing with MeOH. This IPL fraction was hydrolyzed for 3 h. This was done to determine if the IPL-derived CL-GDGT distribution was altered due to degradation during storage. We compared the distribution of IPL derived GDGTs published by Lengger et al., 2012 to our data and found a significant correlation ($\rho$ = < 0.001) between the two analyses. We changed the notation of the statistical significance into: "(r= 0.99, $\rho$ = < 0.001)". We have added a couple of sentences to the material and method section to clarify the aim of this analysis. (line 121-125)*

11. Line 360: add Elling et al. 2014, 2015, 2017 and Schouten et al. 2008 references for more complete IPL inventory of Thaumarchaeota.

*We have added some of the suggested references to the manuscript. It is undoable to list for every statement a complete list of references. We selected the ones that are most original or appropriate. (line 383)*

12. Line 362-363 and 367-369: the cited studies have not looked at stability of HPH-GDGTs. DH-GDGT stability also has not been experimentally assessed by the cited Lengger et al. 2012 and 2014 studies. Please discuss the stability of phosho vs glycolipids (and possibly ester vs ether lipids as no study has compared purely phospho vs glyco ETHER lipids, cf. Logemann et al. 2010) in a more balanced way and refrain from speculation without evidence; (additional) useful references in this context are Lipp and Hinrichs 2009, Logemann et al. 2010, Schouten et al. 2010, Xie et al 2013.

*We acknowledge that Harvey et al. (1986) and Schouten et al. (2010) did not study the degradation of HPH-GDGT but sedimentary phospholipids in general. We clarified this in the manuscript. Lengger et al. (2012) and (2014) did observe that the abundance of HPH-*

*GDGTs decreased with increasing sediment depth in contrast to the MH- and the DH-GDGTs that remained equal or even increased in abundance with increasing sediment depth. This was interpreted to reveal the less stable nature of HPH-GDGTs. We altered the manuscript concerning the references Harvey et al, (1986) and Schouten et al. (2010), the studies describing the labile nature of sedimentary phospholipids. We assumed that the reviewer was referring to Logemann et al. (2011), we have added this reference and Xie et al. (2013) into our discussion. We also raised the topic of phospho vs glyco ether lipids in our discussion. (line 393-396)*

13. Line 371: What about the possibility of other archaeal sources for crenarchaeol, e.g. Lincoln et al. 2014?

    *Lincoln et al. (2014) describes the possibility of the (archaeal) Marine Group II as a potential source for crenarchaeol. We did not detect members of the Marine Group II in our sequencing data, therefore, we do not find it relevant to discuss this here.*

14. Line 381-397: regarding the unknown IPL type for GDGT-0, what other compounds might have been missed? It seems like potentially a major proportion! There is a lot of speculation regarding the source of an undetected GDGT-0, but how sure can you be that there is no methodological problem with the method, especially as it seems to be used for the first time? Lengger et al. 2014 have done semi-preparative IPL separation into head group classes and found abundant GDGT-0 connected to MH, DH and HPH headgroups (Fig. 5 and Table A3 in the supplemental material, here station P900 0-2 cm). Comparison of these values and Tab. 2 for the 885 mbsl surface sample shows major differences and multiple values do not match: e.g. DH-GDGT-0 and DH-GDGTcren are 63.7 and (144+86.5=230.5; incl. regioisomer) ng/g sed, respectively, a ratio of ~1:3. According to Tab 2 in the current manuscript the corresponding values are "ND" and 43.1 %. Why is the DH-GDGT-0 not detected here, assuming a ratio of 1:3 it should have roughly 10% contribution? Another example is found in Tab 3, it seems that the 885 mbsl surface sample is dominated by 98.1% of DH lipids. However, Lengger et al. 2014 report much higher MH abundances than DH (roughly 800 ng/g vs 400 ng/g). What is the authors' explanation for this major difference? Please discuss all data in comparison to Lengger et al. 2014. Is the diol column method not sensitive enough to capture what had been seen with the prep-hydrolysis-APCI method before? Are some compounds not detected with similar efficiency as suspected by reviewer 1? Can the discrepancies be due to that what the authors report as "relative abundance" is in fact the relative proportion of total peak area for compounds which are known to behave dramatically different during ionization and cannot be simply summed together? A meaningful comparison is only possible with standards and correction of response factors. Again, this comparison is important and needs to be discussed, especially before invoking unidentified and undetected IPL headgroup types for GDGT-0 which are somehow not in the analytical window of a new analytical method which has not been previously published.

    *We addressed the comments raised on the analytical issues in previous replies. We would like to point out that Lengger et al. (2012) and (2014) performed an indirect quantification of the GDGT-IPLs. Since this previous work used different approaches than we applied in the current study, it is not unexpected to find some difference. MH-IPL is also a degradation product of other IPLs and a portion of the differences may simply be due to differences in sample handling, which we keep to an absolute minimum. Additionally, and as stated earlier, we acknowledge the underestimation of MH-GDGTs but as stated above, we would like to retain a paragraph of discussion on this. We had a detailed look into our*

*analyses with regards to the discrepancy between the DH-GDGT-0 abundance between our study and Lengger et al. (2014). However, we truly did not detect any DH-GDGT-0 within our sample, the surface sediment at 885 mbsl, whereas our method is clearly able to do so. This discrepancy could be explained by resolution differences in these two studies. Lengger et al., 2014 analyzed the top 0-2 cm, while we analyzed the surface with a resolution of 0-0.5 cm. It is, therefore, not unexpected that results may vary. (line 378-380, 401-410)*

15. Line 384: is the reference Lengger et al. 2014 correct? Or do the authors want to cite the 2012 study? As explained above, the 2014 manuscript is probably as important as the 2012 study.

    *Lengger et al. (2014) is a valid study on the same suite of samples but uses an entirely different method. Also the sampling resolution in the 2014 paper is different compared to the current study. Nevertheless, we see the point of the referee and we compared the Lengger et al (2014) data more closely with our results in the revised discussion. (line 405-410)*

16. Line 403: A known source are the Sulfolobales. Add to discussion, see also line 414- 417 comment.

    *This is mentioned further on in the manuscript as indicated by the reviewer (lines 414-416): "GDGT-0 with a cyclopentanetetraol head group has been previously detected in cultures of the hyperthermophilic crenarchaeal Sulfolobales (Langworthy et al., 1974; Sturt et al., 2004)." We did not find any evidence in our sequencing data for the presence of these hyperthermophilic archaea. We have added to the discussion that we found no evidence of Sulfolobales in our samples. (line 436)*

17. Line 409: is there evidence for selective preservation of water column GDGTs at this station? This would require a dramatic degradation of GDGT-0 which is abundant in SPM within the OMZ (Line 404). It seems more likely that the lipids are produced within the sediments. Rearrange.

    *We compared the anoxic surface sediments with the overlying oxygenated surface sediments. We observe a shift in IPL-GDGT distributions, with lower GDGT-0 and higher GDGT-1, 2, 3 and 4 relative abundances within the subsurface compared to the surface sediments. This could be due to in-situ production with a preference towards IPL-GDGT-1, 2, 3 and 4 or selective preservation. This could be a factor because the majority of IPL-GDGT-0 within the surface sediments consisted with the headgroup HPH whereas the other IPL-GDGTs where dominantly detected with DHs. This is discussed (line 425-429).*

18. Line 410-412: the DH isomers have been reported previously in Elling et al. 2014 and 2017. Elling et al. 2017 studied the lipidome of several thaumarchaeal cultures and found isomers not only for DH-GDGTs but also for hydroxylated DH-GDGTs. Also, they found these structures not only for the acyclic structure (GDGT-0) but also for core lipids with more rings. Please add references to prior studies and discuss why only GDGT-0 has been found in this study.

    *We have altered the manuscript as also indicated in the comment to reviewer 1 (first comments, point 11)*

19. Line 413-414: which moieties? And what cannot be compared? Please clarify.

*We are refereeing to the DH moieties, as mentioned in the previous sentences. We could not compare this because of the missing archaeal community compositions in these depths. We have altered this in the manuscript.*

20. Line 414-417: the GDGT-0 with a cyclopentanetetraol (formerly called GDNT) has not only been found in pure cultures of Sulfolobales but was also detected in sediments (Sturt et al., 2014; Lipp and Hinrichs, 2009). The sources in the present study are likely the same as in these two studies. I assume the authors mean "microbial sources" and not "sources" (Line 416). Please rephrase. Have Sulfolobales been found in the molecular biological data?

    *We have altered this in our manuscript, see earlier comment by this reviewer (point 16) (line 436)*

21. Lines 418-437: I do not think there should be references in the conclusions section. If you want to have references here, add all relevant ones (see below and reviewer 1 comments).

    *We agree with the reviewer; we removed the references from the conclusion section.*

22. Line 420: what does "specially" mean here? I would rephrase to ". . .we have unraveled the high diversity of benthic archaea harbored in anoxic sediments of the Arabian Sea, as well as. . .". Remove "specially" and add "Arabian Sea".

    *With specially we referred to the high diversity of Archaeal groups in the anoxic sediments compared to the oxygenated sediments. We rephrased the sentence according to the reviewers suggestion. (line 445-447)*

23. Line 421: "increasing the repertoire of archaeal intact polar lipids detected" sounds as if many new archaeal IPLs were found. However, all the described lipids (and many many more that have been found in similar environmental samples) have already been described in the literature (see also reviewer 1 comments). Please rephrase or remove statement.

    *We agree with the reviewer on the ambiguity of the sentence. We altered the sentence to clarify the broadening of our analytical window and not the detection of novel IPLs. The focus of our manuscript is not the identification of new archaeal IPLs but to provide more information on their potential sources by a comparison with genetic data.*

24. Line 424: add Elling et al. 2014 reference.

    *We have added the reference to our manuscript, but removed the references from the conclusion section as suggested by the reviewer (point 21).*

25. Line 426: change "important" to "abundant".

    *We removed this part of the conclusion, because this section contained speculation of an unknown IPL-GDGT-0 source. We removed this as discussed in earlier comments.*

26. Line 429: rephrase to ". . .which could either be attributed to a fossil signal. . . or being IPLs synthesized. . ." (add "either").

    *We changed the manuscript according to the reviewers suggestion. (line 455)*

27. Line 431: why are these GDGT-0 derivatives "unusual"? They have been described in the literature before (e.g. Elling et al. 2014, 2017 for DH-GDGT isomers, HCP (then labelled as GDNT) in Sturt et al. 2004 and Lipp and Hinrichs, 2009).

*We altered the manuscript accordingly. (line 456-457)*

28. Line 437: this implies the authors have only assumed a very low diversity before they analyzed the samples. Why is that? It is known that there is a large variety of IPLs in the environment (see all the details that reviewer 1 has posted in his/her comments), what was the rationale for a low diversity in these samples? I suggest removing this statement as it is not relevant what the authors have assumed.

*We removed the statement and altered the sentence. (line 463)*

29. Table 2/3: use consistent names "subsurface/deep".

*We altered this in our manuscript.*

30. Fig 1: add "10-12 cm" to "subsurface", also in tables and other figures.

*We added this to our figures and figure legends.*

[revised manuscript text omitted]

0.02

[Figure]

**Fig. 3.**

[Figure]

0.2

     **Fig. 4.**

[Figure]

**Fig. 5.**

[Figure]

---

## Referee Report (RR1)

**Besseling et al., sample 885 (0-0.5 cm)**

| | GDGT-0 | GDGT-1 | GDGT-2 | GDGT-3 | GDGT-4 | GDGT-5 | sum |
|---|---|---|---|---|---|---|---|
| | | | GDGT relative abundance | | | | |
| MH | 0.3 | 0.1 | 0.1 | | | 1.3 | **1.8** |
| DH | | 1.6 | 29.5 | 17.8 | 6.1 | 43.1 | **98.1** |
| HPH | | | | | | 0.3 | **0.3** |
| | | | | | | | 100.2 more than 100%! |

| | GDGT-0 | GDGT-1 | GDGT-2 | GDGT-3 | GDGT-4 | GDGT-5 |
|---|---|---|---|---|---|---|
| | | | relative distribution per headgroup | | | |
| MH | 16.7% | 5.6% | 5.6% | 0.0% | 0.0% | 72.2% |
| DH | 0.0% | 1.6% | 30.1% | 18.1% | 6.2% | 43.9% |
| HPH | 0.0% | 0.0% | 0.0% | 0.0% | 0.0% | 100.0% |

**Besseling et al., sample 1306 (0-0.5 cm)**

| | GDGT-0 | GDGT-1 | GDGT-2 | GDGT-3 | GDGT-4 | GDGT-5 | sum |
|---|---|---|---|---|---|---|---|
| | | | GDGT relative abundance | | | | |
| MH | 1.1 | 0.1 | | | | 1.4 | **2.6** |
| DH | | 1.5 | 15.4 | 6.9 | 2.7 | 15.5 | **42** |
| HPH | 36.6 | 0.2 | | | | 18.7 | **55.5** |
| | | | | | | | 100.1 more than 100%! |

| | GDGT-0 | GDGT-1 | GDGT-2 | GDGT-3 | GDGT-4 | GDGT-5 |
|---|---|---|---|---|---|---|
| | | | relative distribution per headgroup | | | |
| MH | 42.3% | 3.8% | 0.0% | 0.0% | 0.0% | 53.8% |
| DH | 0.0% | 3.6% | 36.7% | 16.4% | 6.4% | 36.9% |
| HPH | 65.9% | 0.4% | 0.0% | 0.0% | 0.0% | 33.7% |

**Besseling et al., sample 3003 (0-0.5 cm)**

| | GDGT-0 | GDGT-1 | GDGT-2 | GDGT-3 | GDGT-4 | GDGT-5 | sum |
|---|---|---|---|---|---|---|---|
| | | | GDGT relative abundance | | | | |
| MH | 0.5 | | | | | 0.4 | **0.9** |
| DH | 0.1 | 0.2 | 0.9 | 0.4 | | 0.2 | **1.8** |
| HPH | 80.3 | | | | | 17.1 | **97.4** |
| | | | | | | | 100.1 more than 100%! |

| | GDGT-0 | GDGT-1 | GDGT-2 | GDGT-3 | GDGT-4 | GDGT-5 |
|---|---|---|---|---|---|---|
| | | | relative distribution per headgroup | | | |
| MH | 55.6% | 0.0% | 0.0% | 0.0% | 0.0% | 44.4% |
| DH | 5.6% | 11.1% | 50.0% | 22.2% | 0.0% | 11.1% |
| HPH | 82.4% | 0.0% | 0.0% | 0.0% | 0.0% | 17.6% |

**Lengger et al. 2014, sample P900 0-2 cm, Supplementary Table A3**

| | GDGT-0 | 1 | 2 | 3 | 4 | 5 | sum |
|---|---|---|---|---|---|---|---|
| | | | GDGT concentrations, ng/g sed dw. | | | | |
| MH | 168 | 51.3 | 65.2 | 17.6 | | 524 | **826.1** |
| DH | 63.7 | 25.4 | 77.1 | 42.6 | | 230.5 | **439.3** |
| HPH | 8.2 | 1.5 | 1.9 | 0.5 | | 10.3 | **22.4** |

| | GDGT-0 | 1 | 2 | 3 | 4 | 5 |
|---|---|---|---|---|---|---|
| | | | relative distribution | | | |
| MH | 20.3% | 6.2% | 7.9% | 2.1% | 0.0% | 63.4% |
| DH | 14.5% | 5.8% | 17.6% | 9.7% | 0.0% | 52.5% |
| HPH | 36.6% | 6.7% | 8.5% | 2.2% | 0.0% | 46.0% |

**Lengger et al. 2014, sample P1300 0-2 cm, Supplementary Table A3**

| | GDGT-0 | 1 | 2 | 3 | 4 | 5 | sum |
|---|---|---|---|---|---|---|---|
| | | | GDGT concentrations, ng/g sed dw. | | | | |
| MH | 124 | 23 | 24.4 | 6.6 | | 241.3 | **419.3** |
| DH | 33 | 12 | 26.7 | 13.2 | | 81.8 | **166.7** |
| HPH | 23.2 | 2 | 0.7 | 0.2 | | 11.4 | **37.5** |

| | GDGT-0 | 1 | 2 | 3 | 4 | 5 |
|---|---|---|---|---|---|---|
| | | | relative distribution | | | |
| MH | 29.6% | 5.5% | 5.8% | 1.6% | 0.0% | 57.5% |
| DH | 19.8% | 7.2% | 16.0% | 7.9% | 0.0% | 49.1% |
| HPH | 61.9% | 5.3% | 1.9% | 0.5% | 0.0% | 30.4% |

**Lengger et al. 2014, sample P3000 0-2 cm, Supplementary Table A3**

| | GDGT-0 | 1 | 2 | 3 | 4 | 5 | sum |
|---|---|---|---|---|---|---|---|
| | | | GDGT concentrations, ng/g sed dw. | | | | |
| MH | 70.2 | 4.9 | 4.3 | 1 | | 57.7 | **138.1** |
| DH | 13.9 | 4.1 | 6 | 2.5 | | 19.3 | **45.8** |
| HPH | 188 | 4.1 | 0.6 | 2.1 | | 56.7 | **251.5** |

| | GDGT-0 | 1 | 2 | 3 | 4 | 5 |
|---|---|---|---|---|---|---|
| | | | relative distribution | | | |
| MH | 50.8% | 3.5% | 3.1% | 0.7% | 0.0% | 41.8% |
| DH | 30.3% | 9.0% | 13.1% | 5.5% | 0.0% | 42.1% |
| HPH | 74.8% | 1.6% | 0.2% | 0.8% | 0.0% | 22.5% |

885/900 MH-GDGT

1306/1300 MH-GDGT

3003/3000 MH-GDGT

885/900 DH-GDGT

1306/1300 DH-GDGT

3003/3000 DH-GDGT

885/900 HPH-GDGT

1306/1300 HPH-GDGT

3003/3000 HPH-GDGT

---

## Author Response (AR2)

**Comments to the third report of reviewer 1 and additional modifications to the manuscript.**

We thank the reviewer for the comments. The reviewer raises several concerns regarding our manuscript to which we would like to respond below.

1.  In their response letter, Besseling et al. did not respond to 16 of my initial comments (pertaining to lines 360-435 and Figure 1 of the original manuscript). Further, they should try to better implement some of the suggested changes to their manuscript, as detailed below.

*We sincerely apologize for not responding to 16 of the comments of reviewer 1. An unfortunate copy/pasting error let to the omission of the comments in our initial rebuttal. Our rebuttals to the initial comments are now included below.*

2.  Line 402-404: The authors need to include their assessment of ionization biases in the manuscript as this information is crucial for comparison with future studies: "We have assessed the difference between the HPLC-MS method using a diol-column used for the study here and the one that uses a reversed phase LC column (Wörmer et al., 2013) using a fresh sample of North Atlantic SPM. Looking at the ratio of MH:DH:HPH for cren and GDGT-0, the NP IPL method may underestimate the MH IPLs by a factor of 10. However, even when taking this into account, this does not change the conclusion of this study. We have focused our discussions on the HPH variety of the IPLs as this is the best life marker among the IPLs. MH-GDGTs are certainly produced directly by archaea but can also have a sizable fossil contribution and are decay products of DH- and HPH-GDGTs, especially in stored extracts. For the anoxic surface sediment at 885 mbsl we observed a discrepancy between the relative abundance of IPL derived CL-GDGT-0 and the IPL-GDGT-0 relative abundance reported in our manuscript. Even if our method underestimates the relative abundance of MHs, this discrepancy cannot be fully explained. We, therefore, remained this part in our discussion. However, we also highlight the fact that we underestimate the relative abundance of MHs in our study."

*We have incorporated the results of the experiment in which we compare the two methods in the Supplementary Information. In our opinion it does not fit well within the main text, but in the SI it is available to the reader and for comparison with future studies. We added a couple of sentences concerning this addition (lines 404-407, revised version).*

3.  Line 429-431: The authors changed their sentence but still omit the fact that the lipid type has previously been detected in marine sediments (Lipp and Hinrichs, 2009; Sturt et al., 2004) as pointed out by reviewer 2.

*In these lines, we discuss the potential source organisms of the GDGT-0 with a cyclopentanetetraol headgroup and not where the IPL has previously been detected. Therefore, we already mentioned Sturt et al., 2004, the reference proposed by the reviewer.*

4.  Line 121-125: As suggested by reviewer 2, the authors need to show the data of their comparison with the Lengger et al. (2012) paper in a supplementary table.

*As stated below to the response to reviewer 2, we do not agree that both dataset are easily comparable. Besides the sampling resolution issue, Lengger et al. (2014) used semi-preparative*

*HPLC in order to obtain the GDGT distribution of different isolated GDGT-IPLs (MH, DH and HPH). This was the preferred method at the time. However, as we now know, there are numerous GDGT-IPL types (e.g. OH-GDGTs) that may co-elute with the isolated fractions and add their core lipids to those of the intended IPL class after hydrolysis, thus biasing the results.*

Other comments:

5. Supplement: I appreciate the addition of chromatograms to the supplement. For this to be of greater use to future applications of their method, the authors should add a base peak chromatogram showing the elution times of all quantified compounds relative to a standard.

*We believe that a base peak chromatogram would not benefit the manuscript. We analyzed environmental samples with countless compounds, our analyzed compounds would not be clearly visible among these other compounds. All quantified compounds were shown in extracted ion currents (EIC; Fig. S2 and S3 in our revised manuscript) which also shows the corresponding retention times. The retention time of the used internal standard (PAF; 1-O-hexadecyl-2-acetoyl-sn-glycero-3-phosphocholine) is now stated in the figure caption of this supplemental figure 3 providing a reference point to the reader*

6. Table 1: Explain abbreviations "BWO" "OPD" in caption.

*We added the abbreviations in the caption.*

7. Line 387-389: Any experimental evidence for the higher preservation potential?

*We added a sentence in our revised version concerning this comment (lines 390-393 of the revised version).*

Comments from the original review that need to be addressed (line numbers pertaining to original manuscript):

*Again we sincerely apologize for the omission of the comments and rebuttals in earlier documentations. We would like to address the reviewers comments down below.*

8. Line 360: Qin et al. did not study IPLs. A more appropriate reference would be Elling et al. (2017).

*We agree with the reviewer. We added Elling et al. (2017) to our revised manuscript. Qin et al., reported a dominance of (core lipid) GDGT-0 and crenarchaeol in their pure cultures. However the reviewer is right, they didn't report the IPL signal, therefore we removed this reference.*

9. Line 360-363: I am quite confident that the predominance of HPH-GDGTs in the sediment is an artifact of the chromatographic method that results in severe underestimation of MH-GDGT (and potentially DH-GDGT) relative abundances. However, the notion that higher HPH-content relates to higher activity was also supported by culture studies (Elling et al., 2014).

*We acknowledge our underestimation of MH-IPLs with the normal phase method. The references placed in line 360-363 apply to the dominance of GDGT-0 and crenarchaeol with HPH, there was no statement about the activity. However, we appreciated the proposed reference to highlight the active thaumarchaeota community detected in some of our studied sediments. We added this to our manuscript.*

10. Line 363: Neither Harvey et al. or Schouten et al. discuss HPH IPLs and do not provide any experimental evidence for degradation rates of HPH versus MH or DH or any other GDGT IPL types. Rephrase.

*The reviewer is right. We altered this in our revised manuscript accordingly. Similar remarks were previously made by reviewer 2 and so these changes were made earlier already.*

11. Line 369-371: Do you think these fossil IPLs extracellular or intracellular (Braun et al., 2016)? Is some of the DNA also fossil? Would the DNA be preserved differently (degradation rates) than the IPLs and how would this affect the interpretation of your results?

*We detected amoA gene fragments within the anoxic surface sediment at 885 mbsl (Fig. 5b). However, no amoA gene transcripts were detected within this sediment, so likely thaumarchaeota activity is none existing or below detection level. Therefore the detected amoA gene fragments (DNA) and perhaps other DNA molecules were likely fossil components. We did not determine if these detected IPLs and DNA molecules were extracellular or intracellular. Based on the detection of these compounds in the surface sediment within the OMZ and the extent of the OMZ within the water column it could well be that both compounds were also present intracellular within this specific sediment. There are clear advantages to cell separation techniques as used by Braun et al. (2016). However, cell extraction techniques have the disadvantage that they extract inconsistent cell numbers (Schippers et al., 2010) and lower cell numbers (Braun et al., 2016) compared to "direct" analyses. With regards to the different degradation rates of IPLs and DNA. There are varies IPL types and sources, the same goes for DNA molecules, they have almost certainly different degradation rates already within the two groups. It would be too speculative to discuss any rate differences within our samples, under different environmental conditions.*

12. Line 372-373: How did the 0/crenarchaeol ratio change for the anoxic sites? If they change in a similar way, does that point to different sources of 0 versus crenarchaeol or to accumulation of fossil IPLs?

*There was no significant correlation (p > 0.05) between the relative abundance of GDGT-0 and crenarchaeol within the anoxic subsurface sediments. The significant correlation between the relative abundance of GDGT-0 and Crenarchaeol ($r^2$ = 0.94, p = 0.028) within the surface sediments was caused by the increase of HPH-GDGT-0 relative abundance and the shift of DH-crenarchaeol towards HPH-crenarchaeol with increasing depth. This coincided with the increasing bottom water oxygen and oxygen penetration depths. The higher relative abundances of HPH-GDGT-0 within the subsurface (anoxic) samples at 2470 and 3003 mbsl compared to those from 885 and 1306 mbsl could be accumulation of fossil IPLs. However, we do not have the archaeal community composition of these sediments to compare this.*

13. Line 380: Typo. "Acid hydrolysis"

*We altered this in the manuscript*

14. Line 384-386: You should discuss here or elsewhere that some cultures exist from the Thermoplasmatales cluster, although most clades remain uncultivated. E.g. there are many (acidophilic, thermoacidophilic) Thermoplasmatales cultures for which lipids have been analysed and all of them produce GDGTs. Further, the recently cultivated Methanomassiliicoccales (closely related to the uncultivated TMEG group) have been shown to produce IPL-GDGTs such as MH-GDGT-0 (Becker et al., 2016). It is thus very likely that the rest of the uncultivated Thermoplasmatales-like archaea can produce GDGTs.

*We are aware of various Thermoplasmatales clusters/species capable to produce GDGTs. However, the Thermoplasmatales group is a very diverse group with numerous subgroups, some of these, as mentioned by the reviewer, are cultivated and are known GDGT producers. The Thermoplasmatales subgroups that we detected and reported in our manuscript are, to the best of our knowledge, not yet cultivated and therefore their lipid composition remains unknown. We agree with the reviewer that because other Thermoplasmatales groups are capable of GDGT production it would be expected in our reported subgroups. We, therefore, highlighted in our conclusion (line 433-439, original manuscript) that Thermoplasmatales members could be the source of IPL GDGTs in our analyzed sediments.*

15. Line 388: The way these papers are referenced is highly misleading. Four of the five references do not relate to Woesearchaeota and the inference of Villanueva et al. (2017) that Woesearchaeota do not produce GDGTs is circumstantial at best without knowledge of the actual GDGT biosynthetic pathway. Rephrase.

*The four references in our manuscript refer to the missing genes known to be playing a role in the GDGT biosynthetic pathway, i.e. those gene coding for the prenyl synthases involved in the ether bond formation between glycerol-1-phosphate and the isoprenoid side chain, as well as others involved in the isoprenoid synthetic pathwway. We acknowledge that there is only limited information known about the GDGT biosynthetic pathway in general and especially for the uncultivated Woesearchaeta. However, the genes that are currently linked to archaeal lipid biosynthesis are so far not found in Woesearchaeota genomes and, therefore, it is to be expected that they do not synthesize archaeal lipids.*

16. Line 392-394: Becker et al. (2016) showed that BDGTs are detectable in a globally distributed set of marine sediments. It is therefore likely that these compounds would be present in your samples. I suspect that you would detect these compounds using different chromatographic conditions. Recently, Thermoplasma-related methanogens (Methanomassiliicoccales) have been identified as a source of BDGTs in the environment (Becker et al., 2016). Meador et al. (2015) also identified further phosphatidic and MH, DH-GDGTs in MCG-rich samples. Could the MCG be sources of these compounds in your samples?

*Concerning the BDGTs, we replied to a similar comment (reviewer 1, comment #1) within the first rebuttal. We targeted a wide range of phosphatidic compound, as for example, phosphatidyl glycerol (PG). However, these were not detected within our samples. As for the MH and DH-GDGTs, it could well be that MCG members were a source within our sediments. We also detected different MCG community compositions (Table 4) within these sediments. Certain MCG subgroups that were present within the anoxic subsurface sediments were not detected within the anoxic surface sediment at 885 mbsl. This coincided with the presence/absence of the encountered DH-GDGT-0 isomers. However, any specific source assignation would be too speculative.*

17. Line 416: As stated earlier, DH-GDGT-0 isomers have been detected in Thaumarchaeota (Elling et al., 2014, 2015).

*We added references in our previous revised manuscript. Line 176 (revised manuscript): "These isomers were previously also reported in thaumarchaeotal cultures (Elling et al., 2014, 2017)". The reviewer stated the references Elling et al. 2014 and 2017 within his second review.*

18. Line 422: Typo. "sediments"

*We modified the manuscript accordingly.*

19. Line 423: Also Elling et al. (2014).

*Based on the advice of reviewer 2, we decided to remove all references in the conclusion section.*

20. Line 430: Please provide experimental evidence for the "recalcitrant nature of glycosidic bonds".

*As stated in our previous comment, we removed all references within the conclusion section. We agree with the reviewer on the original phrasing. We modified part of the discussion on this topic in our revised version, also based on comments of reviewer 2.*

21. Line 433-434: Again, very misleading references, as in Line 388. These works do not discuss the GDGT lipid biosynthetic pathway. Delete the sentence or rephrase.

*The references were removed, see earlier comments about the conclusion section. We agree with the reviewer that the original references were not mentioning the GDGT biosynthetic pathway. We referred to the lipid biosynthetic pathway, as mentioned in Podar et al. (2013): "In particular, similar to N. equitans, the reduced genome of Nst1 does not encode functional pathways for de novo biosynthesis of lipids, amino acids, coenzymes or nucleotides" and in Waters et al. (2003): "This organism lacks almost all known genes that are required for the de novo biosynthesis of amino acids, nucleotides, cofactors, and lipids". We therefore support this statement in our manuscript.*

22. Line 435: I disagree: Could you please specify how the source assignment of IPL-GDGTs has been improved?

*A direct source of IPLs could not be given as explained in the manuscript. However, the presence of certain IPLs and isomers in connection to the presence of archaeal groups was highlighted. Further investigations could perhaps link specific IPLs with certain (sub)groups of the archaeal domain. Studying environmental samples, like marine sediments, with a diverse archaeal community can derive valuable information on IPL source assignation. Especially since many archaeal (sub)groups have no cultivated representative.*

23. Figure 1: It would be very helpful to have another set of two plots (surface, subsurface) next to A and B that shows summed IPLs by headgroup type or a synthesis of the data in Table 2.

*We have table 3 in our manuscript, which shows the relative abundance of IPL-GDGTs grouped by polar headgroup. This includes all samples, surface and subsurface sediments.*

**Comments to the second report of Dr. Lipp and additional modifications to the manuscript.**

We thank the reviewer for the comments. The reviewer raises several concerns regarding our manuscript to which we would like to respond below.

I would like to thank the authors for adding more information about the analytical method regarding peak separation and the applied quantification method. It is now much easier to assess the results. Many of the previous analytical concerns have been alleviated by the newly introduced supplementary figures. Thank you also for the more balanced discussion of the stability of glycolipids and phospholipids and for addressing the other concerns of both reviewers.

However, my second main concern remains in the revised version, and I think the authors have missed the chance to clarify some of the discrepancies of the new data and the previous results (reviewer 2, general comment 2 and detailed comments 14). I do not understand the "good correspondence between the datasets when it comes to the DH and HPH IPLs" as this is not further explained. Instead, when comparing the relative ring distribution of Lengger et al. (2014) and the current study within each headgroup class, MH-GDGTs show the best resemblance, whereas DH- and HPH-GDGTs differ somewhat (see plots in attached PDF of Excel table). The lack of GDGT-0 in DH-GDGTs of sample 885 and 1306 and of HPH-GDGT in sample 885 in Besseling et al. is curious. Both studies used normal phase chromatography with diol column chemistry to separate the IPL-GDGT classes. If an unknown IPL-GDGT-0 is responsible for the observed GDGT-0 in the hydrolyzed DH- and HPH-GDGT fractions of Lengger et al. (2014) it would likely also elute in a similar retention time range on the diol column of Besseling et al. (between 19 and 35 min where DH- and HPH-GDGTs elute) and should therefore be visible in the chromatogram. Possibilities why the unknown IPL-GDGT-0 is not visible in the chromatogram are (a) they elute very early with the column void volume, (b) they elute very late and are retained on the column, and (c) the response of this compound is very low and they are "invisible" on the MS. Scenario (a) and (b) could be tested by reversed phase analysis to gain more insight into polarity ranges outside the optimal RT range for the diol column. Since the authors have access to RP chromatography as they state in the rebuttal letter this should not be too time consuming.

*We thank the reviewer for taking the time and effort to compare both datasets. However, besides the resolution issue, are both datasets not comparable. Lengger et al. (2014) used semi-preparative HPLC in order to obtain the GDGT distribution of different isolated GDGT-IPLs (MH, DH and HPH). This was the preferred method at the time. However, as we now know, there are numerous GDGT-IPL types (e.g. OH-GDGTs) that may co-elute with the isolated fractions and add their core lipids to those of the intended IPL class after hydrolysis, thus biasing the results.*

*We have intensively searched for an additional source of GDGT-0 in the IPLs using the diol chromatography. This included extending the gradient to higher end % of eluent B, extending the mass range (up to 6000 m/z), and performing a so called "all fragments-all the time" experiment where throughout the chromatogram (including the void volume), the entire mass range is send to the collision chamber and all fragments are then analyzed in the orbitrap. None of these experiments yielded additional fragments that could be related to unknown sources of GDGT-0. We do indeed have access to the revered phase column, but unfortunately our Q-Exactive HRMS is currently under repair and we can therefore not perform the requested experiment within the time frame allowed for this rebuttal.*

I agree that the differences in sampling resolution might explain some of the differences between the two studies. However, I would not expect a dramatic heterogeneity within 1.5 cm of sediment that could lead to the observed large differences. This explanation would require a 0-0.5cm layer almost devoid of GDGT-0 (to explain Besseling et al.) and an underlying 0.5-2cm layer loaded with GDGT-0 (to explain the integrated signal 0-2 cm in Lengger et al. 2014) which I find somewhat unlikely. Instead, Lengger et al. 2014 show similar GDGT distributions for the three studied surface sediment samples (0-2 cm depth), even within each IPL-class. These distributions also closely resemble GDGT distributions observed by direct analysis (without hydrolysis, similar to the Besseling et al. method) in pure cultures of thaumarchaea (e.g. Elling et al., 2014 GCA) which are abundant in the 885 sample (~30%, Fig. 1). What is the explanation for the 50% contribution of GDGT-2 in DH-GDGT of sample 3003? Can this unusual value be trusted?

*The heterogeneity could be due to the oxygen penetration depth (OPD), causing heterogeneity within the first mm of the core. OPD at the surface sediment at 885 mbsl is 0.1 mm, indicating an almost fully anoxic surface sediment. However, at 1306 mbsl the OPD is 2.9 mm resulting in a partly oxygenated/anoxic sediment, this is clearly reflected in the archaeal community composition (mixture of an oxygenated sediment archaeal community with an anoxic sediment archaeal community). This would most likely have an impact when sampling with a 0-2 cm resolution, probably showing an anoxic sediment archaeal community composition. This is also the case in the other surface sediments from deeper waters (OPDs of 9.8 and 19.0 mm).*

*Concerning the rather high relative abundance of DH-GDGT-2 within the sample at 3003 mbsl. We detected a relative abundance of DH-GDGT-2 between 0.8 % and 29.5 % (of the sum of IPLs; table 2), this relative abundance decreased with increasing depth / increasing bottom water oxygen / increasing oxygen penetration depth. We have checked the chromatograms multiple times and it does not seem odd. The relative abundance of DH-GDGT-2 within the anoxic subsurface sediments were also consistent (21.9% -29.7 %; table 2).*

My impression from the authors comments and the manuscript is that (a) MH abundance is problematic (perhaps up to factor 10 underestimated, this needs to be pointed out in the manuscript) and (b) IPL-GDGT-0 abundance is questionable for some headgroup classes. This implies that not all new data can be compared to the 2014 data which is unfortunate. Which dataset (2014 or the present one) is more credible? And which ring and IPL headgroup distribution should be referenced in future studies if the differences are so large?

*We have replied to point (a) in our reply to reviewer 1 and below. With regard to point (b): it is indeed odd not to find GDGT-0 in a certain class of IPLs as it is often a major core lipid (although depending on the strains analyzed and the specific conditions such as temperature). However, in the samples where the GDGT-0-IPLs could not be detected (for example 885, 0-0.5 cm), we were capable of detecting GDGT-1, -2, -3, 4 and crenarchaeol with MH, DH and/or HPH head groups. This data comes from the same analytical run and as the various core lipids with the same head group elute closely together it seems highly unlikely that the MS was not functioning only for the brief moment that the IPL with the GDGT-0 core was eluting or that the ionization conditions were dramatically different. We can therefore only conclude that the GDGT-0-IPLs were therefore apparently below the detection limit if present at all.*

*Regarding the question which dataset is most reliable and should be referenced: the dataset presented here was produced with improved chromatography and state of the art mass spectrometry and is also a much more direct way of determining relative abundances (in MS1) compared to the SRM MS method or the preparative HPLC/hydrolysis approach used. By using improved analytical approaches we may observe details not observed before or change our insights. We would not attempt to publish this work if we did not believe it was an improvement on previous work on these samples.*

Detailed comments

1. Line 106: delete "ppm"

*We modified the manuscript accordingly*

2. Line 115: typo, "assess"

*We modified the manuscript accordingly*

3. Line 121-123 and Line 213-216: I assume the reference should be Lengger et al. (2014)?

*Lengger et al. (2012) is a correct reference. They used the same method for extracting and analyzing IPL-derived CL-GDGTs as described in Lengger et al. (2014).We chose to refer to the earlier paper.*

4. Line 294: "and, therefore, are likely"

*We modified the manuscript accordingly, we added "is" because it refers to the OM in singular.*

5. Line 405: typo, "hydrolysis". The sampling depth in Lengger et al. (2014) was different  0-2 cm as pointed out earlier.

*We modified the manuscript accordingly, also based on the comments of reviewer 1. About the resolution issue. The resolution in the Lengger et al. (2014) paper was 0-2 cm for the semi-preparative HPLC. However, they also determined IPL-derived CL-GDGTs, on a higher sampling resolution (0-0.5 cm). We also analyzed a 0-0.5 cm sample, with the same resolution, this to compare the distribution of IPL-derived CL-GDGTs and check for any degradation, that was perhaps induced during storage.*

6. Line 408: the underestimation of MH is mentioned in the rebuttal letter (reviewer 1, 2nd comments, 2) but is not included in the manuscript. This should be added to the method description or to the results. I do not agree with the authors' statement to "retain a paragraph of discussion on this". It is an important result necessary to judge the results.

*In line 402-403 of the previously revised manuscript we refer to the underestimation of MH IPLs by our analytical method. We have now include supplementary information (supplementary material and method plus a supplementary figure to compare both methods) to detail the underestimation and refer to it in the text (line 404-407).*

7. Line 448: typo "sediments"

*We do not see the typo in this line as suggested by the reviewer. We refer to sediments in plural because we discuss multiple subsurface anoxic sediments.*

8. Table 2: check that numbers add up to 100% for each sample (e.g., sample 885-crenarchaeol: 1.3+43.1+0.3=46.7, also sum for sample 885=100.2%)

*This is a standard problem with rounding numbers. It can be solved by adding another decimal, then the "rounding error" decreases 10 fold to e.g. 100.01. However adding another decimal to the table would not benefit the clarity of the table it will also give a false indication of analytical precision.*

[revised manuscript text omitted]

0.02

[Figure]

**Fig. 3.**

[Figure]

**Fig. 4.**

[Figure]

[Figure]

---

## Author Response (AR3)

**Comments to the report of reviewer 1 and additional modifications to the manuscript.**

*We thank the reviewer for the additional comments. The reviewer raises several concerns regarding our manuscript to which we would like to respond below.*

1. Although the authors go to great length rebutting the comments of both reviewers, for most instances their reasoning is not reflected in the revised manuscript (in particular regarding the detailed technical explanations in the second report of reviewer 2). Although the manuscript would be acceptable for publication after minor revisions, I am taken aback by the authors' reluctance to make meaningful changes to the text after three revisions. The review process should provide an opportunity for the authors to make their reasoning more accessible to the readers instead of trying to just brush off criticism. We want to understand your research and help make it more impactful. It is sad that the authors did not grasp this opportunity.

*We are sorry that the reviewer feels that we missed the chance to improve our manuscript during the past revisions. We really appreciate the detailed comments by both reviewers during this lengthy process, but we also feel that we also addressed these comments with great detail and modified our manuscript accordingly. The reviewer also recommended to re-analyze the samples, which as discussed before is not feasible due to the decay of the compounds in the already extracted samples. However, in our last reply to the reviewer's comments we included new data comparing the different analytical methods by using other freshly extracted samples. This addition contributes to the clarity of the manuscript and addresses the reviewer's concerns. Also, we have added many caveats and statements to our manuscript regarding our analytical approach, such as the lack of quantitation and the underestimation of the MH-GDGTs. Furthermore, we have addressed all line by line comments and made modifications as recommended. We believe this demonstrates our willingness to consider the reviewer's comments. However, it is clear we have differing scientific views on the topic of this paper and we retain the right to express our views.*

Line comments:

2. Line 347-357: I would suggest adding the recent paper by Yu et al. (2018, PNAS) to this discussion, which shows putative growth of group 8 MCG archaea on lignin and incorporation of bicarbonate into GDGTs.

*We have added the recent paper by Yu et al. (2018). It is a very interesting study, which sheds some light on the factors that drive (or do not drive) Bathyarchaeota in marine sediments. However lignin is not present in substantial concentration in our studied subsurface samples (cf. Cowie et al., 1999). Therefore, it does not explain the relative abundances of MCG-8 (Bathy-8) in our archaeal compositions. We also added the reference of Cowie et al. (1999) to our manuscript.*

3. Line 393: Please be precise on the details here and rephrase. Logemann et al. did not study GDGT degradation. It would be useful to discuss Xie et al. (2013, PNAS), who also studied archaeal glycolipids degradation. It is also not clear how the Logemann experiment would be relevant to the discussion of relative degradation rates of phospho- and glyco-GDGTs (or glyco vs. phosphor ether lipids), as they did not study this process.

*We have modified the text accordingly. We have added the proposed reference.*

4. Line 410-411: Please rephrase. After the last revision, it is no longer clear what "this discrepancy" refers to.

*We rephrased this sentence in the manuscript.*

5. Line 414-417: Following our discussion in the review and rebuttal, rephrase "GDGT biosynthetic pathway" to "archaeal lipid biosynthetic pathway", since only this statement is reasonably supported by the references. Alternatively, re-arrange the references to match each statement regarding lack of "archaeal lipid biosynthetic pathway (Jahn et al., 2004; Podar et al., 2013; Waters et al., 2003)" vs. "putatively lack the GDGT biosynthetic pathway (Villanueva et al., 2017)"

*We rephrased the sentence. Villanueva et al. (2017) also refers, as the other references, to the archaeal biosynthetic pathway.*

[revised manuscript text omitted]

0.02

[Figure]

**Fig. 3.**

[Figure]

0.2

    **Fig. 4.**

[Figure]

**Fig. 5.**

[Figure]